# A Free Lunch with Influence Functions? Improving Neural Network Estimates of the Average Treatment Effect with Concepts from Semiparametric Statistics

## Abstract

Parameter estimation in empirical fields is usually undertaken using parametric models, and such models readily facilitate statistical inference. Unfortunately, they are unlikely to be sufficiently flexible to be able to adequately model real-world phenomena, and may yield biased estimates. Conversely, non-parametric approaches are flexible but do not readily facilitate statistical inference and may still exhibit residual bias. Using causal inference (specifically, Average Treatment Effect estimation) as an application domain example, we explore the potential for Influence Functions (IFs) to (a) improve initial estimators without needing more data (b) increase model robustness and (c) facilitate statistical inference. We begin with a broad, tutorial-style introduction to IFs and causal inference, before proposing a neural network method 'MultiNet', which seeks the diversity of an ensemble using a single architecture. We also introduce variants on the IF update step which we call 'MultiStep', and provide a comprehensive evaluation of different approaches. The improvements are found to be dataset dependent, indicating an interaction between the methods used and nature of the data generating process. Our experiments highlight the need for practitioners to check the consistency of their findings, potentially by undertaking multiple analyses with different combinations of estimators. This finding is especially relevant to practitioners working in the domain of causal inference, where ground-truth with which one could assess the performance of one's estimators is not available.

## 1 Introduction

Most methods being utilized in empirical fields such as psychology or epidemiology are parametric models (van der Laan & Rose, 2011; Blanca et al., 2018), which are convenient because they facilitate closed-form statistical inference and confidence intervals (*e.g.* for the purpose of null hypothesis testing). Indeed, being able to perform statistical tests and reliably quantify uncertainty is especially important when evaluating the efficacy of treatments or interventions. One approach to perform such tests is by assuming a parametric model for the underlying generating mechanism, and *e.g.* normally distributed estimates. However, it has been argued that linear models are incapable of modeling most realistic data generating processes and that we should instead be using modern machine learning techniques (van der Laan & Rose, 2011; van der Laan & Gruber, 2012; van der Laan & Starmans, 2014; Vowels, 2021). Unfortunately, most machine learning models are non-parametric insofar as the estimates derived using such techniques are not directly parameterizable as (*e.g.*) a Gaussian with a mean and variance. As such, the estimates derived using such non-parametric techniques are not readily amenable to null-hypothesis significance testing or other common statistical inference tasks. Furthermore, even though machine learning algorithms are more flexible, they are still likely to be biased because they are not targeted to the specific parameter of interest (van der Laan & Rose, 2011). So, what can we do?

By leveraging concepts from the field of semiparametric statistics we can try to address these issues. Indeed, by combining elements of semiparametric theory with machine learning methods, we can, at least theoretically, enjoy the best of both worlds: We can avoid having to make unreasonably restrictive assumptions

about the underlying generative process, and can nonetheless undertake valid statistical inference. Furthermore, we can also leverage an estimator update process to achieve greater precision in existing estimators, without needing additional data (van der Laan & Rose, 2011; Tsiatis, 2006; Bickel et al., 1998), an advantage which we might call a 'free lunch'[1], and one we wish to explore and test empirically in this paper.

One example of an existing method which combines machine learning and semiparametric theory is targeted learning (van der Laan & Rose, 2011; van der Laan & Starmans, 2014).[2] Unfortunately, this technique, and many related techniques involving influence functions (IFs) and semiparametric theory, have primarily been popularized outside the field of machine learning. In parallel, machine learning has focused on the development of equivalent methods using deep neural network (NN) methods for causal inference (see *e.g.*, Bica et al., 2020; Wu & Fukumizu, 2020; Shalit et al., 2017; Yoon et al., 2018; Louizos et al., 2017), which, owing to their 'untargeted' design (more on this below), may exhibit residual bias. As such, many of the principles and theory associated with semiparametrics and IFs are underused and underappreciated within the machine learning community, and it remains unknown to what extent these techniques can be applied to NN based estimators.

More generally, and in spite of a large body of work describing the theoretical properties of semiparametric methods for estimation outside of machine learning, there has been little empirical comparison of techniques like targeted learning against those considered state of the art at the intersection of machine learning and causal inference. In particular, there now exist numerous NN based methods, and practitioners may find themselves choosing between the alluring 'deep learning' based methods and those which perhaps, rightly or wrongly, have less associated hype. Such a comparison is therefore extremely important, especially given that a theoretical framework for establishing the statistical guarantees of NNs is yet elusive (Curth et al., 2021a), although one notable recent contribution is presented by Farrell et al. (2021).

We explore the potential for semiparametric techniques, in particular, various applications of IFs, to (a) improve the accuracy of estimators by 'de-biasing' them, (b) yield estimators which are more robust to model misspecification (double-robustness), and (c) derive confidence intervals for valid statistical inference. Our motivating application example is chosen but not limited to be the estimation of the causal effect of a treatment or intervention on an outcome from observational data. Experiments highlight that, even for simple datasets, some NN methods do not yield estimators close enough to be amenable to improvement via IFs (as we will discuss below, the assumption is that the bias of the initial estimator can be approximated as a linear perturbation). We propose a new NN pseudo-ensemble method 'MultiNet' with constrained weighted averaging (see Fig. 1) as a means to adapt to datasets with differing levels of complexity, in a similar way to the Super Learner ensemble approach (van der Laan et al., 2007), which is popular in epidemiology.

The associated contributions of this paper are:

- A top-level introduction to the basics behind semiparametric theory, influence functions (including an expression for deriving influence functions for general estimands and the code to do so automatically)[3], and causal inference.
- A new method 'MultiNet' which attempts to mimic the performance of an ensemble with a single NN
- A new update step method 'MultiStep' which attempts to improve upon existing update methods by continuously optimizing the solution according to two criteria which characterize the optimum solution (namely, finding the IF with the smallest expectation and variance)
- An extensive comparison of the estimation performance of NNs and other algorithms with and without semiparametric techniques

We evaluate causal inference task performance in terms of (a) precision in estimation (and the degree to which we can achieve debiasing), (b) double robustness, and (c) normality of the distribution of estimates (thus,

---

[1]The term 'free lunch' is a reference to the adage of unknown origin (but probably North American) 'there ain't no such thing as a free lunch'. It was famously used by Wolpert and Macready in the context of optimization (Wolpert & Macready, 1997).

[2]For an overview of some other related methods see (Curth et al., 2021a).

[3]Code for models, experiments, and automatic IF derivation is provided in supplementary material.

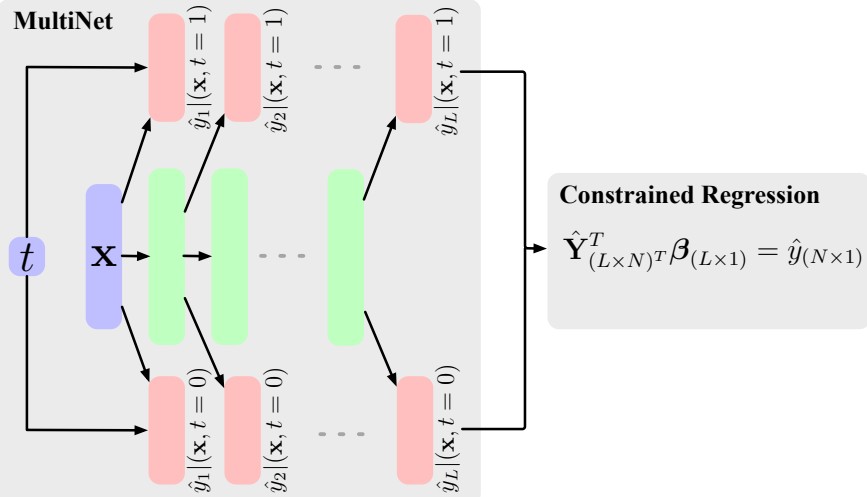

Figure 1: Block diagram for MultiNet. At each layer $l = \{1, ..., L\}$ of the network, the outcome $y$ is estimated using covariates $\mathbf{x}$ (which can include treatment $t$). The treatment is used to select between two estimation arms. Once the network has been trained, the outcomes from each layer are combined and a constrained regression is performed. The weights $\boldsymbol{\beta}$ in the regression are constrained to be positive and sum to 1. An equivalent single-headed network can be used for the treatment model $\hat{t}|\mathbf{x}$.

by implication, whether it is possible to use closed-form expressions for confidence intervals and statistical inference). We find our MultiNet and MultiStep methods provide competitive performance across datasets, and observe that initial estimation methods can sometimes benefit from the application of the semiparametric techniques. In general, however, the improvements are both dataset and technique dependent, highlighting possible interactions between the underlying data generating process, sample sizes, and the estimators and update steps used. The conclusion is thus that practitioners should take care when interpreting their results, and attempt to validate them by undertaking multiple analyses with different estimators. This is particularly important for the task of causal inference where, in real-world applications, ground truth is unlikely to be available.

The paper is structured as follows: We begin by reviewing previous work in Sec. 2 and provide background theory on the motivating case of estimating causal effects from observational data in Sec. 3. In this section, we also provide a top level introduction to IFs (Sec. 3.2) and a derivation of the IF for a general graph (Sec. 3.3). In Sec. 4 we discuss how to use IFs debias estimators and we present our own update approach MultiStep. Our NN method MultiNet is presented in Sec. 5. The evaluation methodology is described in Sec. 6 and at the beginning of this section, we summarise the open questions which inform our subsequent evaluation design. We present and discuss results in Sec. 7 and finally, we provide a summary of the experiments, conclusions, and opportunities for further work in Sec. 8.

## 2 Previous Work

The possible applications of semiparametrics in machine learning are broad but under-explored, and IFs in particular have only seen sporadic application in explainable machine learning (Koh & Liang, 2017; Sani et al., 2020), natural language processing (Han et al., 2020) models, causal model selection (Alaa & van der Schaar, 2019) and uncertainty quantification for deep learning (Alaa & van der Schaar, 2020). Outside of machine learning, in particular in the fields of epidemiology and econometrics, semiparametric methods are becoming more popular, and include targeted learning (van der Laan & Rose, 2011) and the well-known double machine learning approach by Chernozhukov et al. (2018). In statistics, alternatives have been developed which include doubly robust conditional ATE estimation (Kennedy, 2020) and IF-learning (Curth et al., 2021a).

However, within the field representing the confluence of causal inference and machine learning, the focus seems to have been on the development of NN methods (see CEVAE (Louizos et al., 2017), CFR-Net (Shalit et al., 2017), GANITE (Yoon et al., 2018), Intact-VAE (Wu & Fukumizu, 2022) etc.), without a consideration for statistical inference or semiparametric theory, and this gap has been noted by Curth et al. (2021b) and Curth & van der Schaar (2021). Indeed, to the best of our knowledge, the application of semiparametric theory to debias neural network-based estimators has only be used three times in the field representing the confluence of machine learning and causal inference. Firstly, in DragonNet (Shi et al., 2019), a method designed for ATE estimation; secondly in TVAE (Vowels et al., 2021), a variational, latent variable method for conditional ATE and ATE estimation; and thirdly, by Farrell et al. (2021) where a restricted class of multilayer perceptrons were evaluated for their performance potential as plug-in estimators for semiparameteric estimation of causal effects and shown to yield promising performance. The first two methods incorporate targeted regularization, but do not readily yield statistical inference because to do so requires asymptotic normality (and this is not evaluated in the studies) as well as explicit evaluation of the IF. More broadly, semiparametrics has been discussed in relation to theory in machine learning, for example Bhattacharya et al. (2020) provides a discussion of influence functions in relation to Directed Acyclic Graphs with hidden variables, Rotnitzky & Smucler (2020) and Henckel et al. (2020) discuss the application of semiparametric techniques for identifying efficient adjustment sets for causal inference tasks, Zhong & Wang (2021) apply semi-parametrics with deep neural networks to achieve statistical inference in the partially linear quantile regression setting, and Jung et al. (2020) generalize the coverage of work on semiparametric estimation to general causal estimands. However, in general the work is quite sparse, particularly in relation to the applicability of the theory to neural networks, and the accessibility of the relevant theory to general practitioners of machine learning.

As part of this work, we propose a new estimator 'MultiNet' for the ATE which aims to compete with the methods described above. It represents a combination of ideas from two well-known approaches in causal inference - the Super Learner (van der Laan et al., 2007), and CFR-Net, which we already mentioned above. The Super Learner carries some beneficial guarantees which are afforded by the fact that it is an ensemble of candidate learners. By incorporating a diverse range of candidate learners into the Super Learner ensemble, it is able to achieve efficient rates of convergence, and we aim to incorporate the ensemble idea into a new approach using CFR-Net as a backbone. It is worth noting that other ensemble neural network approaches exist, such as the BatchEnsemble approach by Wen et al. (2020), the well-known use of dropout by Gal & Ghahramani (2016). Whilst we incorporate dropout into MultiNet's training procedure, we do not use the dropout itself as a source of ensemble amortisation, but this represents an interesting opportunity for further work. We instead explore 'tapping' each subsequent layer in a neural network to collect intermediate outcome predictions, and combining them with constrained regression the manner of the Super Learner. We indeed observe improvements in performance over the CFR-Net backbone, highlighting that this simple modification can provide enough diversity to yield competitive, ensemble-style performance.

Finally, other comparisons of the performance of semiparametric approaches exist. For example, the robustness of targeted learning approaches to causal inference on nutrition trial data was presented by Li et al. (2021) and includes a useful summary table of previous findings and includes its own evaluations. However, it does not include comparisons with NN-based learners, and seeks the answers to different questions relevant to practitioners in the empirical fields. Another example evaluation was undertaken by Luque-Fernandez et al. (2018) but has a didactic focus. We therefore note the need for increased coverage and exposure to semiparametric theory, particularly at the intersection of causal inference and neural network estimation, as well a need for an evaluation of the application of semiparametric theory to current methods.

## 3 Causal Inference and Influence Functions

### 3.1 Causal Inference

The concepts in this paper are applicable to estimation tasks in general, but we focus on the specific task of estimating a causal effect, which is of the upmost importance for policy making (Kreif & DiazOrdaz, 2019), the development of medical treatments (Petersen et al., 2017), the evaluation of evidence within legal frameworks (Pearl, 2009; Siegerink et al., 2016), and others. A canonical characterization of the problem of

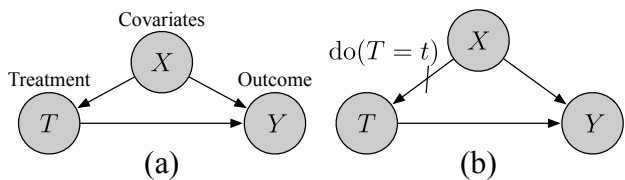

Figure 2: Directed Acyclic Graphs (DAGs) for estimating the effect of treatment $T = t$ on outcome $Y$ with confounding $X$.

causal inference from observational data is depicted in the Directed Acyclic Graphs (DAGs) shown in Fig. 2a and 2b, and we provide an overview of causal inference in this section. We also point interested readers towards accessible overviews by Guo et al. (2020a) and Pearl et al. (2016).

Regarding notation, we use upper-case letters *e.g.* $A, B$ to denote random variables, and bold font, upper-case letters to denote sets of random variables *e.g.* $\mathbf{A}, \mathbf{B}$. Lower-case $a$ and $b$ indicate specific realisations of random variables $A$ and $B$. Specifically, we use $\mathbf{x}_i \sim P(\mathbf{X}) \in \mathbb{R}^m$ to represent the $m$-dimensional, pre-treatment covariates (we use bold symbols to signify multi-dimensional variables) for individual $i$ assigned factual treatment $t_i \sim P(T|\mathbf{X}) \in \{0, 1\}$ resulting in outcome $y_i \sim P(Y|\mathbf{X}, T)$. Together, these constitute dataset $\mathcal{D} = \{[y_i, t_i, \mathbf{x}_i]\}_{i=1}^n$ where $n$ is the sample size, sampled from a 'true' population distribution $\mathcal{P}$. Fig. 2a is characteristic of observational data, where the outcome is related to the covariates as well as the treatment, and treatment is also related to the covariates. For example, if we consider age to be a typical covariate, young people may opt for surgery, whereas older people may opt for medication. Assuming that an age-related risk mechanism exists, then age will confound our estimation of the causal effect of treatment on outcome. One of the goals of a Randomized Controlled Trial (RCT) is to reduce this confounding by making the assignment of treatment (asymptotically) statistically independent of treatment by randomly assigning it. This enables us to compare the outcomes for the people who were treated, and those who were not (or equivalently to compare multiple alternative treatments).

One of the most common causal estimands is the Average Treatment Effect (ATE):

$$\tau(\mathbf{x}) = \mathbb{E}_{\mathbf{x} \sim P(\mathbf{X})}[\mathbb{E}_{y \sim P(Y|do(T=1)\mathbf{X}=\mathbf{x})}[y] - \mathbb{E}_{y \sim P(Y|do(T=0)\mathbf{X}=\mathbf{x})}[y]] \tag{1}$$

Here, the use of the *do* operator (Pearl, 2009) in $do(T = 1)$ and $do(T = 0)$ simulates interventions, setting treatment to a particular value regardless of what was observed. One can also denote the outcomes corresponding with each of these possible interventions as $Y(1)$ and $Y(0)$, respectively, and these are known as *potential outcomes* (Imbens & Rubin, 2015). In practice, we only have access to one of these two quantities for any example in the dataset, whilst the other is missing, and as such the typical supervised learning paradigm does not apply. In Fig. 2b, such an intervention removes the dependence of $T$ on $\mathbf{X}$, and this graph is the same as the one for an RCT, where the treatment is unrelated to the covariates.[4] Using *do*-calculus we can establish whether, under a number of strong assumptions[5], the desired causal estimand can be expressed in terms of a function of the observed distribution, and thus whether the effect is *identifiable*. Causal identification and the associated assumptions are both extremely important topics in their own right, but fall beyond the scope of this paper (we are primarily concerned with estimation). Suffice it to say that for the graph in Fig. 2a, the outcome under intervention can be expressed as:

$$\mathbb{E}_{y \sim P(Y|do(T=t'))}[y] = \int y p(y|\mathbf{X} = \mathbf{x}, T = t') p(\mathbf{X} = \mathbf{x}) d\mathbf{x}, \tag{2}$$

which is estimable from observational data. Here, $t'$ is the specific intervention of interest (*e.g.*, $t' = 1$). In particular, it tells us that adjusting for the covariates $\mathbf{X}$ is sufficient to remove the bias induced through the

---

[4]Of course, one may still observe finite-sample associations between these variables, which occur as a natural consequence of random sample variation.

[5]These assumptions are the Stable Unit Treatment Value Assumption (SUTVA), Positivity, and Ignorability/Unconfoundedness - see Section 3.1.1 below for more information.

'backdoor' path $\mathbf{X} \to T \to Y$. This particular approach is sometimes referred to as backdoor adjustment. Once we have the expression in Eq. 2, we can shift our focus towards its estimation. Note that even once the problem has been recast as an estimation problem, it differs from the typical problem encountered in supervised learning. Indeed, instead of simply learning a function, we wish to indirectly learn the *difference* between two functions, where these functions represent 'response surfaces' - *i.e.*, the outcome/response under a particular treatment.

### 3.1.1 Causal Assumptions

The causal quantity can be estimated in terms of observational (and therefore statistical) quantities if a number of strong (but common: Yao et al., 2020; Guo et al., 2020b; Rubin, 2005; Imbens & Rubin, 2015; Vowels et al., 2021) assumptions hold: (1) Stable Unit Treatment Value Assumption (SUTVA): the potential outcomes for each individual or data unit are independent of the treatments assigned to all other individuals. (2) Positivity: the assignment of treatment probabilities are non-zero and non-deterministic $P(T = t_i | \mathbf{X} = \mathbf{x}_i) > 0, \forall\ t, \mathbf{x}$. (3) Ignorability/Unconfoundedness/Conditional Exchangeability: There are no unobserved confounders, such that the likelihoods of treatment for two individuals with the same covariates are equal, and the potential outcomes for two individuals with the same latent covariates are also equal s.t. $T \perp\!\!\!\perp (Y(1), Y(0)) | \mathbf{X}$.

### 3.1.2 Estimation

One may use a regression to approximate the integral in Eq. 2, and indeed, plug-in estimators $\hat{Q}$ can be used for estimating the ATE as:

$$\hat{\boldsymbol{\tau}}(\hat{Q}; \mathbf{x}) = \frac{1}{n} \sum_{i=1}^{n} (\hat{Q}(T = 1, \mathbf{X} = \mathbf{x}_i) - \hat{Q}(T = 0, \mathbf{X} = \mathbf{x}_i)), \tag{3}$$

We use the circumflex/hat ($\hat{\ }$) notation to designate an estimated (rather than true/population) quantity. In the simplest case, we may use a linear or logistic regression for the estimator $\hat{Q}$, depending on whether the outcome is continuous or binary. Unfortunately, if one imagines the true joint distribution to fall somewhere within an infinite set of possible distributions, we deliberately handicap ourselves by using a family of linear models because such a family is unlikely to contain the truth. The consequences of such model misspecification can be severe, and results in biased estimates (Vowels, 2021; van der Laan & Rose, 2011). In other words, no matter how much data we collect, our estimate will converge to the incorrect value, and this results in a false positive rate which converges to 100%. This clearly affects the interpretability and reliability of null-hypothesis tests. Furthermore, even with correct specification of our plug-in estimators, our models are unlikely to be 'targeted' to the desired estimand, because they often estimate quantities superfluous to the estimand but necessary for the plug-in estimator (*e.g.*, other relevant factors or statistics of the joint distribution). As a result, in many cases there exist opportunities to reduce residual bias using what are known as *influence functions*.

## 3.2 Influence Functions

Semiparametric theory and, in particular, the concept of Influence Functions (IFs), are known to be challenging to assimilate (Fisher & Kennedy, 2019; Levy, 2019; Hines et al., 2021). Here we attempt to provide a brief, top-level intuition, but a detailed exposition lies beyond the scope of this paper. Interested readers are encouraged to consider work by Kennedy (2016); Fisher & Kennedy (2019); Hampel (1974); Ichimura & Newey (2021); Hines et al. (2021); Bickel et al. (1998); Newey (1994; 1990); Chernozhukov et al. (2017); van der Laan & Rubin (2006), and Tsiatis, 2006.

An estimator $\Psi(\hat{\mathcal{P}}_n)$ for an estimand $\Psi(\mathcal{P})$ (for example, the ATE) has an IF, $\phi$, if it can be expressed as follows:

$$\sqrt{n}(\Psi(\hat{\mathcal{P}}_n) - \Psi(\mathcal{P})) = \frac{1}{\sqrt{n}} \sum_{i=1}^{n} \phi(z_i, \mathcal{P}) + o_p(1) \tag{4}$$

where $z_i$ is a sample from the true distribution $\mathcal{P}$, $\hat{\mathcal{P}}_n$ is the empirical distribution or, alternatively, a model of some part thereof (*e.g.*, a predictive distribution parameterized by a NN, or a histogram estimate for a density function, etc.), $o_p(1)$ is an error term that converges in probability to zero, and $\phi$ is a function with a mean of zero and finite variance (Tsiatis, 2006, pp.21). The $\sqrt{n}$ scales the difference such that when the difference converges in distribution we can also say that the difference converges at a parametric root-$n$ rate.

Overall, Eq. 4 tells us that the difference between the true quantity and the estimated quantity can be represented as the sum of a bias term and some error term which converges in probability to zero. The IF itself is a function which models how much our estimate deviates from the true estimand, up to the error term. If an estimator can be written in terms of its IF, then by central limit theorem and Slutsky's theorem, the estimator converges in distribution to a normal distribution with mean zero and variance equal to the variance of the IF. This is a key result that enables us to derive confidence intervals and perform statistical inference (such as null hypothesis significance testing). Note that the convergence in distribution to a normal distribution does not impact our assumptions about the functional form governing the data generating process itself. In other words, by deriving a normally distributed parameter estimate, we do not sacrifice flexibility in how we model the observed distribution, or flexibility in the functions used to derive the estimates themselves. The normality relates only to the variation in the parameter estimate itself, across sub-samples from the population.

### 3.2.1   A Simple Example

By way of example, consider the targeted estimand to be the expectation $\mathbb{E}_{y \sim P(Y)}[y]$, where $Y$ is a random variable constituting true distribution $\mathcal{P}$. This can be expressed as:

$$\mathbb{E}_{y \sim \mathcal{P}}[y] = \Psi(\mathcal{P}) = \int y p(y) dy \tag{5}$$

In the case where we have access to an empirical distribution $\hat{\mathcal{P}}_n$, the expectation example may be approximated as follows:

$$\Psi(\mathcal{P}) \approx \Psi(\hat{\mathcal{P}}_n) = \frac{1}{n} \sum_{i=1}^{n} y_i \tag{6}$$

where the subscript $n$ is the sample size. According to Eq. 4, the degree to which our resulting estimator is biased can therefore be expressed as:

$$\begin{aligned}
\sqrt{n}(\Psi(\hat{\mathcal{P}}_n) - \Psi(\mathcal{P})) &= \sqrt{n} \left( \frac{1}{n} \sum_{i=1}^{n} y_i - \int y d\mathcal{P}(y) \right) \\
&= \frac{1}{\sqrt{n}} \sum_{i}^{n} (y_i - \mu) \xrightarrow{\mathcal{D}} \mathcal{N}(0, \sigma^2)
\end{aligned} \tag{7}$$

where $\mu$ and $\sigma^2$ are the mean and variance of $Y$, respectively, and the second line is a consequence of the central limit theorem. This shows that the empirical approximation of the estimand is a *consistent* estimator, insofar as it converges to the true value as the sample size increases.

### 3.2.2   Parametric Submodel and Pathwise Derivative

In many cases $\hat{\mathcal{P}}_n$ is not equivalent to the sample distribution, perhaps because some or all of it is being modelled with estimators. As a result, the error does not converge in probability to zero and some residual

error remains. This situation can be expressed using the IF, as per Eq. 4. Here, the IF $\phi$ is being used to model the residual bias that stems from the fact that $\hat{\mathcal{P}}_n$ is no longer equivalent to a direct sample from $\mathcal{P}$. We will discuss the details relating to this function shortly. If we assume that the difference is asymptotically linear, then we can represent $\hat{\mathcal{P}}_n$ as a perturbed version of $\mathcal{P}$. This also results in convergence in distribution as follows:

$$\frac{1}{\sqrt{n}} \sum_i^N \phi(z_i, \mathcal{P}) \xrightarrow{\mathcal{D}} \mathcal{N}\left(0, \mathbb{E}(\phi\phi^T)\right),$$

$$\sqrt{n}(\Psi(\hat{\mathcal{P}}_n) - \Psi(\mathcal{P})) \xrightarrow{\mathcal{D}} \mathcal{N}\left(0, \mathbb{E}(\phi\phi^T)\right). \tag{8}$$

We can imagine the sample distribution $\hat{\mathcal{P}}_n$ lies on a linear path towards the true distribution $\mathcal{P}$. This linear model can be expressed using what is known as a parametric submodel, which represents a family of distributions indexed by a parameter $\epsilon$:

$$\mathcal{P}_\epsilon = \epsilon\hat{\mathcal{P}}_n + (1-\epsilon)\mathcal{P} \tag{9}$$

It can be seen that when $\epsilon = 0$, we arrive at the true distribution, and when $\epsilon = 1$, we have our current empirical distribution or model. We can therefore use this submodel to represent the perturbation from where we want to be $\mathcal{P}$ in the direction of where we are with our current estimator(s) $\hat{\mathcal{P}}_n$. The direction associated with $\mathcal{P}_\epsilon$ can then be expressed as a pathwise derivative in terms of the function representing our estimand $\Psi$:

$$\frac{d\Psi(\epsilon\hat{\mathcal{P}}_n + (1-\epsilon)\mathcal{P})}{d\epsilon} \tag{10}$$

When this derivative exists (under certain regularity conditions), it is known as the Gateaux derivative. We can evaluate this when $\epsilon = 0$ (*i.e.*, evaluated at the true distribution according to the parametric submodel). Then by the Riesz representation theorem (Frèchet, 1907; Riesz, 1909), we can express the linear functional in Eq. 10, evaluated at $\epsilon = 0$, as an inner product between a functional $\phi$ and its argument:

$$\left.\frac{d\Psi(\epsilon\hat{\mathcal{P}}_n + (1-\epsilon)\mathcal{P})}{d\epsilon}\right|_{\epsilon=0} = \int \phi(y, \mathcal{P})\{d\hat{\mathcal{P}}_n(y) - d\mathcal{P}(y)\} \tag{11}$$

The function $\phi$ is the *Influence Function* (IF) evaluated at the distribution $\mathcal{P}$ in the direction of $y$. Eq. 11 can be substituted back into Eq. 4 to yield:

$$\sqrt{n}(\Psi(\hat{\mathcal{P}}_n(y)) - \Psi(\mathcal{P}(y))) = \int \phi(y, \mathcal{P})\{d\hat{\mathcal{P}}_n(y) - d\mathcal{P}(y)\} + o_p(1) \tag{12}$$

which equivalently allows us to express the estimate of the target quantity as:

$$\Psi(\hat{\mathcal{P}}_n) = \Psi(\mathcal{P}) + \left.\frac{d\Psi(\epsilon\hat{\mathcal{P}}_n + (1-\epsilon)\mathcal{P})}{d\epsilon}\right|_{\epsilon=0} + o_p(1/\sqrt{n}) \tag{13}$$

Eq. 13 expresses the estimated quantity $\Psi(\hat{\mathcal{P}}_n)$ in terms of the true quantity $\Psi(\mathcal{P})$, whereas it would be more useful to do so the other way around, such that we have the true quantity in terms of things we can

estimate. Hines et al. (2021) provide an exposition in terms of the Von Mises Expansion (VME), which is the functional analogue of the Taylor expansion, such that the true quantity can be expressed as:

$$\Psi(\mathcal{P}) = \Psi(\hat{\mathcal{P}}_n) + \frac{1}{n} \sum_i^n \phi(y_i, \hat{\mathcal{P}}_n) + o_p(1/\sqrt{n}) \tag{14}$$

Which, it can be seen, is in the same form as Eq. 13, except that $\phi$ is being evaluated at $\hat{\mathcal{P}}_n$, rather than $\mathcal{P}$. This also accounts for the change in direction otherwise absorbed by a minus sign when expressing $\Psi(\mathcal{P})$ in terms of $\Psi(\hat{\mathcal{P}}_n)$. Finally, note that in Eq. 11 the pathwise derivative expresses the expectation of $\phi$. However, in cases where we substitute $\hat{\mathcal{P}}$ for a Dirac function (see Sec. 3.2.3 for an example), the integral will evaluate to the value of $\phi$ at one specific point. Of course, if we have multiple values we wish to evaluate at (e.g. an empirical distribution represented with Dirac delta functions at each point), then the result is the empirical approximation to the expectation, as indicated by the $\frac{1}{n} \sum_i^n$ notation in Eq. 14.

### 3.2.3 Influence Function for the Average Treatment Effect

A second example (in addition to the expectation given in Sec. 3.2.1) concerns the ATE, which we can break down in terms of an expected difference between two potential outcomes. For the DAG: $T \to Y$, $T \leftarrow X \to Y$ (also see Fig. 2a), the expectation of the potential outcome under treatment can be expressed as (Hines et al., 2021; Hahn, 1998):

$$\Psi(\mathcal{P}) = \mathbb{E}_{\mathbf{x} \sim P(\mathbf{X})}[\mathbb{E}_{y \sim P(Y|T=t, \mathbf{X}=\mathbf{x})}[y]] = \int y f(y|T=1, \mathbf{X}=\mathbf{x}) f(\mathbf{X}=\mathbf{x}) dy d\mathbf{x}$$
$$= \int \frac{y f(y, t, \mathbf{x}) f(\mathbf{x})}{f(t, \mathbf{x})} dy d\mathbf{x}, \tag{15}$$

where $\mathbf{Z} = (\mathbf{X}, T, Y)$. Following the same steps as before, the IF can be derived by first substituting each density

$$f_\epsilon(y, t, \mathbf{x}) = \epsilon \delta_{\tilde{y}, \tilde{t}, \tilde{\mathbf{x}}}(y, t, \mathbf{x}) + (1 - \epsilon) f(y, t, \mathbf{x}), \tag{16}$$

for $f(y, t, \mathbf{x})$, and equivalently for $f(\mathbf{x})$, and $f(t, \mathbf{x})$:

$$\phi(\mathbf{Z}, \mathcal{P}_\epsilon) = \int y \frac{d}{d\epsilon}\bigg|_{\epsilon=0} \frac{y f_\epsilon(y, t, \mathbf{x}) f_\epsilon(\mathbf{x})}{f_\epsilon(t, \mathbf{x})} dy d\mathbf{x}. \tag{17}$$

In a slight abuse of notation, $\delta_{\tilde{y}}$ is the Dirac delta function at the point at which $y = \tilde{y}$, where $\tilde{y}$ can be a datapoint in our empirical sample (note the shift from specific datapoint $y_i$ to generic empirical samples $\tilde{y}$). Then, taking the derivative, and setting $\epsilon = 0$:

$$\phi(\mathbf{Z}, \mathcal{P}) = \int y f(y|t, \mathbf{x}) f(\mathbf{x}) \left[ \frac{\delta_{\tilde{y}, \tilde{t}, \tilde{\mathbf{x}}}(y, t, \mathbf{x})}{f(y, t, \mathbf{x})} + \frac{\delta_{\tilde{\mathbf{x}}}(\mathbf{x})}{f(\mathbf{x})} - \frac{\delta_{\tilde{t}, \tilde{\mathbf{x}}}(t, \mathbf{x})}{f(t, \mathbf{x})} - 1 \right] dy d\mathbf{x}, \tag{18}$$

$$\phi(\mathbf{Z}, \mathcal{P}) = \int \frac{y f(y|t, \mathbf{x}) f(\mathbf{x}) \delta_{\tilde{y}, \tilde{t}, \tilde{\mathbf{x}}}(y, t, \mathbf{x})}{f(y|t, \mathbf{x}) f(t|\mathbf{x}) f(\mathbf{x})} dy d\mathbf{x} + \int \frac{y f(y|t, \mathbf{x}) f(\mathbf{x}) \delta_{\tilde{\mathbf{x}}}(\mathbf{x})}{f(\mathbf{x})} dy d\mathbf{x}$$
$$- \int \frac{y f(y|t, \mathbf{x}) f(\mathbf{x}) \delta_{\tilde{t}, \tilde{\mathbf{x}}}(t, \mathbf{x})}{f(t|\mathbf{x}) f(\mathbf{x})} dy d\mathbf{x} - \int y f(y|t, \mathbf{x}) f(\mathbf{x}) dy d\mathbf{x}, \tag{19}$$

$$\phi(\mathbf{Z}, \mathcal{P}) = \delta_{\tilde{t}}(t) \int \frac{y \delta_{\tilde{y}}(y)}{f(t|\tilde{\mathbf{x}})} dy + \int y f(y|t, \tilde{\mathbf{x}}) dy - \delta(t) \int \frac{y f(y|t, \tilde{\mathbf{x}})}{f(t|\tilde{\mathbf{x}})} dy - \Psi(\mathcal{P})$$
$$= \frac{\delta_{\tilde{t}}(t)}{f(t|\tilde{\mathbf{x}})} \left( \tilde{y} - \mathbb{E}_{y \sim P(Y|T=t, \mathbf{X}=\tilde{\mathbf{x}})}[y] \right) + \mathbb{E}_{y \sim P(Y|T=t, \mathbf{X}=\tilde{\mathbf{x}})}[y] - \Psi(\mathcal{P}), \tag{20}$$

Which yields our IF:

$$\phi(\mathbf{Z}, \mathcal{P}) = \frac{\delta_{\tilde{t}}(t)}{f(t|\tilde{\mathbf{x}})} \left( \tilde{y} - \mathbb{E}_{y \sim P(Y|T=t, \mathbf{X}=\tilde{\mathbf{x}})}[y] \right) + \mathbb{E}_{y \sim P(Y|T=t, \mathbf{X}=\tilde{\mathbf{x}})}[y] - \Psi(\mathcal{P}). \tag{21}$$

Once again, in order to evaluate this we need to evaluate it at $\hat{\mathcal{P}}_n$, and we also need plug-in estimators $\hat{G}(\tilde{\mathbf{x}}) \approx f(t|\tilde{\mathbf{x}})$ (propensity score model), and $\hat{Q}(t, \tilde{\mathbf{x}}) \approx \mathbb{E}_{y \sim P(Y|T=t, \mathbf{X}=\tilde{\mathbf{x}})}[y]$ (outcome model). The propensity score model represents a nuisance parameter and contributes to bias. This finally results in:

$$\phi(\mathbf{Z}, \hat{\mathcal{P}}_n) = \frac{\delta_{\tilde{t}}(t)}{\hat{G}(\tilde{\mathbf{x}})} \left( \tilde{y} - \hat{Q}(t, \tilde{\mathbf{x}})] \right) + \hat{Q}(t, \tilde{\mathbf{x}}) - \Psi(\mathcal{P}). \tag{22}$$

Note that for non-discrete $T$, it may be impossible to evaluate precisely due to the Dirac function. However, and as Hines et al. (2021) and Ichimura & Newey (2021) note, this issue may be circumvented by using a substitute probability measure with a bandwidth parameter which approaches a point mass when the bandwidth parameter is equal to zero.

Equation 22 depicted the influence function for the potential outcome mean, but if we wish to derive the influence function for the average treatment *effect* (*i.e*, the difference between the outcomes from $T = 1$ and $T = 0$) one may note that the last line in Equation 15 can be duplicated and subtracted by setting the value of $T$ to the desired contrast value. The influence functions for each potential outcome can then be derived independently, and the result is equivalent to their direct combination (van der Laan & Rose, 2011):

$$\phi_{ATE}(\mathbf{Z}, \hat{\mathcal{P}}_n) = \left( \frac{\delta_{\tilde{t}}(1)}{\hat{G}(\tilde{\mathbf{x}})} - \frac{1 - \delta_{\tilde{t}}(0)}{1 - \hat{G}(\tilde{\mathbf{x}})} \right) \left( \tilde{y} - \hat{Q}(t, \tilde{\mathbf{x}})] \right) + \hat{Q}(1, \tilde{\mathbf{x}}) - \hat{Q}(0, \tilde{\mathbf{x}}) - \Psi_{ATE}(\mathcal{P}). \tag{23}$$

An alternative approach to the derivation of influence functions exists, and involves the use of the derivative of the log-likelihood (the score) (Levy, 2019). The approach presented here is arguably more straightforward and follows the presentation by Ichimura & Newey (2021); Hines et al. (2021), although it depends on pathwise differentiability of the estimand.

### 3.2.4 Statistical Inference with Influence Functions

As described earlier, machine learning techniques, while parameterized *per se*, are non-parametric insofar as the estimates they yield are not directly parameterizable as (*e.g.*) a Gaussian with a mean and a variance. However, such parameterization is extremely helpful in facilitating statistical inference, including the ubiquitous null hypothesis significance test, which is straightforward when the parameter being tested is normally distributed. Whilst other approaches to statistical inference with non-parametric methods exist, such as the bootstrap approach (Efron & Gong, 1983), influence functions have been recognized as a valid approach for some time (Bickel et al., 1998; Tsiatis, 2006).

Following van der Laan & Rose (2011, p.75) we can derive 95% confidence intervals from the influence function to be (assuming normal distribution):

$$\widehat{\text{Var}}(\phi) = \frac{1}{n} \sum_i^n \left[ \phi(\mathbf{z}_i) - \frac{1}{n} \sum_j^n \phi(\mathbf{z}_j) \right]^2,$$

$$\widehat{\text{se}} = \sqrt{\frac{\widehat{\text{Var}}(\phi)}{n}}, \tag{24}$$

$$\Psi^*(\hat{\mathcal{P}}_n) \pm 1.96\widehat{\text{se}},$$

$$p_{val} = 2\left[1 - \Phi\left(\left|\frac{\Psi^*(\hat{\mathcal{P}}_n)}{\widehat{\text{se}}}\right|\right)\right],$$

where $\Psi^*(\hat{\mathcal{P}}_n)$ is the estimated target quantity after bias correction has been applied, $\Phi$ is the CDF of a normal distribution, $\widehat{\text{se}}$ is the standard error, and $p_{val}$ is the $p$-value.

### 3.3 IFs for General Graphical Models

In this paper, we focus on the estimation of average treatment effect in the setting of Fig 2a. However, the methods discussed in this paper can be applied for more complex estimands with an arbitrary causal graph structure, as long as the estimand at hand is causally *identifiable* from the observed data. In this section, we discuss the derivation of IFs for a general form of an estimand in a general graphical model.

#### 3.3.1 Influence Function of an Interventional Distribution

The causal identification of interventional distributions is well-studied in the literature. In the case of full observability, any interventional distribution is identifiable using (extended) g-formula (Ezzati et al., 2004; Robins, 1986). If some variables of the causal system are unobserved, all interventional distributions are not necessarily identifiable. Tian & Pearl (2002) and Shpitser & Pearl (2006) provided necessary and sufficient conditions of identifiability in such models. The causal identification problem in DAGs with unobserved (latent) variables can equivalently be defined on *acyclic directed mixed graphs* (ADMGs) (Richardson & Spirtes, 2003; Richardson et al., 2017; Evans & Richardson, 2019). ADMGs are acyclic mixed graphs with directed and bidirected edges, that result from a DAG through a latent projection operation onto a graph over the observable variables (Verma & Pearl, 1990).

Pearl's do-calculus is shown to be complete for the identification of interventional distributions (Huang & Valtorta, 2006). Let $\mathbf{V}$ denote the set of all observed variables. Starting with an identifiable interventional distribution $P(\mathbf{y}|do(\mathbf{T} = \mathbf{t}'))$, an identification functional of the following form is derived using do-calculus:

$$\mathcal{P}(\mathbf{y}|do(\mathbf{T} = \mathbf{t}')) = \oint_{\mathbf{S}} \frac{\Pi_i \mathcal{P}(\mathbf{a_i}|\mathbf{b_i})}{\Pi_j \mathcal{P}(\mathbf{c_j}|\mathbf{d_j})}, \tag{25}$$

where $\mathbf{a_i}, \mathbf{b_i}, \mathbf{c_j}$, and $\mathbf{d_j}$ are realizations of $\mathbf{A_i}, \mathbf{B_i}, \mathbf{C_j}$, and $\mathbf{D_j}$, respectively, and $\mathbf{A_i}, \mathbf{B_i}, \mathbf{C_j}, \mathbf{D_j}, \mathbf{S}$ are subsets of variables such that for each $i$ and $j$, $\mathbf{A_i} \cap \mathbf{B_i} = \varnothing$ and $\mathbf{C_j} \cap \mathbf{D_j} = \varnothing$. Note that the sets $\mathbf{B_i}$ and $\mathbf{D_j}$ might be empty. The $\oint$ symbol in Eq. 25 indicates a summation over the values of the set of variables $\mathbf{S}$ in the discrete case, and an integration over these values in the continuous setting. To derive the influence function of Eq. 25, we begin with a conditional distribution of the form $\mathcal{P}(\mathbf{a}|\mathbf{b})$. If $\mathbf{b} \neq \varnothing$, we can write

$$\mathcal{P}_\epsilon(\mathbf{v}) = (1-\epsilon)\mathcal{P}(\mathbf{v}) + \epsilon\delta_{\tilde{\mathbf{v}}}(\cdot),$$
$$\mathcal{P}_\epsilon(\mathbf{a}|\mathbf{b}) = \frac{\mathcal{P}_\epsilon(\mathbf{a},\mathbf{b})}{\mathcal{P}_\epsilon(\mathbf{b})},$$
$$\frac{d\mathcal{P}_\epsilon(\mathbf{a}|\mathbf{b})}{d\epsilon}\bigg|_{\epsilon=0} = \frac{\delta_{\tilde{\mathbf{a}},\tilde{\mathbf{b}}}(\mathbf{a},\mathbf{b}) - \mathcal{P}(\mathbf{a},\mathbf{b})}{\mathcal{P}(\mathbf{b})} - \frac{\mathcal{P}(\mathbf{a},\mathbf{b})[\delta_{\tilde{\mathbf{b}}}(\mathbf{b}) - \mathcal{P}(\mathbf{b})]}{\mathcal{P}^2(\mathbf{b})} \tag{26}$$
$$= \mathcal{P}(\mathbf{a}|\mathbf{b}) \cdot \left(\frac{\delta_{\tilde{\mathbf{a}},\tilde{\mathbf{b}}}(\mathbf{a},\mathbf{b})}{\mathcal{P}(\mathbf{a},\mathbf{b})} - \frac{\delta_{\tilde{\mathbf{b}}}(\mathbf{b})}{\mathcal{P}(\mathbf{b})}\right),$$

where $\tilde{\mathbf{v}}$ is the point that we compute the influence function at, and $\tilde{\mathbf{a}}, \tilde{\mathbf{b}}$ are the values of sets of variables $\mathbf{A}, \mathbf{B} \subseteq \mathbf{V}$ that are consistent with $\tilde{\mathbf{v}}$. For an empty $\mathbf{b}$, using similar arguments, we have:

$$\frac{d\mathcal{P}_\epsilon(\mathbf{a})}{d\epsilon}\bigg|_{\epsilon=0} = \mathcal{P}(\mathbf{a}) \cdot \left(\frac{\delta_{\tilde{\mathbf{a}}}(\mathbf{a})}{\mathcal{P}(\mathbf{a})} - 1\right). \tag{27}$$

With slight abuse of notation, for $\mathbf{b} = \varnothing$, we define $\frac{\delta_{\tilde{\mathbf{b}}}(\mathbf{b})}{\mathcal{P}(\mathbf{b})} = 1$. Using Eq. 26 and Eq. 27, we can now derive the IF of Eq. 25.

$$\phi(\tilde{\mathbf{v}}, \mathcal{P}) = \frac{d((1-\epsilon)\mathcal{P} + \epsilon\delta_{\tilde{\mathbf{v}}})}{d\epsilon}\bigg|_{\epsilon=0} =$$
$$\sum_{\mathbf{S}} \frac{\Pi_i \mathcal{P}(\mathbf{a_i}|\mathbf{b_i})}{\Pi_j \mathcal{P}(\mathbf{c_j}|\mathbf{d_j})} \cdot \left[\sum_i \left(\frac{\delta_{\tilde{\mathbf{a_i}},\tilde{\mathbf{b_i}}}(\mathbf{a_i},\mathbf{b_i})}{\mathcal{P}(\mathbf{a_i},\mathbf{b_i})} - \frac{\delta_{\tilde{\mathbf{b_i}}}(\mathbf{b_i})}{\mathcal{P}(\mathbf{b_i})}\right) - \sum_j \left(\frac{\delta_{\tilde{\mathbf{c_j}},\tilde{\mathbf{d_j}}}(\mathbf{c_j},\mathbf{d_j})}{\mathcal{P}(\mathbf{c_j},\mathbf{d_j})} - \frac{\delta_{\tilde{\mathbf{d_j}}}(\mathbf{d_j})}{\mathcal{P}(\mathbf{d_j})}\right)\right]. \tag{28}$$

Note that we used $\frac{d}{d\epsilon}\frac{1}{\mathcal{P}_\epsilon(\mathbf{c}|\mathbf{d})} = -\frac{\frac{d}{d\epsilon}\mathcal{P}_\epsilon(\mathbf{c}|\mathbf{d})}{\mathcal{P}_\epsilon^2(\mathbf{c}|\mathbf{d})}$. Note also that Equation 18, which is the influence function for the potential outcome mean, is of the same form as Equation 28. Equation 28 is the foundation to the approach that shall be discussed in the following section for deriving the IF of a general class of estimands.

### 3.4 Influence Function of a General Estimand

We have so far discussed the influence function of a causal effect of the form $\mathcal{P}(\mathbf{y}|do(\mathbf{T} = \mathbf{t}'))$. In this section, we show how IFs can be derived for any general estimand of the form:

$$\Psi(\mathcal{P}) = \mathbb{E}_\mathcal{P}[\kappa(\mathcal{P})], \tag{29}$$

where $\kappa(\cdot)$ is a functional. Then we have:

$$\mathcal{P}_\epsilon = \epsilon\hat{\mathcal{P}}_n + (1-\epsilon)\mathcal{P},$$
$$\Psi(\mathcal{P}_\epsilon) = \int \kappa(\mathcal{P}_\epsilon)\mathcal{P}_\epsilon d\mathbf{v},$$
$$\frac{d\Psi(\mathcal{P}_\epsilon)}{d\epsilon}\bigg|_{\epsilon=0} = \int \left(\frac{d\mathcal{P}_\epsilon}{d\epsilon} \cdot \kappa(\mathcal{P}_\epsilon) + \frac{d\kappa}{d\mathcal{P}_\epsilon} \cdot \frac{d\mathcal{P}_\epsilon}{d\epsilon} \cdot \mathcal{P}_\epsilon\right)\bigg|_{\epsilon=0} d\mathbf{v}$$
$$= \int \left(\kappa(\mathcal{P}) + \frac{d\kappa}{d\mathcal{P}} \cdot \mathcal{P}\right) \cdot \frac{d\mathcal{P}_\epsilon}{d\epsilon}\bigg|_{\epsilon=0} d\mathbf{v} \tag{30}$$
$$= \int \kappa(\mathcal{P}) \cdot \frac{d\mathcal{P}_\epsilon}{d\epsilon}\bigg|_{\epsilon=0} d\mathbf{v} + \mathbb{E}_\mathcal{P}\left[\frac{d\kappa}{d\mathcal{P}} \cdot \frac{d\mathcal{P}_\epsilon}{d\epsilon}\bigg|_{\epsilon=0}\right].$$

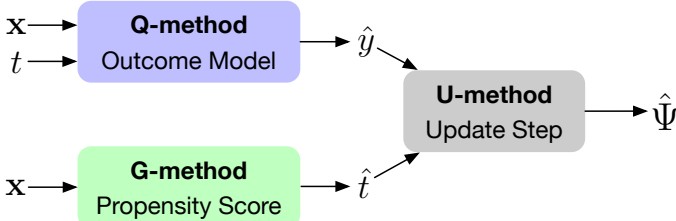

Figure 3: This figure illustrates the components involved in using IFs to improve our estimates of the Average Treatment Effect (ATE), where the ATE is our target estimand $\Psi$. We combine the output from an outcome model $\hat{Q}$, with a propensity score model $\hat{G}$ and an update step method U. This yields an estimate $\hat{\Psi}$.

The value of $\frac{d\mathcal{P}_\epsilon}{d\epsilon}\big|_{\epsilon=0}$ can be plugged into Eq. 30 using Eq. 28 and Eq. 11, which completes the derivation of the IF for the estimand in Eq. 29. As an example, if the queried estimand is the average density of a variable $Y$, that is, $\kappa$ is the identity functional, then:

$$\Psi(\mathcal{P}) = \int \mathcal{P}^2(y) dy,$$

$$\frac{d\Psi(\mathcal{P}_\epsilon)}{d\epsilon}\bigg|_{\epsilon=0} = \int (\mathcal{P} + 1 \cdot \mathcal{P}) \cdot \frac{d\mathcal{P}_\epsilon}{d\epsilon}\bigg|_{\epsilon=0} dy$$

$$= \int 2\mathcal{P}(y) \cdot \frac{d\mathcal{P}_\epsilon}{d\epsilon}\bigg|_{\epsilon=0} dy.$$

Algorithm 1 summarises the steps of our proposed automated approach to derive the influence function of an estimand of the form presented in Eq. 29, given a general graphical model. Note that if the effect is identifiable, this algorithm outputs the analytic influence function, and otherwise, throws a failure. A demonstrative example can be found in the associated code repository in the form of a notebook, and/or in the attached supplementary code.

---

**Algorithm 1** IF of an identifiable effect.

---

  **input:** An estimand $\Psi(\mathcal{P})$ of the form of Eq. 29, an interventional distribution $\mathcal{P}$, causal graph $\mathcal{G}$
  **output:** The analytic IF of $\Psi(\mathcal{P})$ if $\mathcal{P}$ is identifiable, fail o.w.
1: **if** $\mathcal{P}$ is identifiable **then**
2:     $\tilde{\mathcal{P}} \leftarrow$ the identification functional of $\mathcal{P}$ (Eq. 25) using do-calculus
3:     $\phi \leftarrow$ the IF of $\mathcal{P}$ as in Eq. 28
4:     $\frac{d\Psi(\mathcal{P}_\epsilon)}{d\epsilon}\big|_{\epsilon=0} \leftarrow$ the formulation as in Eq. 30
5:     $\Phi \leftarrow$ Plug $\phi$ into $\frac{d\Psi(\mathcal{P}_\epsilon)}{d\epsilon}\big|_{\epsilon=0}$ using Eq. 11
6:     **return** $\Phi$
7: **else**
8:     **return FAIL**

---

## 4  Updating/Debiasing our Estimators with IFs

If we can estimate the IF $\phi$ then we can update our initial estimator $\Psi(\hat{\mathcal{P}}_n)$ according to Eq. 14 in order to reduce the residual bias which the IF is essentially modeling. To be clear, this means we can improve our initial NN estimators, without needing additional data. We consider four ways to leverage the IF to reduce bias which we refer to as (1) the one-step update, (2) the submodel update (sometimes referred to as a targeted update), (3) our own proposed MultiStep procedure, and (4) targeted regularization. The first three approaches can be trivially applied to estimators which have already been trained, making them attractive as post-processing methods for improving estimation across different application areas. To illustrate these approaches, we consider the ATE to be our chosen target estimand, the IF for which is defined in Equation 23.

Figure 3 provides an illustration of the components involved in updating the estimator in the context of ATE estimation.

## 4.1 One-Step and Submodel Approach

Using the *one-step* approach, the original estimator $\Psi(\hat{\mathcal{P}}_n)$ can be improved by a straightforward application of the Von Mises Expansion (VME) of Eq. 14 - one takes the initial estimator and adds to it the estimate of the IF to yield an updated estimator which accounts for the 'plug-in bias'. In the case of the ATE, this yields the augmented inverse propensity weighted (AIPW) estimator (Hines et al., 2021; Neugebauer & van der Laan, 2005; Kurz, 2021).

The second *submodel* approach updates the initial estimate by solving $\sum_{i=1}^{n} \phi(\mathbf{z}_i, \hat{\mathcal{P}}_n) = 0$, which can be done by estimating the degree to which the principal estimator $Q$ is being biased, and correcting for it. This approach works by first constructing a parametric submodel in terms of the plug in estimator $Q(t, \mathbf{X})$ and a function $H$ of the propensity score $G$ (which represents the biasing quantity), and derives an updated plug-in estimator $Q^*(t, \mathbf{x})$.

In the expressions which follow, we assume binary treatment $T$ and have replaced the Dirac delta functions with indicator functions. First, the updated estimator $Q^*(t, \mathbf{x})$ can be expressed in terms of a biasing quantity $H$ as follows:

$$\hat{Q}^*(T = 1, \mathbf{x}_i) = \hat{Q}(T = 1, \mathbf{x}_i) + \hat{\gamma} H(\mathbf{z}_i, T = 1),$$
$$\text{where } H(\mathbf{z}_i, T = 1) = \frac{\mathbb{1}_{t_i}(1)}{\hat{G}(\tilde{\mathbf{x}})}.$$

The equivalent for when treatment is 0 can be expressed as: $\qquad$ (31)

$$\hat{Q}^*(T = 0, \mathbf{x}_i) = \hat{Q}(T = 0, \mathbf{x}_i) + \hat{\gamma} H(\mathbf{z}_i, T = 0),$$
$$\text{where } H(\mathbf{z}_i, T = 0) = -\frac{1 - \mathbb{1}_{t_i}(0)}{1 - \hat{G}(\tilde{\mathbf{x}})}.$$

$H(\mathbf{z}_i, t_i)$ is known as the clever covariate. The parameter $\hat{\gamma}$ is estimated as the coefficient in the associated intercept-free 'maximum-likelihood linear regression' reflected in the first line of Eq. 31 above. When the updated $\hat{Q}^*$ is substituted into Eq.23, one solves in one update what is known as the *efficient influence function*, and following the update, the mean of the IF, as well as the residual bias will be zero. In practice, the two methods yield different results with finite samples (Porter et al., 2011; Benkeser et al., 2017). In particular, the one-step / AIPW estimator may yield estimates outside of the range of values allowed according to the parameter space, and be more sensitive to near-positivity violations (*i.e.*, when the probability of treatment is close to zero) owing to the first term on the RHS of Eq. 23 (Luque-Fernandez et al., 2018). In contrast, the submodel approach will not, because it is constrained by the intercept-free regression step - for instance, if the outcome $Y$ is binary, the logistic link function will prevent any 'out-of-bounds' convergence behavior which might otherwise result in an unstable estimate for the coefficient $\hat{\gamma}$. Thus, both the one-step and submodel approach achieve the same aim: solving for the efficient influence function. However, the one-step approach does so naively according to the VME, and the second does so according to an additional regression stage (but nonetheless still only requires a single update to the original outcome model $Q$).

**Model Robustness:** One of the consequences of finding the efficient IF is that we also achieve improved model robustness. This is because, in cases where multiple plug-in models are used to derive an unbiased estimate, we achieve consistent estimation (*i.e.*, we converge in probability to the true parameter as the sample size increases) even if one of the models is misspecified (*e.g.*, the ATE requires both a propensity score model and an outcome model, and thus the IF facilitates *double* robustness). Furthermore, in cases where both models are well-specified, we achieve efficient estimation. It is worth noting, however, that this double-robustness property does not apply to the limiting distribution of the estimates being Gaussian when data-adaptive plug-in estimators are used (Benkeser et al., 2017; van der Laan, 2014). In other words, if only one or both of the two models is/are incorrectly specified, the estimates may not be normally distributed, thus

invalidating statistical inference. In our later evaluation, we thus might expect models to fail at achieving *normally distributed* estimates before they fail at yielding *unbiased* estimates. It is possible to extend the framework such that the double robustness property also applies to the limiting normal distribution of the estimates (Benkeser et al., 2017; van der Laan, 2014), but we leave this to future work. For more technical details on the double robustness property see van der Laan & Rose (2011); Hines et al. (2021); Benkeser et al. (2017), and Kurz (2021).

## 4.2 MultiStep Approach

In this section we present our own variant of the estimator update process which we call MultiStep updates. In order to motivate the development of these methods, we begin by noting the limitations of the one-step and submodel update processes. In general, these updates are performed only once (Hines et al., 2021; van der Laan & Rose, 2011), and as described in Section 4.4, the efficacy of these update steps rests on the assumption that we are 'good enough' to begin with. In other words, the bias of our initial estimator must be able to be approximated by a linear submodel, such that taking a step in the direction of the gradient takes us in the right direction. We attempt to improve the empirical robustness of the one-step and submodel update steps by modifying the objective in the update step itself.

Under the assumptions described above, the one-step and the submodel update approaches yield the efficient influence function. That is, $\sum_i^n \phi(\mathbf{z}_i, \hat{\mathcal{P}}_n) \approx 0$. Furthermore, this influence function is also the one with the smallest variance (Tsiatis, 2006). Indirectly, the submodel process achieves this by finding the least-squares (or maximum-likelihood) solution to Eq. 31, updating the initial estimator $\hat{Q}(t, \mathbf{x}_i)$ with some quantity $\hat{\gamma}$ of clever covariate $H(\mathbf{z}_i)$. We refer to this process as 'indirect' because the objective used to find $\hat{\gamma}$ can, alternatively, be specified explicitly.

We refer to our update variant as MultiStep because whilst it still uses the linear submodel of Eq. 31, we optimize the expression 32 below by searching over $\hat{\gamma} \in \Gamma$:

$$\min_{\hat{\gamma} \in \Gamma} \left[ \alpha_1 [\hat{\mathbb{E}}[\phi(\mathbf{z}_i, \hat{\mathcal{P}})]] + \alpha_2 [\widehat{\text{Var}}[\phi(\mathbf{z}_i, \hat{\mathcal{P}})]] \right]. \tag{32}$$

In words, rather than implicitly finding the solution to the IF via maximum-likelihood, we explicitly specify that the solution should minimize empirical approximations (circumflex/hat notation) of both the expectation and/or the variance of the influence function. The degree to which each of the constraints are enforced depends on hyperparameters $\alpha_1 \in \mathcal{R}^+$ and $\alpha_2 \in \mathcal{R}^+$ which weight the two constraints. In this objective, $\hat{\gamma}$ is related to the influence function by:

$$\phi_{ATE}(\mathbf{z}_i, \hat{\mathcal{P}}_n) = H(\mathbf{z}_i, t_i) \left( y_i - \hat{Q}(t_i, \mathbf{x}_i) - \hat{\gamma} H(\mathbf{z}_i) \right)$$
$$+ (\hat{Q}(1, \mathbf{x}_i) + \hat{\gamma} H(\mathbf{z}_i, 1)) - (\hat{Q}(0, \mathbf{x}_i) + \hat{\gamma} H(\mathbf{z}_i, 0)) - \Psi_{ATE}(\hat{\mathcal{P}}_n). \tag{33}$$

In other words, $\hat{\gamma}$ is the coefficient on the clever covariate which itself represents the quantity which is biasing our principal estimator $Q$. The objective in expression 32 therefore finds the $\hat{\gamma}$ which corrects for the biasing action of $H$ by minimizing the mean and the variance of the influence function, which itself is computed according to Eq. 33. Finally, note also the link tying the influence function, $\hat{\gamma}$, and $H$ to the target parameter:

$$\Psi_{ATE}(\hat{\mathcal{P}}_n) = \frac{1}{n} \sum_i^n \left( (\hat{Q}(1, \mathbf{x}_i) + \hat{\gamma} H(\mathbf{z}_i, 1)) - (\hat{Q}(0, \mathbf{x}_i) + \hat{\gamma} H(\mathbf{z}_i, 0)) \right). \tag{34}$$

## 4.3 Targeted Regularization

Finally, we can use targeted regularization which, to the best of our knowledge, has only been used twice in the NN literature, once in DragonNet (Shi et al., 2019), and once in TVAE (Vowels et al., 2021), both of

which were applied to the task of causal inference. The idea is to solve the efficient influence curve *during* NN training, similarly to Eq. 31, on a per-batch basis. The parameter $\hat{\gamma}$ in Eq. 31 is treated as a learnable parameter, trained as part of the optimization of the NN. The submodel update in Eq. 31 is thereby recast as a regularizer which influences the weights and biases of the outcome model $\hat{Q}(t, \mathbf{x})$. In total, then, the training objective is given by Eq. 35, where $\mathcal{L}_i^q$ is a negative log-likelihood (NLL) of the outcome model $\hat{Q}(t, \mathbf{x})$ which has parameters $\theta$ (which comprises NN weights and biases), and $\mathcal{L}_i^{tl}$ is the NLL of the updated outcome model $\hat{Q}^*(t, \tilde{\mathbf{x}})$, which is parameterized by both $\theta$ and $\hat{\gamma}$.

$$\mathcal{L} = \min_\theta \left[ \sum_i^n \left( \mathcal{L}_i^q + \mathcal{L}_i^{tl} \right) \right]. \tag{35}$$

As the second NLL term involves the clever covariate $H$, which in turn involves the plug-in estimator for the propensity score $G(Z)$, we also need a model for the treatment which may be trained via another NLL objective, or integrated into the same NN as the one for the outcome model. Due to the paucity of theoretical analysis for NNs, it is not clear whether targeted regularization provides similar guarantees (debiasing, double-robustness, asymptotic normality) to the one-step and submodel approaches, and this is something we explore empirically.

### 4.4 Conditions for IF Updates to Work

The conditions necessary for the key relationships above to hold are that our estimator is regular and asymptotically linear such that the second order remainder term $o_p(\cdot)$ tends in probability to zero sufficiently quickly. These properties concern the sample size, the smoothness of the estimator, and the quality of the models we are using to approximate the relevant factors of the distribution. Clearly, if our initial model(s) is(are) poor/misspecified then a linear path (or equivalently, a first order VME) will not be sufficient to model the residual distance from the estimand, and the update steps may actually worsen our initial estimate.

In summary, as long as our initial estimator is 'good enough' (insofar as it is regular and asymptotically linear), we can describe any residual bias using IFs. Doing so enables us to (a) reduce the residual bias by performing an update to our original estimator using the efficient IF (via the one-step, submodel, or targeted learning approaches), (b) achieve a more robust estimator, and (c) undertake statistical inference (because the updated estimate is normally distributed with a variance equal to the variance of the IF). Unfortunately, we are not currently aware of a way to assess 'good enough'-ness, particularly in the causal-inference setting, where explicit supervision is not available. There may exist a way to use the magnitude of the IF to assess the validity of the assumption of asymptotic normality, and use this as a proxy for model performance, but we leave this to future work.

## 5 MultiNet

One of the primary considerations when choosing estimation algorithms/models is whether the estimator can represent a family of distributions which is likely to contain the true Data Generating Process (DGP). Indeed, one of the motivations for semiparametrics is to be able to use non-parametric data-driven algorithms which have the flexibility to model complex DGPs, whilst still being able to perform statistical inference. If the estimator is functionally misspecified (*i.e.*, misspecified in terms of the functional form used to model the relationships in the DGP - linear, quadratic, spline etc.), then we are unlikely to arrive at an estimator which is asymptotically linear and therefore also amenable to the Influence Function update process. This behooves us to seek estimators which 'let the data speak' (van der Laan & Starmans, 2014).

Consider the Super Learner (SL) (van der Laan et al., 2007), which is an ensemble method especially designed for parameter estimation in the context of causal inference. The SL process involves taking a weighted average of predictions from each candidate learner, and this quantity is taken as the output. The advantage of a SL is that the candidate library includes sufficient diversity with respect to functional form and complexity such that the true DGP is likely to fall within the family of statistical models which can be represented by the ensemble. The motivation for reducing bias resulting from *functional* misspecification is therefore similar

to the motivation for influence functions. In both cases, accuracy/precision in estimation is the priority in the domain of causal inference, where the parameters concern critical decision making processes, such as the efficacy of medications.

In contrast with the SL, many methods (including boosted trees, neural networks, or linear regressions) are based on single learners with one outcome prediction. As part of the development and evaluation of Influence Function updating methods for reducing bias and deriving consistent estimators, here we also present a new neural-network based estimator / learning algorithm. Early experimentation highlighted to us that even though NNs are flexible universal function approximators (Hornik, 1993; Hornik et al., 1989), they may nonetheless yield estimators which are not 'good enough' to enable us to leverage their asymptotic properties (such as bias reduction with IFs). In such cases, the IF update may actually *worsen* the initial estimate, pushing us further off course. This problem arose even for simple datasets with only quadratic features. Indeed, the problem with using neural networks for 'tabular' data (as opposed to, say image data) is well known in the machine learning community, and interested readers are directed towards the survey by Kadra et al. (2021). Researchers have, in general, noted that gradient boosted trees (Freund & Schapire, 1997) to consistently outperform neural network based learners (Shwartz-Ziv & Armon, 2021; Kadra et al., 2021; Borisov et al., 2022). However, Borisov et al. (2022) also found that ensembles of boosted trees and neural networks can nonetheless outperform boosted trees alone, and Kadra et al. (2021) found that sufficiently regularized neural networks could yield competitive performance, or even exceed the performance of boosted trees. Thus, in our view the avenues for research into neural network methods for tabular data are still open (and research on the subject continues regardless - see TVAE, CFR-net, and DragonNet, for example). Furthermore, if neural network based methods work well in ensemble combinations with boosted trees, we should attempt to maximise the performance of the neural network learners in order to maximise the performance of the associated ensemble.

A block diagram for MultiNet is shown in Figure 1. The method comprises two main elements: a Counter-Factual Regression (CFR) network backbone (Shalit et al., 2017) (without the integral probability metric penalty) with outcome 'taps' at each layer, and a constrained regression procedure designed to reflect the equivalent idea in the Super Learner. The idea therefore represents a combination of two well-known methods in causal inference - CFR-net, and Super Learner. CFR is a popular NN method for causal inference tasks. It includes separate outcome arms depending on the treatment condition, and forms the backbone of MultiNet. For each layer in MultiNet, we predict $y|t, \mathbf{x}$ for $t = \{0, 1\}$ and compute the corresponding layerwise cross-entropy loss (for a binary outcome). This simulates the multiple outputs of a typical ensemble method - each layer represents a different (and increasingly complex) function of the input. The constrained regression is only applied after MultiNet has been trained. For each treatment condition, we concatenate the layerwise predictions into a matrix $\hat{\mathbf{Y}}$ which has shape $(L \times N)$ where $L$ is the number of layers and $N$ is the number of datapoints. We then solve $\hat{\mathbf{Y}}^T \boldsymbol{\beta} = y$, with layerwise weights $\boldsymbol{\beta}$ which are constrained to sum to one and be non-negative. For this we use a SciPy (Jones et al., 2001) non-negative least squares solver. The weights are then used for subsequent predictions. Note that one of the strengths of this approach is that the layerwise outputs and constrained regression techniques can be flexibly applied to other neural network architectures. We may also interpret $\boldsymbol{\beta}$ to understand which layers are the most useful for solving the constrained regression, but leave this to future work.

In order to explore the possibility for increased performance, we explore some additional variations on the core idea. Firstly, we allow each layerwise loss gradient to influence all prior network parameters. This is similar to the implementation of the auxiliary loss idea in the Inception network (Szegedy et al., 2015), and we refer to this variant as 'MN-Inc'. The second variant involves only updating the parameters of the corresponding layer, preventing gradients from updating earlier layers. We call this variant the 'cascade' approach, and refer to this variant as 'MN-Casc'.

Finally, in order to increase the diversity across the layers and to approximate the diversity of an ensemble, we also explore the use of loss masking. For this, we partition the training data such that each layer has a different 'view' of the observations. The loss is masked such that each layer is trained on a different, disjoint subset of the data. We refer to variants of MultiNet with loss masking as 'MN+LM'. The objective function of MultiNet is therefore:

$$\mathcal{L} = \min \left[ \frac{1}{n} \sum_i^n \frac{1}{L} \sum_l^L m_i^l \mathcal{L}_i^l \right], \tag{36}$$

where $m_i^l$ is the mask for datapoint $i$ in layer $l$ (this is set to 1 for variants without loss masking), and $\mathcal{L}_i^l$ is the cross-entropy loss for datapoint $i$ and layer $l$.

## 6 Experimental Setup

### 6.1 Open Questions

So far, we have presented the relevant background for causal inference and IFs, presented a way to derive the IF for a general graph (and, indeed, a general estimand), proposed a new MultiStep update process and proposed a new NN based estimator called MultiNet. A top level illustration is shown in Fig. 3. The following open questions remain: (1) Can estimation methods be improved using the one-step, submodel, MultiStep (ours), or targeted regularization approaches? (2) How do various different outcome, propensity score, and update step methods compare? We aim to answer these questions through an extensive evaluation of different methods (Sec. 7). In particular, we examine the performance of the different approaches in terms of (a) precision in estimation, (b) robustness, and (c) statistical inference (normality of the distribution of estimates). We use these open questions to inform the design of our experiments, which are described below.

### 6.2 Data

Recent work has highlighted the potential for the performance of modern causal inference methods to be heavily dataset-dependent, and has recommended the use of bespoke datasets which transparently test specific attributes of the evaluated models across different dimensions (Curth et al., 2021b). We therefore undertake most of the evaluation using variants of a DGP which we refer to as the LF-dataset and which has been used for similar evaluations in the literature (Luque-Fernandez et al., 2018). We also evaluate using the well-known IHDP dataset (Hill, 2011; Dorie, 2016).

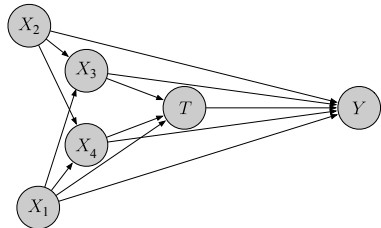

Figure 4: Graph for the 'LF' dataset used by Luque-Fernandez et al. (2018).

### 6.2.1 LF Dataset Variants

The initial and original LF-dataset variant, (v1), models 1-year mortality risk for cancer patients treated with monotherapy or dual therapy. One motivation for starting with this DGP is that its polynomial functional form is not sufficiently complex to unfavourably bias the performance of any method from the start. The dataset also exhibits near-positivity violations, and will therefore highlight problems associated with the propensity score models which are necessary for the update process. We also adjust the level of non-linearity in order to assess the robustness of each method to increased complexity. Accordingly, we introduce an exponential response into the potential outcome under monotherapy ($t = 1$) for the second variant (v2). Our LF-datasets comprise 100 samples from a set of generating equations. Both variants are designed to highlight problems which may arise due to near positivity violations.

The graph for the synthetic 'LF' dataset used in work by Luque-Fernandez et al. (2018) is given in Fig. 4. The DGP is based on a model for cancer patient outcomes for patients treated with monotherapy ($t = 1$) and dual therapy ($t = 0$) and the generating equations are as follows:

$$
\begin{aligned}
& X_1 \sim Be(0.5), \quad X_2 \sim Be(0.65), \\
& X_3 \sim \text{int}[U(0,4)], \quad X_4 \sim \text{int}[U(0,5)], \\
& T \sim Be(p_T), \quad \text{where} \\
& p_T = \sigma(-5 + 0.05X_2 + 0.25X_3 + 0.6X_4 + 0.4X_2X_4), \\
& Y_1 = \sigma(-1 + 1 - 0.1X_1 + 0.35X_2 + 0.25X_3 + 0.2X_4 + 0.15X_2X_4), \\
& Y_0 = \sigma(-1 + 0 - 0.1X_1 + 0.35X_2 + 0.25X_3 + 0.2X_4 + 0.15X_2X_4),
\end{aligned}
\tag{37}
$$

where int[.] is an operator which rounds the sample to the nearest integer, $Be$ is a Bernoulli distribution, $U$ is a uniform distribution, $\sigma$ is the sigmoid function, and $Y_1$ and $Y_0$ are the counterfactual outcomes when $T = 1$ and $T = 0$, respectively. Covariate $X_1$ represents biological sex, $X_2$ represents age category, $X_3$ represents cancer stage, and $X_4$ represents comorbidities.

We create a variant (v2) of this DGP by introducing non-linearity into the outcome, and then into the treatment assignment as follows:

$$
Y_1 = \sigma(exp[-1 + 1 - 0.1X_1 + 0.35X_2 + 0.25X_3 + 0.2X_4 + 0.15X_2X_4]).
\tag{38}
$$

The two variants are designed to yield near positivity violations in order to highlight weaknesses in methods which depend on a reliable propensity score model. Figs. 5 and 6 provide information on the propensity scores for the v1 and v2 variants (the second version has the same propensity score generating model as v1). Finally, for LF (v1) and LF (v2) we create further variants with different sample sizes $n = \{500, 5000, 10000\}$ in order to explore sensitivity to finite samples.

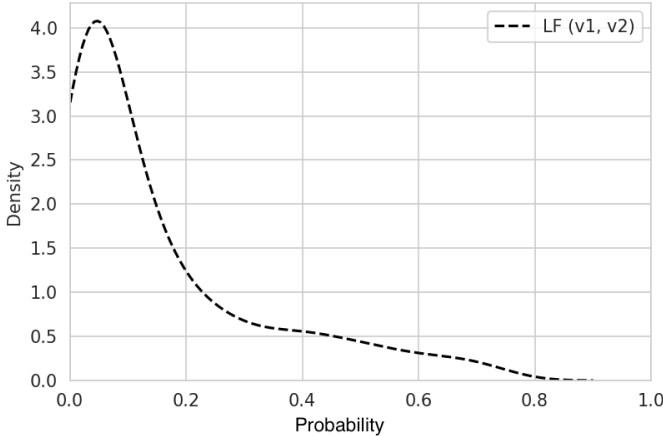

Figure 5: Marginal propensity scores for the LF (v1) and LF (v2) datasets. Note that the minimum probability of treatment in a random draw from the DGP is 0.007. The datasets are intentionally designed such that certain subgroups are unlikely to receive treatment, resulting in near-positivity violations.

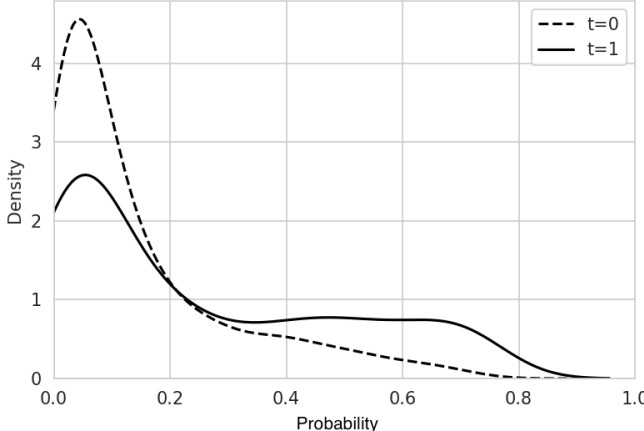

Figure 6: Propensity scores by treatment assignment for a sample from the LF (v1) dataset.

### 6.2.2 IHDP

The second dataset comprises 100 simulations from the well-known IHDP[6] dataset. We use the version corresponding with usual setting A of the NPCI data generating package Dorie, 2016 (see Shi et al., 2019; Shalit et al., 2017, and Yao et al., 2018) and comprises 608 untreated and 139 treated samples (747 in total). This variant actually corresponds with variant B from Hill (2011). There are 25 covariates, 19 of which are discrete/binary, and the rest are continuous. The outcome generating process is designed such that under treatment, the potential outcome is exponential, whereas under no treatment the outcome is a linear function of the covariates (Curth et al., 2021b).

This dataset represents a staple benchmark for causal inference in machine learning. However, it is worth noting that recent work has shown it to preferentially bias certain estimators (Curth et al., 2021b), so we include this dataset for completeness but discount our interpretation of the results accordingly.

### 6.3 Methods, and Evaluation Criteria

We evaluate a number of different methods in terms of their ability to estimate the ATE. A summary of the complete set of methods explored as part of the evaluation is shown in Table 1. As described above, we are interested in three properties relating to performance: estimation precision, robustness, and normality. Estimation precision is evaluated using mean squared error (MSE) calculated as $r^{-1} \sum_i^r [\hat{\tau}_i - \tau]^2$ where $r = 100$ is the number of simulations, and the standard error (s.e.) of the ATE estimates is computed as the standard deviation of $\hat{\tau}$. The MSE was chosen for its sensitivity to large errors (arguably important in the application domain of causal inference), and because it was found that it provided a more informative spread of results than the root-MSE. Robustness will be evaluated by comparing initial estimators that fail to exhibit the desired properties, with the results once these estimators have been updated. For normality, we examine the empirical distribution of the estimates. Using these distributions, we provide $p$-values from Shapiro-Wilk tests for normality (Shapiro & Wilk, 1965). Doing so provides an indication of the estimator's asymptotic linearity and whether the IFs are facilitating statistical inference as intended.

### 6.3.1 Algorithms/Estimators

For the outcome model $Q$ we compare linear/logistic regression (LR); a Super Learner (SL) comprising a LR, a LR with extra quadratic features, a Support Vector classifier, a random forest classifier (Breiman, 2001), a nearest neighbours classifier (Altman, 1992), and an AdaBoost classifier (Freund & Schapire, 1997); an implementation of the backbone to CounterFactural Regression network (without the integral probability

---

[6]Available from `https://www.fredjo.com/`

| Q Method | G Method | U Method | Datasets | Evaluation Criteria |
|---|---|---|---|---|
| Linear/Logistic Regression (Q-LR) | Linear/Logistic Regression (G-LR) | OneStep (U-ones) | LF (v1) n={500, 5000, 10000} | Mean Squared Error (MSE) |
| SuperLearner (Q-SL) | SuperLearner (G-SL) | Submodel (U-sub) | LF (v2) n={500, 5000, 10000} | Shapiro-Wilk Test ($p$) |
| CFR (Q-CFR) | CFR (G-CFR) | MultiStep (U-multi) | IHDP | Standard Error of Estimation (s.e.) |
| MultiNet (Q-MN) + variants | MultiNet (G-MN) + variants | Targeted Regularization (treg) | | |
| TVAE (Q-TVAE) | P-learner (G-P) | None (U-Base) | | |
| DragonNet (Q-D) | DragonNet (G-D) | | | |
| S-learner (Q-S) | | | | |
| T-learner (Q-T) | | | | |

Table 1: A summary of all variants and metrics explored as part of the evaluation. Note that additional results for other metrics (*e.g.*, mean absolute error) may be derived using the code in supplementary material.

Table 2: Hyperparameter search space for CFR and MN based methods.

| Parameter | Min | Max |
|---|---|---|
| Batch size | 10 | 64 |
| L2 Weight Penalty | 1e-5 | 1e-3 |
| No. of Iterations | 2000 | 10000 |
| Learning Rate | 1e-5 | 1e-2 |
| No. Layers | 2 | 14 |
| Dropout Prob. | 0.1 | 0.5 |
| No. Neurons per Layer | 5 | 200 |

metric penalty) (Shalit et al., 2017) (CFR); DragonNet (D) with and without targeted regularization (Shi et al., 2020); TVAE (Vowels et al., 2021) (which includes targeted regularization); T-learner (T) (Kunzel et al., 2019) with a gradient boosting machine (Friedman, 2001); S-learner (S) (Kunzel et al., 2019) with a gradient boosting machine (Friedman, 2001); and our MultiNet (MN) variants (*MN-Inc*, *MN-Casc*, *MN-Inc+LM*, *MN-Casc+LM*). When estimating the IF of the ATE, we also need estimators for the propensity score / treatment model, which we refer to as $G$. For this we use LR and SL, ElasticNet 'P-learner' (Zou & Hastie, 2005), DragonNet, as well as CFR and MN. The latter two NN methods must be modified for this task, and for this we simply remove one of the outcome arms, such that we can estimate $t|\mathbf{x}$. This seletion of algoritms was chosen to represent a suitable diverse set of common yet weak learners (*e.g.*, logistic regression), modern/well-known neural network causal inference methods (*e.g.*, CFR-net), neural network methods which already incorporate semi-parameteric techniques (DragonNet and TVAE), and recent and/or popular methods proposed in the causal inference or biostatistics literature (*e.g.* Super Learner, T- and S-learners).

The LR and SL approaches are implemented using the default algorithms in the scikit-learn package (Pedregosa et al., 2011), whilst the the DragonNet, S-learner, T-learner, and P-learner, are implemented using the CausalML package (Chen et al., 2020). For DragonNet the number of neurons per layer was set to 200, the learning rate set to $1 \times 10^{-1}$, number of epochs = 30, and batch size = 64. For TVAE the dimensionality of all latent variables was set to 5, the number of layers set to 2, batch size = 200, number of epochs = 100, learning rate = $5 \times 10^{-4}$, and targeted regularization weight of 0.1.

For CFR and MN, we undertake a Monte-Carlo train-test split hyperparameter search with 15 trials, for every one of the 100 samples from the DGP. The best performing set of hyperparameters is then used to train CFR and MN on the full dataset. For the hyperparameter search itself, we undertake 15 trials on a train/test split for each of the 100 samples from the DGP, and additional, separate hyperparameter searches are undertaken for methods using targeted regularization. The hyperparameters which are included in the search space for CFR and MN are present in Table 2. Note that the iteration count is not in terms of epochs - it represents the number of batches sampled randomly from the dataset. The number of iterations can be multiplied by the batch size and divided by the dataset size to approximately determine the equivalent number of epochs this represents.

Note that, unlike in traditional supervised learning tasks, using the full data with causal inference is possible because the target estimand is not the same quantity as the quantity used to fit the algorithms (Farrell et al., 2021). Indeed, whilst cross-fitting is used for the hyperparameter search, subsequent use of the full data has been shown to be beneficial, especially in small samples (Curth & van der Schaar, 2021). It is reassuring

to note that overfitting is likely to worsen our recorded estimates, rather than misleadingly improve them. Similarly, even though the SL is trained and the corresponding weights derived using a hold-out set, the final algorithm is trained on the full dataset for estimation. Logistic regression is simply trained on the full dataset without any data splitting. This is what motivated us to ask the question as to whether or not it is possible to have a 'free lunch' with IFs. Indeed, if no additional data is required, but we can nonetheless improve our estimates and achieve valid statistical inference (for the purposes, for example, of null hypothesis significance tests), then this would represent a valuable gain. For all treatment models, we bound predictions to fall in the range $[.025, .975]$ (Li et al., 2021).

### 6.3.2 Update Steps

We evaluate the onestep (U-ones), submodel (U-sub), MultiStep (U-multi), and targeted regularization (Treg) approaches to the update process.

The MultiStep update variants are optimized using the Adam (Kingma & Ba, 2017) optimizer. For small datasets ($n < 1000$) we undertake full gradient descent (*i.e.*, using the full data), and for larger datasets we use stochastic mini-batch gradient descent. The batch size for datasets with a sample size $n > 1000$ is set to 500, we undertake 4000 steps of optimization, and the learning rate for the Adam optimizer is set to $5 \times 10^{-4}$. The MultiStep objective has hyperparameters $\alpha_1$ and $\alpha_2$ which weight the constraints in the objective (expectation and variance of the influence function, respectively). We set both to one.

Table 3: Initial results over a restricted set of model variations. All update steps use the same propensity score G- algorithm as their Q-model algorithm, unless indicated by 'w/ G-SL', which indicates the use of a SuperLearner. Mean Squared Errors (MSE) and standard error (s.e.) (lower is better) and Shapiro-Wilk test $p$-values for normality (higher is better) for 100 simulations. Best results are those competing across all three dimensions. **Bold** indicates the best result for each algorithm, **bold and underline** indicates the best result *for each dataset variant*. Multiple methods may perform equally well.

| Dataset | Q Model | U-Base | | | U-ones | | | U-sub | | | Treg | | | Treg+U-sub | | | U-ones w/ G-SL | | | U-sub w/ G-SL | | |
|---|---|---|---|---|---|---|---|---|---|---|---|---|---|---|---|---|---|---|---|---|---|---|
| | | $p$ | MSE | s.e. | $p$ | MSE | s.e. | $p$ | MSE | s.e. | $p$ | MSE | s.e. | $p$ | MSE | s.e. | $p$ | MSE | s.e. | $p$ | MSE | s.e. |
| | LR | .001 | .0004 | .002 | **.276** | **.0007** | **.003** | .248 | **.0008** | **.003** | - | - | - | - | - | - | **.378** | **.0006** | **.003** | .591 | **.0008** | **.003** |
| | SL | .001 | .0004 | .002 | **.53** | **.0008** | **.003** | .651 | **.0009** | **.003** | - | - | - | - | - | - | - | - | - | - | - | - |
| | CFR | .0 | .0114 | .008 | .001 | .0042 | .004 | .01 | .01 | .003 | .07 | .0113 | .008 | .0 | .0105 | .002 | **.396** | **.0006** | **.003** | .909 | .0015 | .003 |
| LF (v1) | MN-Inc | .052 | .0008 | .003 | **.78** | **.0007** | **.003** | .394 | .001 | .003 | .729 | .0012 | .003 | .681 | .001 | .003 | **.639** | **.0008** | **.003** | .329 | .001 | .003 |
| | MN-Inc+LM | .135 | .0009 | .003 | .141 | .0007 | .003 | .578 | .0009 | .003 | .0 | .0017 | .004 | .957 | .0011 | .003 | **.969** | **.0008** | **.003** | .786 | .0009 | .003 |
| | MN-Casc | .0 | .0018 | .004 | .231 | .0014 | .002 | .0 | .0018 | .003 | .083 | .0086 | .007 | .702 | .0045 | .004 | **.831** | **.0007** | **.003** | .339 | .0009 | .003 |
| $n = 5000$ | MN-Casc+LM | .053 | .0058 | .006 | .018 | .002 | .003 | .204 | .0037 | .003 | .0 | .0091 | .008 | .74 | .0036 | .003 | **.747** | **.0007** | **.003** | .625 | .001 | .003 |
| | LR | .066 | .0024 | .002 | **.752** | **.0007** | **.003** | .497 | **.0008** | **.003** | - | - | - | - | - | - | **.785** | **.0007** | **.003** | .867 | **.0009** | **.003** |
| | SL | .349 | .0017 | .003 | **.938** | **.0008** | **.003** | .92 | **.0009** | **.003** | - | - | - | - | - | - | - | - | - | - | - | - |
| | CFR | .0 | .0185 | .01 | .0 | .006 | .005 | .0 | .0151 | .002 | .0 | .035 | .01 | .008 | .0162 | .002 | **.623** | **.0007** | **.003** | .065 | .0015 | .003 |
| LF (v2) | MN-Inc | .119 | .001 | .003 | **.204** | **.0006** | **.003** | .211 | .0008 | .003 | .002 | .0009 | .002 | **.029** | **.0008** | **.003** | .058 | .0007 | .003 | **.049** | **.0008** | **.003** |
| | MN-Inc+LM | .0 | .0011 | .003 | **.438** | **.0009** | **.003** | .813 | .0011 | .003 | .139 | .0071 | .005 | .678 | .0026 | .003 | **.959** | **.0005** | **.002** | .949 | .0009 | .003 |
| | MN-Casc | .0 | .002 | .004 | .013 | .0033 | .002 | .892 | .0043 | .003 | .77 | .014 | .006 | .365 | .0101 | .002 | **.272** | **.0007** | **.003** | .264 | .0011 | .003 |
| $n = 5000$ | MN-Casc+LM | .257 | .0113 | .007 | .349 | .0032 | .003 | .001 | .0083 | .002 | .066 | .0295 | .007 | .0 | .0112 | .002 | **.897** | **.0006** | **.003** | .241 | .0013 | .003 |
| | LR | .022 | .1818 | .019 | .0 | .0576 | .035 | .0 | .0461 | .044 | - | - | - | - | - | - | .0 | .1322 | .019 | **.0** | **.0597** | **.03** |
| | SL | .0 | .0466 | .032 | **.0** | **.0311** | **.033** | .0 | .0346 | .034 | - | - | - | - | - | - | - | - | - | - | - | - |
| | CFR | .0 | .7709 | .098 | .0 | .2865 | .074 | **.0** | **.0439** | **.052** | .0 | 25.5 | .3 | .0 | .0604 | .051 | .0 | .2626 | .063 | .0 | 1.7 | .114 |
| IHDP | MN-Inc | .0 | .0324 | .042 | .0 | .0297 | .044 | .0 | 8.7 | .299 | .0 | .0482 | .042 | .0 | 30.8 | .537 | **.0** | **.0243** | **.044** | .0 | .0425 | .042 |
| | MN-Inc+LM | .0 | .0393 | .045 | .0 | .0259 | .043 | .0 | .9849 | .099 | .0 | .1332 | .038 | .0 | 1.9 | .138 | **.0** | **.0243** | **.044** | .0 | .0327 | .042 |
| | MN-Casc | .0 | .1977 | .046 | .0 | .0737 | .04 | .0 | .064 | .04 | .0 | 2.9 | .115 | .0 | .102 | .042 | .0 | .0816 | .042 | **.0** | **.0383** | **.047** |
| $n = 747$ | MN-Casc+LM | .0 | 4.7 | .158 | .0 | 1.4 | .093 | .0 | .2118 | .049 | .0 | 23.9 | .164 | **.0** | **.1824** | **.06** | .0 | 1.1 | .079 | .0 | 4.7 | .202 |

## 7 Experimental Results

Given the large number of combinations in a full-factorial design (approximately 5000 results), we undertake an initial set of experiments to narrow down the evaluation space to focus on the most competitive methods. With this 'shortlist', we investigate the contribution of each Q-, G-, and U-method across the 7 different dataset variants.

### 7.1 Initial Evaluation

We share initial results in Table 3. These results were used to inform a subsequent set of experiments with a restricted set of variants. Specifically, we used these to select the most successful variant of MultiNet.

For **LF (v1)**, we see that the base CFR performs significantly worse in all considered metrics than LR and SL. Base LR and base SL achieved the best results in terms of MSE and s.e., although note that none of the base algorithms achieve asymptotic normality. Notice that LR's base MSE performance on LF (v1) is actually better than its MSE performance using the one-step and submodel updates. Such behaviour has been noted before by Luque-Fernandez et al. (2018), and occurs when the base learner is already close and/or when both outcome and treatment models are misspecified. Unlike CFR, our *MN-Inc* and *MN-Casc* variants worked well as either outcome or treatment models, yielding the best results with the one-step update. The other two of our *MN-* variants also performed well with the one-step and submodel updates but required a SL treatment model to do so.

The potential improvements for LR in combination with update steps is more striking for **LF (v2)**. Here, the LR base outcome model is misspecified (LF v2 has an exponential outcome model). Combining the LR with the SL one-step and submodel update processes enabled the LR method to perform well in spite of the non-linearity of the outcome. This is a demonstration of double-robustness - even though the outcome model is misspecified, the treatment model is not (or at least, it is sufficiently correctly specified), owing to the use of a SL, and the estimates are improved. As with the LF (v1) dataset, combining CFR with IFs resulted in a substantial improvement, especially when using an SL treatment model, yielding a competitive MSE, s.e., and normally distributed estimates (thus amenable to statistical inference). These results demonstrate the power of semiparametric methods for improving our estimation with NNs, and again illustrate the double-robustness property: the CFR outcome model was poorly specified, but was able to recover with an SL treatment model. Similar performance for our *MN-* variants on LF (v1) was observed with LF (v2).

Unfortunately, no method variant yielded normally distributed estimates with the **IHDP** dataset. The worst performing estimator across any combination of semiparametric techniques was LR. This makes sense given the non-linearity in the IHDP outcome process (Curth et al., 2021b). The SL with the one-step or submodel updates performed equally (poorly) as the best CFR and *MN-Casc* variants, although the SL provided a smaller s.e.. Overall, the best methods were our *MN-Inc* and *MN-Inc+LM* variants in combination with either a one-step update, or a one-step update using a SL treatment model.

The MultiNet variant which performed the best and most consistently across **all datasets** was our *MN-Inc* (or equally, *MN-Inc+LM*) with the one-step update. Whereas other methods benefited from the help of a SL treatment model, *MN-Inc* worked well as both an outcome and a treatment model, making it the best all-rounder across datasets, as well as the least dependent on the SL for correction. For all NN based approaches, targeted regularization made little difference, and sometimes resulted in instability and high MSEs. Further work is required to investigate this, although it may relate to which treatment model is used, and the associated sensitivity to positivity violations. A prior application also described the potential for the regularization to be inconsistent (Shi et al., 2019).

For all base learners, we observe the potential for improvement using the semiparameteric techniques, primarily for improving the associated MSE. It is also worth noting that in general, the base CFR method has consistently higher (*i.e.*, worse) s.e. than the *MN-variants*, although combining CFR with an udpdate step (*e.g.*, one-step w/ SL) significantly tightened the s.e..

In summary, we identified that CFR did not perform sufficiently well to warrant further investigation. Furthermore, the best performing MN variant was MN-Inc+LM, and we use this variant for the subsequent analyses. Finally, targeted regularization was inconclusive. However, previous work has identified its potential to improve DragonNet and TVAE (Shi et al., 2020; Vowels et al., 2021) and so we restrict the application of targeted regularization to these methods only, in the main evaluation presented below.

### 7.2 Main Evaluation

Owing to the large number of Q (outcome), G (propensity), and U (update step) method combinations, as well as the 7 different dataset variants and three different performance metrics (precision, normality, standard error), the number of results is large so we have attempted to summarize them in Figs. 7-11, but include further results in the Appendix in Table 4 and Figures 18-24. We also include the results of a Shapley value analysis (Shapley, 1953; Lundberg et al., 2020) of the results in Figures 12-14 (discussed further below).

Note that the following results do not include Q-CFR, G-CFR, or targeted regularization, as these were not shown to yield competitive performance in the initial evaluation above.

Whilst it is possible and potentially helpful to simply present the full set of results, it does not help us understand whether the use of particular Q-, G- or U-methods are more or less likely to improve or worsen the performance in any particular combination. Therefore, Figs. 7 and 8 provide results for $p(O|M) = p(M|O)p(O)/p(M)$ across the LF dataset variants for MSE and s.e., respectively. Here, $M$ is the method, and $O$ is the quantile (we split into 5 quantiles) for MSE and s.e., respectively. In words, the associated plots provide an estimation for the probability of achieving a performance result in each quantile $O$, for a given method $M$, thereby providing a means to directly assess the relative performance of each Q-, G-, and U-method. For instance, we can split the MSE results into equal probability quantiles, and count the number of times the use of each outcome, propensity score, and update method results in a performance which falls into each of these quantiles. Using Bayes rule we get an estimate for the probability of achieving results in a particular quantile (*e.g.*, the best performing methods fall in the zeroth quantile of MAE results), given a particular choice of method. Using these calculated probabilities, we also select all results from the best quantile, and see how the performance shifts over different sample sizes. Note that because these results are based on a rank ordering, it is not possible to judge absolute performance, only relative performance. Indeed, the purpose of the initial results above was to use the absolute performance as a way of shortlisting the methods so that a more comparative evaluation could be undertaken using the more competitive methods.

*Note:* When interpreting the results shown in Figures 7-10, it may be useful to recognise the 'ideal/desired' curve as one which takes on high values on the left-hand-side, representing a high-probability of the method yielding results in the top quantile(s). In contrast, curves which take on high values on the right hand side represent those with a high probability of yielding poor results in the lower quantile(s), relative to the other methods.

To evaluate the normality of the estimates, after calculating the *p*-value from the Shapiro-Wilk test, we calculate the proportion of each Q-, G-, and U-methods which yield normally distributed estimates ($p > 0.01$). For example, if a particular Q-method has a high 'probability of normality' according to *e.g.* Fig. 10, this means that a large proportion of the results yielded normally distributed estimates.

In Sections 7.2.1-7.2.7 we review the performance of each method for each of the three performance metrics in turn.

### 7.2.1   Q-Methods - MSE

Beginning with Fig. 7, the results for the outcome model Q-methods on the LF dataset variants are shown in the first column. In Fig. 7a we see that our Q-MN achieves the highest probability of being in the best quantile for MSE when used as an outcome model Q for **LF (v1)** $n = 500$, followed closely by Q-LR and Q-SL, and Q-TVAE and Q-D in the second-best quantile. In contrast, Q-D without targeted regularization, Q-T, and Q-S all had higher probabilities of yielding results in the later quantiles (*i.e.*, their performance was worse). Increasing the sample size to $n = 5000$, and considering Fig. 7d, we see similar results, with MN again yielding the highest probability of the achieving the best results, with Q-D, Q-S, Q-T, and Q-D without targeted regularization performing the worst. Finally, for LF (v1) $n = 10000$, we see in Fig. 7e that Q-MN is superseded by Q-LR and Q-SL, followed by Q-TVAE. Q-T, Q-S, and Q-D perform poorly again.

These results suggest that Q-LR and Q-SL perform consistently well over different sample sizes, and that Q-MN can perform well in small sample sizes, but may start to overfit as the sample size increases. Recall that the task of causal inference is different from the typical supervised learning task, and more data does not necessarily imply that it is easier to estimate the difference between two response surfaces, particularly when this difference (which is the treatment effect) is of low-complexity relative to the response surfaces themselves.

Now consider Figs. 7(j, m, p) for **LF (v2)**, which introduces additional non-linearity into the outcome model. We initially observe similar results for $n = 500$ in 7j, with Q-MN, Q-LR, and Q-SL achieving the best results, and Q-S, Q-D without targeted regularization, and Q-T populating the later quantiles. Increasing the sample size to $n = 5000$, we see in Fig. 7m that Q-TVAE now becomes the most likely to yield the best results,

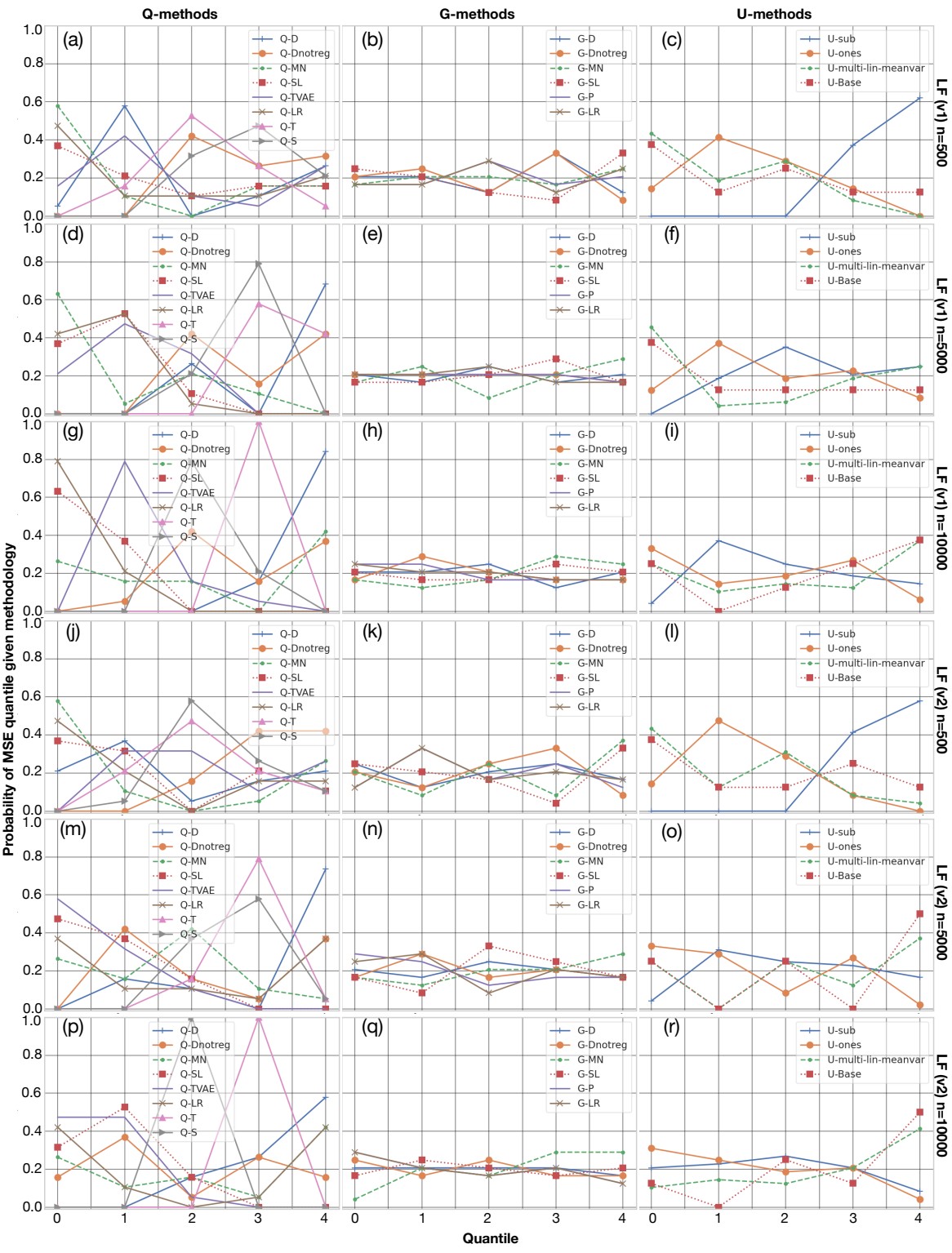

Figure 7: After recording the MSE for each Q (outcome), G (propensity score), and U (update step) method combination, we rank order them (from lowest to highest MSE), and calculate $p(O|M)$ where $O$ is the MSE quantile, and $M$ is the method. For 5 quantiles, this enables us to find *e.g.*, the probability of getting a MSE in the best quantile given a particular method $p(O = 0|M = m)$. If a method performs well, we expect to have high probability of achieving an MSE in the top two quantiles.

followed by Q-SL and, interestingly, Q-D without targeted regularization. Q-D, Q-T, and Q-S, however, still perform poorly. Finally, for $n = 10000$, we see Q-TVAE maintain the lead, once again followed by Q-SL. The worst performers were, again, Q-D, Q-S, and Q-T. This suggests once again that Q-SL provides consistent performance across sample sizes, and that Q-MN is a good option for smaller sample sizes.

For the IHDP dataset, we use a fixed sample size of $n = 747$, and the results are shown in Fig. 9. Here it can be seen that Q-T and Q-TVAE achieve the best results, followed by Q-S and Q-MN. The worst performer was Q-LR. These results are consistent with previous work which highlighted state-of-the-art performance of TVAE on IHDP (Vowels et al., 2021). Similarly, the fact that LR did so poorly possibly highlights the non-linearity of the data generating process for IHDP. The fact that Q-S and Q-T did so well is surprising given their relatively poor performance on the LF datasets described above. Such dataset dependence for the performance of causal estimators has also been previously noted by Curth et al. (2021b).

### 7.2.2 G-Methods - MSE

The MSE results for the propensity score G-methods can be seen in the second column of Figs. 7 and 9. Interestingly, there is very little dependence between the performance of the different methods. Arguably, there is some evidence that G-MN performs slightly worse than other methods in Fig. 7q, and that G-D performs worse in Fig. 9b but the differences are not convincing. This suggests that, at least in our experiments, the MSE results are relatively robust to the choice of propensity score model.

### 7.2.3 U-Methods - MSE

The MSE results for the update U-methods are shown in the third column of Fig. 7 for the LF datasets. In Fig. 7c we see that the U-Base model and the U-multi update methods perform the best, with the U-ones model close behind. The submodel update is more likely to be the lower quantiles. As the sample size increases to $n = 5000$ and $n = 10000$ in Figs. 7f and 7i we see the U-sub and, to a lesser extent, the U-ones performance shift. This behaviour has been observed before in work by Neugebauer & van der Laan (2005), who found that the performance of U-ones increased with sample sizes. Indeed, their own proposition for a multistep update process also performed more consistently in small samples, as does our U-multi. Similar patterns of performance are seen in Figs. 7l, 7o, and 7r for the LF (v2) dataset.

In Figure 9 we see that the U-sub and U-ones performed approximately equally well, whereas U-multi and U-Base had worse performance, relative to the other methods.

### 7.2.4 Q-Methods - s.e.

The standard error (s.e.) results are shown in Fig. 8 and the bottom row of plots in Fig. 9. Starting with Fig. 8a, we find the methods yielding the tightest distribution of estimates for the LF (v1) dataset $n = 500$ are Q-MN, Q-LR, and Q-TVAE, followed by Q-D, Q-SL, and Q-T. At the lower end we find Q-D without targeted regularization, and Q-S. As the same size increases to $n = 5000$ Q-MN provides estimates which are even more likely to be the tightest, followed again by Q-LR, Q-TVAE, and Q-SL. Q-D is not far behind, with Q-S, Q-T, and Q-D without targeted regularization performing the worst. With $n = 10000$, Q-MN is overtaken by Q-LR in terms of the tightness of the estimation, which is understandable given that Q-MN has a large number of hyperparameters (Q-LR has none), which contributes to variability in performance. Q-TVAE once again follows closely behind, with the worst performers being Q-D without targeted regularization, Q-S, and Q-T. Interestingly Q-MN exhibits a rise in the probability of being one of the worst performers, suggesting that there may exist better or worse combinations of G- and U-methods with Q-MN. Once again, it is worth consulting the full set of rank-ordered results in the Appendix. With the results for LF (v2) in Figs. 8j, 8m, and 8p we see a similar pattern of results, in spite of the introduction of additional non-linearity in this dataset variant.

Finally, for the IHDP results in Fig. 9d we see Q-LR and Q-SL provide the tightest estimates, followed by Q-MN, Q-D without targeted regularization, then Q-TVAE, Q-S, and Q-T.

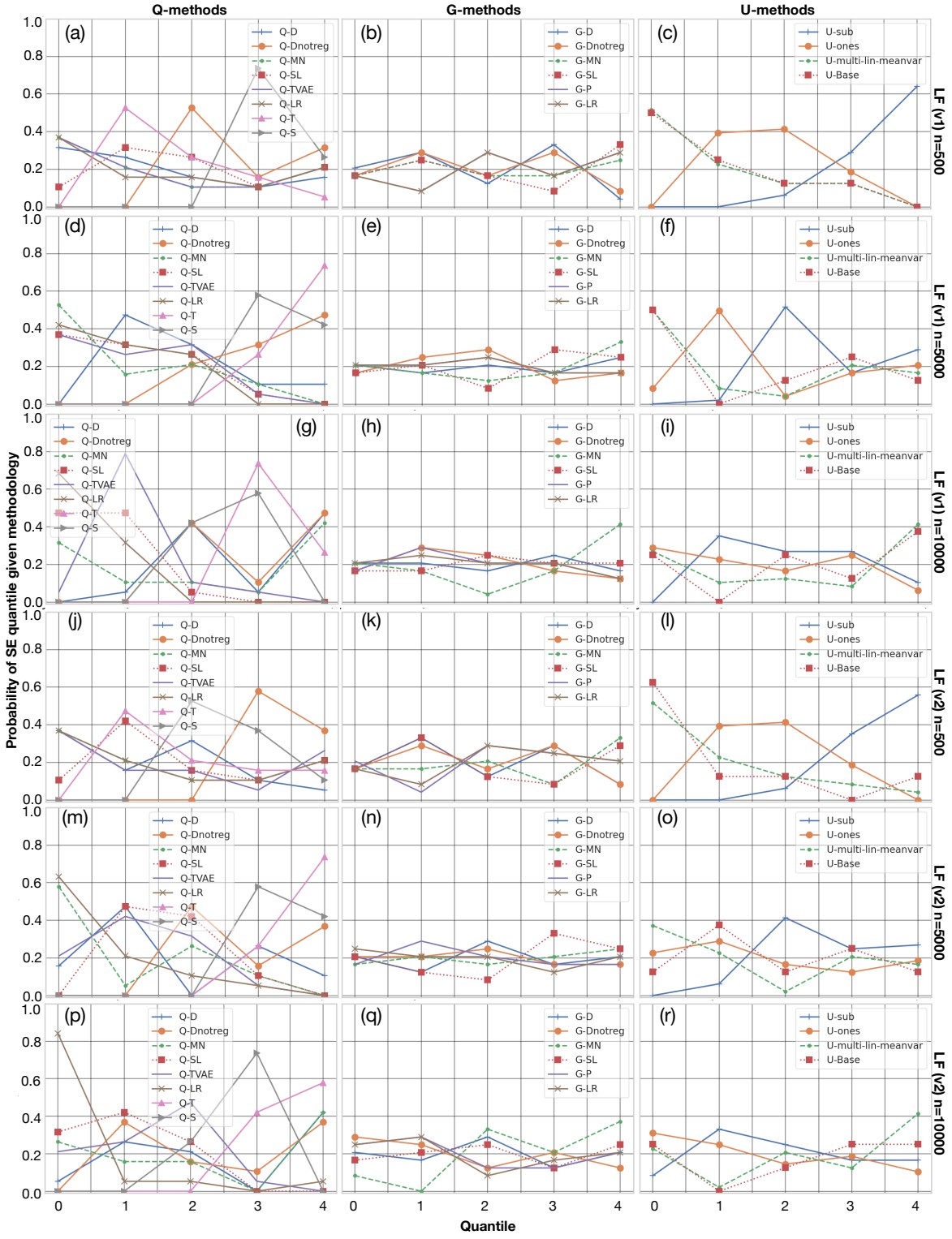

Figure 8: After recording the standard error (s.e.) of the 100 ATE estimates for each LF dataset and for each Q (outcome), G (propensity), and U (update step) method combination, we rank order them (from low to high), and calculate $p(O|M)$ where $O$ is the quantile, and $M$ is the method. This enables us to find the probability of getting a s.e. in the best quantile given a particular method $p(O = 0|M = m)$. If a method performs well, we expect to have high probability of achieving an s.e. in the top two quantiles.

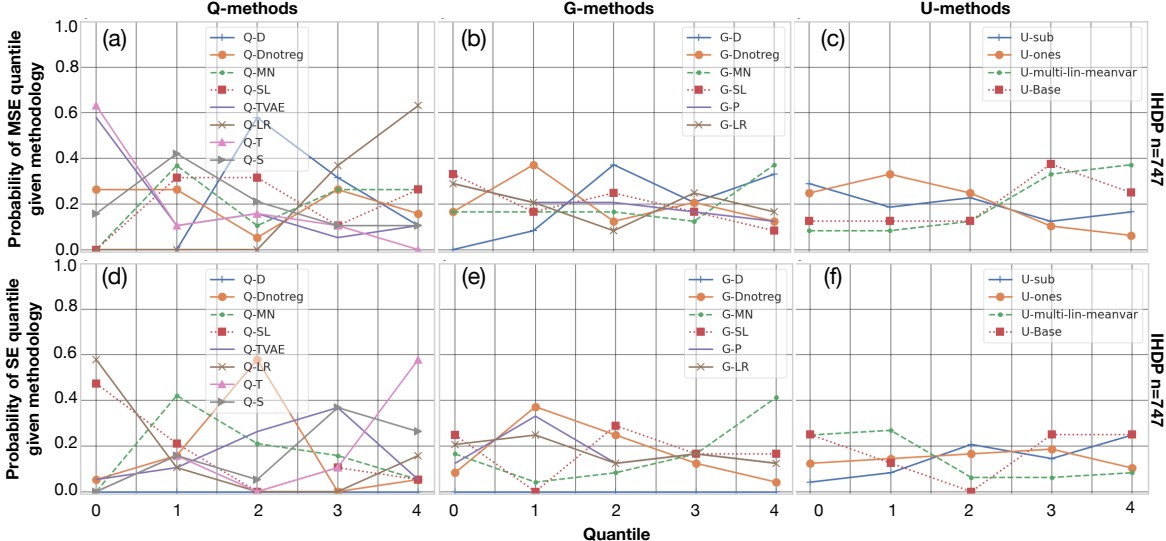

Figure 9: After recording the MSE and standard error (s.e.) of the 100 ATE estimates for the IHDP dataset and for each Q (outcome), G (propensity score), and U (update step) method combination, we rank order them (from lowest to highest), and calculate $p(O|M)$ where $O$ is the MSE (top row) or s.e. (bottom row) quantile, and $M$ is the method. For 5 quantiles, this enables us to find the probability of getting a MSE or s.e. in the best quantile given a particular method $p(O = 0|M = m)$. If a method performs well, we expect to have high probability of achieving an MSE and/or s.e. in the top first or second quantiles, and a low probability of achieving an MSE and/or s.e. in the last quantiles. Best viewed in colour.

### 7.2.5 G-Methods - s.e.

The s.e. results for the choice of propensity score G-method can be found in the central column of Fig. 8 and Fig. 9e. As was found for the MSE results, the choice of G-method was not decisive, besides the poor performance of G-MN for IHDP dataset, and for the $n = 10000$ LF datasets. It is reassuring to again find that the choice of G-method does not have a strong impact on the tightness of the estimates.

### 7.2.6 U-Methods - s.e.

The s.e. results for the choice of update U-method are presented in the right-hand column of Fig. 8 and Fig. 9f. In contrast to the choice of G-method, the choice of U-method had a significant impact on the tightness of the associated estimates, and the pattern of performance is similar to the pattern for MSE. For low sample sizes, it can be seen from both Figs. 8c and 8l that the tightest estimates are achieved using U-multi and U-Base, with U-sub yielding the least tight estimates. Increasing the sample size shifts the performance of U-sub and U-ones, making them competitive with the other methods. For the IHDP dataset, it can be seen in Fig. 9f that the choice of U-method had little impact on the tightness of the estimates, but the best performers were U-Base (*i.e.*, no update), and U-multi.

### 7.2.7 Q-, G-, U-Methods - Normality

The results evaluating the normality of the estimates are provided in Fig. 10 for the LF dataset variants, and Fig. 11 for IHDP. For the LF datasets, each plot provides the proportion of results from the respective method which yielded normally distributed estimates ($p > 0.01$) for each of the different dataset sizes $n = \{500, 5000, 10000\}$. In Fig. 11a it can be seen that most Q-methods performed well across all sample sizes with LF (v1), with the exception of Q-D which was less likely to yield normally distributed estimates, and we observe a drop in performance for Q-MN as sample size increases. Once again, and as indicated by Fig. 11b, the choice of G-method was not found to impact the likelihood of normally distributed estimates. Figs. 11c

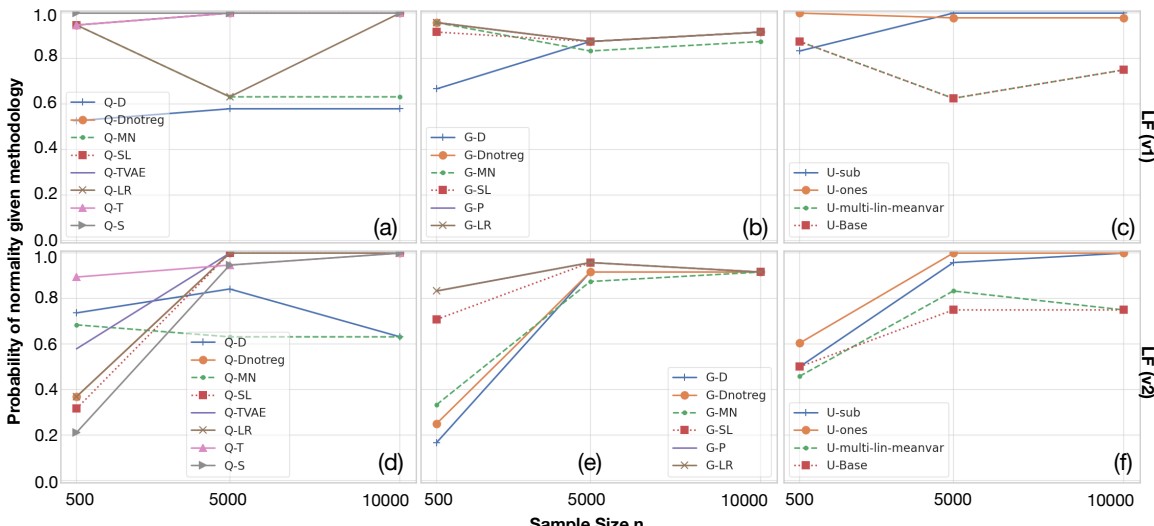

Figure 10: Probability of $p > 0.01$ for the Shapiro-Wilk test of normality for each Q (outcome), G (propensity score), and U (update step) method with the LF datasets $n = \{500, 5000, 10000\}$. Because we undertook all combinations of $G$ and $U$, each point represents a marginalization over the other dimension(s). For instance, for the Q methods, Q-D (DragonNet) is an average probability result when combining Q-D with all possible other G and U methods. Best viewed in colour.

indicates that the likelihood of U-Base and U-multi yielding normally distributed estimates dropped slightly with sample size, with U-ones yielding consistently normally distributed estimates regardless of sample size.

For LF (v2), the results in Figs. 11d-11f indicate more variability, possibly as a result of the additional non-linearity in the outcome model. When $n = 500$, the outcome Q-method most likely to yield normally distributed estimates was Q-T, followed by Q-D and Q-MN. However, for $n = 5000$ and $n = 10000$, the only methods not yielding consistently normally distributed results were Q-D and Q-MN. For the propensity score G-methods, the method most likely to yield normally distributed results with $n = 500$ was G-LR, followed by G-SL. The other methods did not perform well until the sample size was increased to $n = 5000$ or $n = 10000$ for which all methods performed equally well. For the U-methods, the best performing result across all sample sizes was U-ones, followed by U-sub, U-multi, and finally U-base.

Finally, the likelihood of achieving normally distributed estimates are shown in Fig. 11. The sample size is fixed for this dataset, and the results for the Q-, G-, and U- methods are presented together (hence the different graph format). It can be seen that Q-D provided the highest likelihood of normally distributed estimates, with the other methods yielding comparable (and low) likelihood. Similarly, G-D yielded the highest likelihood of normally distributed estimates, with the other G-methods being relatively equal (and low). Finally, none of the U-methods provided a high likelihood of normally distributed estimates.

## 7.3 Summary of the Main Evaluation

Note that in some Figures, certain methods may not have a monotonic probability which starts high and ends low, or vice versa. For example, in Fig. 7p, Q-LR has a u-shaped probability, suggesting that for some combinations of Q-LR with certain other G- and U-methods, its performance is good, and with others it is poor. In such cases it may be more informative to consult the full results in the Appendix, to attempt to understand whether there is any particular combination dependence.

### 7.3.1 MSE Summary

Our Q-MN performed well on the LF datasets, particularly in smaller samples. We found that both Q-LR and Q-SL also performed consistently across the different sample sizes, even with the introduction of non-linearity with LF (v2). Indeed, with the introduction of this non-linearity, we found Q-TVAE to yield good performance, and this competitive edge held up with IHDP as well. We did not find that the choice of G-method had a large impact on the results, although G-MN tended to do slightly worse. With smaller sample sizes $n = \{500, 5000\}$ and/or simpler datasets (LF v1), our U-multi performed the best as an update method. As sample size increased, we found that the onestep U-ones became the best performer, and similar behaviour has been found in other work (Neugebauer & van der Laan, 2005). For more complex datasets like IHDP, we found that U-ones and U-sub performed well.

### 7.3.2 Standard Error Summary

Once again, our Q-MN provided the tightest estimates, and did so consistently over all sample sizes and datasets except IHDP. The next best and most consistent estimator (including good performance on IHDP) in terms of the tightness of its estimates, was Q-SL. Once again, we did not find that the choice of G-method had a large impact on the results, but G-MN tended to do slightly worse than others. Our U-multi yielded consistently tight estimates across all datasets (including IHDP), although in general, the base models (without update steps) also performed well in this regard. As with the MSE results, U-ones and U-sub performed more competitively as the sample size increased.

### 7.3.3 Normality Summary

The choice of Q-method did not have a big impact on the likelihood of normally distributed estimates for the LF datasets, although Q-D performed poorly, and the performance of Q-MN dropped as sample size increased. Surprisingly, these results reversed for the IHDP dataset, with Q-D providing the most frequently normally distributed estimates, with the other methods yielding generally poor performance. Both G-LR and G-SL worked well as propensity score models for the LF-datasets, yielding a high likelihood of normally distributed estimates. However, on IHDP only the propensity score estimates from G-D were found to work well. U-ones and U-sub were found to yield consistently normally distributed errors across the LF datasets, with our U-multi unfortunately yielding little advantage over the base model.

In some ways, the relatively disappointing results with respect to the normality of the estimates is not surprising. Benkeser et al. (2017) and van der Laan (2014) showed that the double-robustness property relating to a normal limiting distribution which is afforded by estimators satisfying the efficient influence function does not apply when data-adaptive estimators are used (such as superlearners). In order for the double-robustness property to hold (with respect to the normal limiting distribution) with data-adaptive estimators, additional conditions must be satisfied. The failure to yield normally distributed estimates for many of the evaluated methods in this work thus may well be due to some degree of misspecification in the treatment or outcome models (or, indeed, both). One would expect that using the additional update steps proposed by Benkeser et al. (2017) and van der Laan (2014) would yield improved results and this presents a promising direction for future evaluations and development.

### 7.4 Shapley Value Analysis

In addition to the presentation of the results given in Figures 7-11, as well as those given below in Figures 18-24, we also explored whether a meta-analysis using the Shapley value approach (Shapley, 1953; Lundberg et al., 2020; 2017; Lundberg & Lee, 2017) could provide additional insights. Indeed, one of the limitations of the way the earlier results are presented is that they involve marginalization over one or more methodological components (*e.g.*, to obtain quantile probabilities for the Q-methods, we have to marginalize over all G- and U-methods). This can make it difficult to identify meaningful interactions between the choice of components.

The results presented in Figures 12-14, as well as those in the Appendix in Table 5 and Figures 15-17, represent the output from the SHapley Additive exPlanations (SHAP) machine learning explainability technique (Lundberg et al., 2020). The process is as follows: we take the full set of factorial results from our

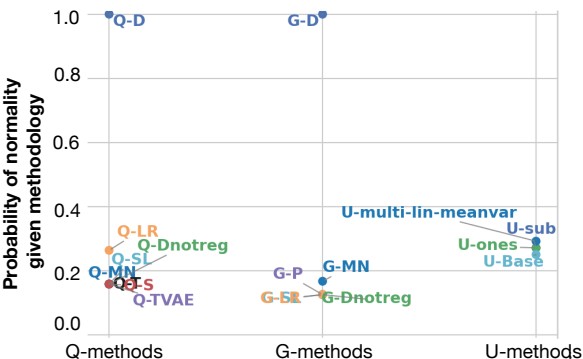

Figure 11: Probability of $p > 0.01$ for the Shapiro-Wilk test of normality for each Q (outcome), G (propensity score), and U (update step) method with the IHDP dataset. Because we undertook all combinations of $G$ and $U$, each point represents a marginalization over the other dimension(s). For instance, for the Q methods, Q-D (DragonNet) is an average probability result when combining Q-D with all possible other G and U methods. Best viewed in color.

experiments (all combinations from Table 1), and use the choice dataset and choice/combination of Q-, G-, and U-methods as predictors in a regression with each of the MSE, ATE estimate standard error, and Shapiro-Wilk test $p$-value results as different outcomes. We use a random forest algorithm (Breiman, 2001) as the regressor, with the default values in the scikit-learn implementation which have been shown to yield stable and consistent performance (Pedregosa et al., 2011; Probst et al., 2018). Then, the SHAP package conceives of the regression task as a game, with each predictor representing an agent which, in collaboration with other predictors, seek to maximise the performance of the regressor. The output of the Shapley analysis includes global predictor importances (which tell us, on average, how useful each predictor is in explaining the regressor output), the individual impact of each datapoint in the dataset on the regressor's output, and a quantification of the degree of interaction between predictors. These results therefore help us identify whether there exist strong interactions between the choice of methods and/or datasets.

Figures 12, 13, and 14 show Shapley interaction heatmaps for the three outcomes MSE, ATE estimate standard error, and $p$-values, respectively. Brighter values indicate the presence of an interaction between predictors (where the combined information tells us something more about the likely value of the outcome than the individual predictors alone).

For MSE in Figure 12, we see interactions between the dataset, Q-CFR, G-MN, and, to a lesser extent, the U-sub update approach. Otherwise, practically no other methods show strong interactions with each other or the datasets, lending weight to the marginalized results for MSE presented in previous sections.

For the ATE estimate standard error results in Figure 13, we again see some strong interaction between Q-CFR and the dataset, as well as the U-multi without the mean-zero constraint and the dataset. Otherwise, there are few notable interactions indicated by these results.

Finally, for the Shapiro-Wilk test for normality $p$-value results in Figure 14, we predominantly see an interaction between the sample size and the dataset. This link is, perhaps, not quite as obvious as it might immediately appear given that $p$-values are functions of sample size. Remember here that the $p$-value is an empirical test for the normality of the distribution of the estimates that we compute over the number of simulations, which is fixed across datasets/experiments. We also see some decreasing interactions between sample sze and the update step, as well as Q-D, Q-MN, and Q-T.

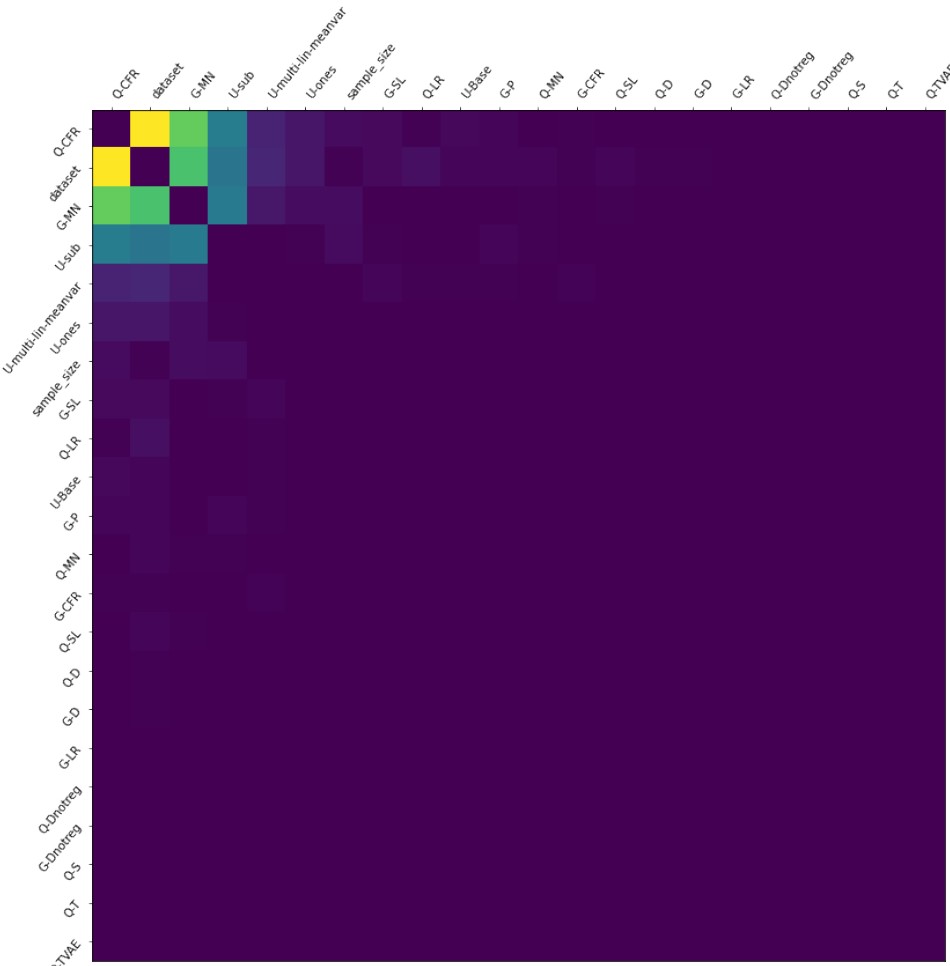

Figure 12: Shapley predictor interaction heatmap for MSE outcome.

## 8 Discussion

In this paper we have introduced some key aspects of semiparametric theory and provided the expression and code for deriving influence functions for estimands from a general graph automatically. We have undertaken an comprehensive evaluation of the potential of semiparametric techniques to provide a 'free' performance improvement for existing estimators without needing more data, and without needing to retrain them. We also proposed a new pseudo-ensemble NN method 'MultiNet' for simulating an ensemble approach with a single network, a new update step variant 'MultiStep'. Our evaluation included a discussion of the choice of outcome 'Q' method, propensity score 'G' method, and the update 'U' method.

The summary of results is fairly nuanced, and even methods which yielded the best results were subject to variation across datasets and sample size (this was particularly evident when comparing the results on the LF datasets with those of the IHDP dataset, and when reviewer the results of the Shapley interaction plots). This highlights a dependence of the performance on the method-dataset combination which is difficult to alleviate. That said, some methods were more stable across datasets than others. A review of the Shapley interaction plots indicated relative stability of performance in terms of MSE, ATE estimate standard error, and $p$-values for all methods evaluated in the main evaluation, besides Q-CFR, G-MN, Q-D, and G-D.

Such dataset dependence was highlighted by Curth et al. (2021b), and it is something which practitioners should be aware of, especially in the causal inference setting where we do not have access to ground-truth.

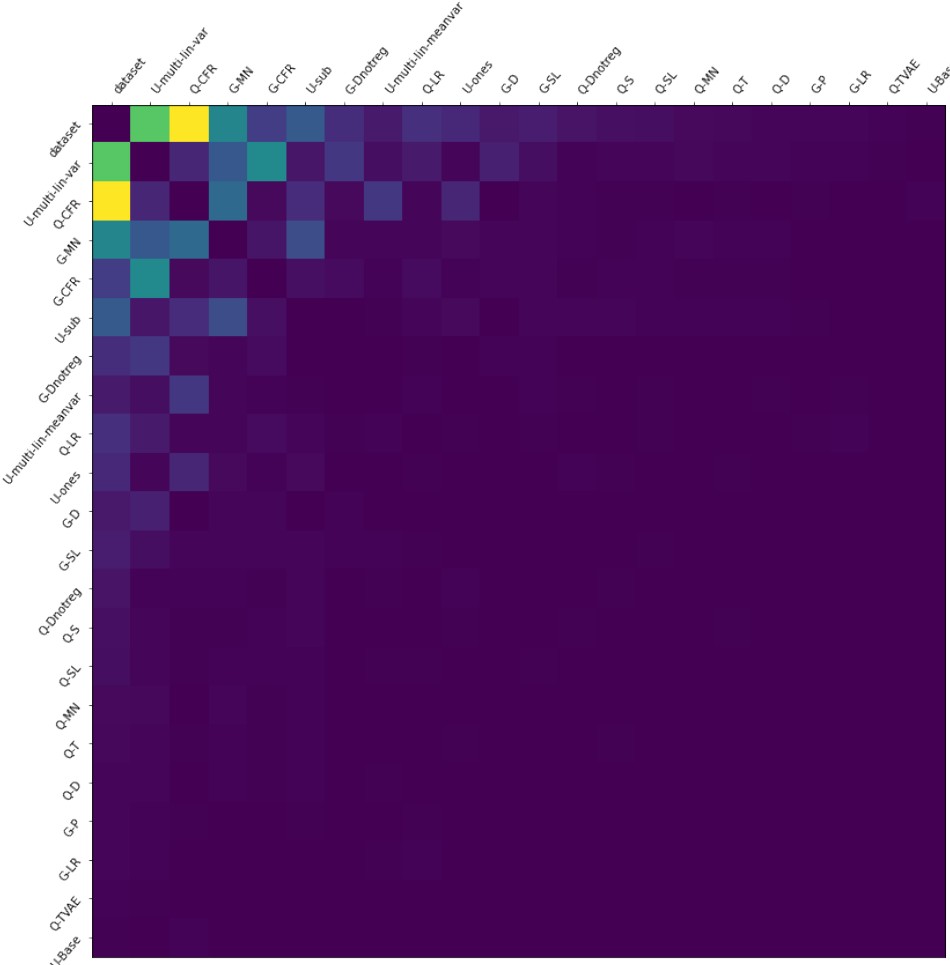

Figure 13: Shapley predictor interaction heatmap for ATE estimate standard error outcome.

Researchers developing such methods should also, of course, be aware of this issue, because it can significantly inform the evaluation design for testing and comparing different methods. These caveats notwithstanding, we found our MultiNet method to perform well as an outcome method, yielding state of the art on a number of evaluations, and performing particularly well on datasets with smaller sample sizes. Indeed, to an extent these results conflict with those of Farrell et al. (2021), who showed that relatively basic neural networks were capable of excellent performance when combined with semiparametric techniques and evaluated on their own simulated data as well as data for a direct mail marketing campaign. Arguably, their conclusions and results are less mixed than ours, although it is worth remembering that we restricted the set of methods used for the main evaluation to those with already competitive performance, and the remaining spread of our 'mixed' results may already be reassuringly tight. Unfortunately, it is difficult to say what is an acceptable level of performance, although recent large-scale work by Gordon et al. (2022) suggests that the primary challenge will be in satisfying identifiability - that is, ensuring that our estimand can be expressed as a function of the observational data, that our model is structurally well-specified, and that no unobserved confounders exist which otherwise bias our estimates.

Many of the methods failed to yield normally distributed estimates. This is somewhat expected given that the double robustness guarantees do not apply to the limiting distribution. Benkeser et al. (2017) and van der Laan (2014) provide a means to augment the update step frameworks to include additional conditions which, when satisfied, extend the double robustness guarantees to the (normal) limiting distribution of the estimates.

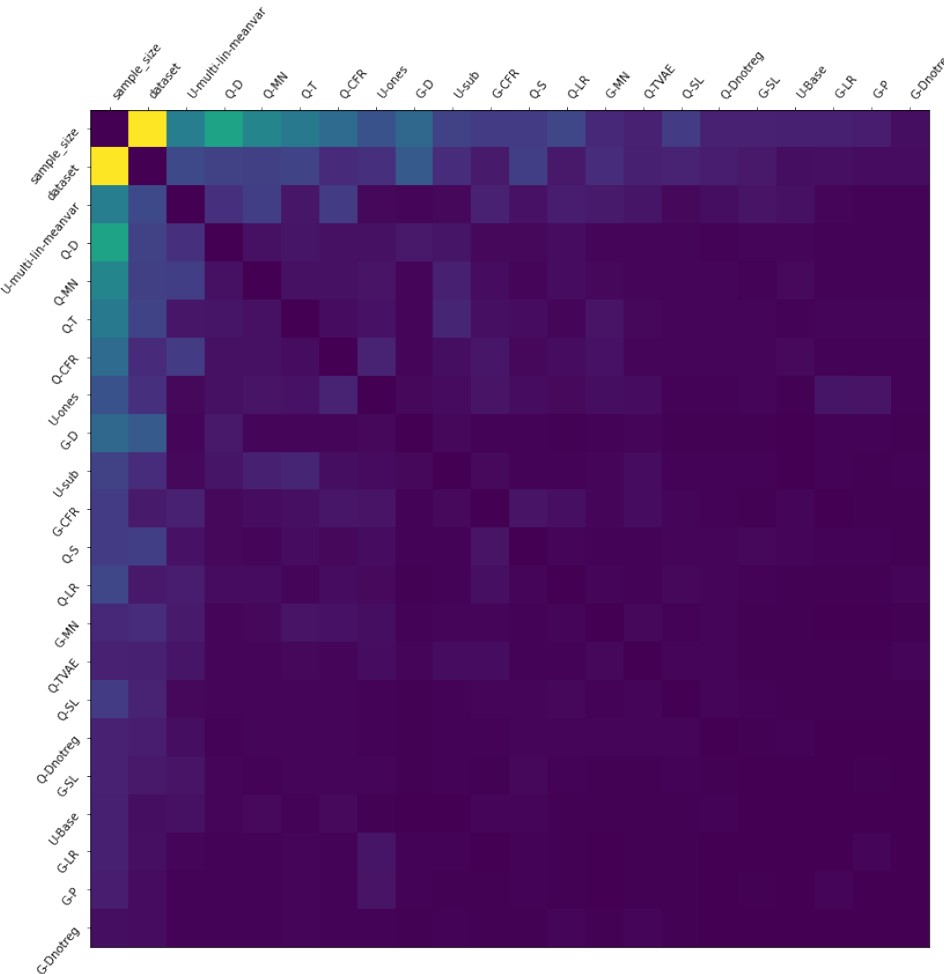

Figure 14: Shapley predictor interaction heatmap for Shapiro-Wilk *p*-value outcome.

Many open questions remain: a similar set of experiments should be undertaken for other estimands (such as the conditional ATE). Also, one may derive higher order IFs (Carone et al., 2014; van der Laan et al., 2021; van der Vaart, 2014; Robins et al., 2008) which introduce new challenges and opportunities. Additionally, it may be possible to use IFs to derive a proxy representing 'good enough'-ness, *i.e.*, whether the initial estimator is close enough to the target estimand for the remaining bias to be modelled linearly. This, in turn, may also provide a way to assess the performance of causal inference methods, which would be highly advantageous given that explicit supervision will rarely be available in real-world causal inference settings. The extensions of Benkeser et al. (2017) and van der Laan (2014) also represent an interesting avenue for further development, particularly in relation to the goal of undertaking valid statistical inference with nonparametric estimators.

## 9   Broader Impact

It is always important to remember that the reliability of causal inference depends on strong, untestable assumptions (not least because there is rarely any access to ground-truth in the domain of causal inference). Given the variability of the performance of the evaluated methods across datasets, in particular with regards to the normality of the estimates (and therefore also the validity of subsequent inference) any practical application of causal inference methods must be undertaken with caution. Indeed, we recommend researchers

establish the extent to which their inference depends on the methods used, by undertaking the same analysis with multiple approaches/estimators.

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

# A  Things that Did Not Work

## A.1  Calibration

One of the initial possibilities that we considered which might explain why some methods (*e.g.*, CFR) were not performing as well as others, was that the calibration of the output might be poor (Guo et al., 2017). However, we tried calibrating the trained outcome and treatment model networks using temperature scaling. We found it to be unsuccessful, and we leave an exploration of why it failed to future work.

## A.2  Restricted Hyperparameter Search

Additionally, we tried only performing hyperparameter search with a held-out test set *once* at the beginning of the 100 subsequent simulations for each model and dataset variant, rather than performing it for every single simulation. This did not work, and we found that if the first network 'designed' through hyperparameter search happened to be degenerate with respect to its performance as a plug-in estimator (notwithstanding its potentially adequate performance as an outcome model), then it will be degenerate for all simulations, and yield incredibly biased results. However, performing hyperparameter search for every simulation more accurately represents the use of these algorithms in practice.

This problem also highlights the importance of fitting multiple neural networks on the same data. As supervision is not available, the usual metrics for hyperparameter search (based on *e.g.*, held out data loss scores) can be a poor indicator for the efficacy of the network as a plug-in estimator. By re-performing hyperparameter search, even on the same data (put perhaps, with different splits), one can effectively bootstrap to average out the variability associated with the hyperparameter search itself. Indeed, as the results show, the average estimates for the ATE using CFR net are close to the true ATE, even if the variance of the estimation is relatively high. We leave a comparison of the contribution of variance from hyperparameter search to further work.

## A.3  MultiStep Update Variants

Relating to our proposed MultiStep objective, we also tried a non-linear, generalized variant with an objective which still attempts to minimize the mean and variance of the influence function but such that the update step is parameterized as follows:

$$\hat{Q}(t, \mathbf{x}_i) + g_\theta(\nu_1 \hat{Q}(t, \mathbf{x}_i), \nu_2 H(\mathbf{z}_i)) \tag{39}$$

It can be seen that instead of optimizing over the domain of $\hat{\gamma} \in \Gamma$ in Eq. 33, we instead optimize over $\theta \in \Theta$, where $\theta$ are the parameters of a shallow NN function $g$. Here, $\nu_1 \in \{0, 1\}$ and $\nu_2 \in \{0, 1\}$ are hyperparameters determining whether the NN function $g_\theta$ should be taken over just the clever covariate $H$, or over both the clever covariate and the outcome model $Q$.

In practice however, this approach did not yield good estimates. Furthermore, we found that MultiStep update step given in Eq. 32, setting $\alpha_1 = 0$ (*i.e.*, no mean-zero penalty) also did not work well. This result was surprising because a similar approach in Neugebauer & van der Laan (2005), which did not include a mean-zero penalty, yielded an improvement. However, it is also intuitive that if the two properties of the Efficient Influence Function are (1) mean-zero and (2) minimum variance, then it makes sense that an optimization objective should benefit from the inclusion of both of these conditions.

Table 4: Unmarginalized results over a restricted set of model variations. Mean Squared Errors (MSE) and standard error (s.e.) (lower is better) and Shapiro-Wilk test *p*-values for normality (higher is better) are provided ad computed over 100 simulations. Best results are those competing across all three dimensions. **Bold** indicates the best result for each algorithm, **__bold and underline__** indicates the best result *for each dataset variant*. Multiple methods may perform equally well.

| Dataset | Q Model | U-Base | | | G-SL + U-multi | | | G-SL+sub | | | G-MN+U-multi | | | G-MN+U-sub | | |
|---|---|---|---|---|---|---|---|---|---|---|---|---|---|---|---|---|
| | | $p$ | MSE | s.e. | $p$ | MSE | s.e. | $p$ | MSE | s.e. | $p$ | MSE | s.e. | $p$ | MSE | s.e. |
| LF (v1) ($n=500$) | Q-MN | **__0.985__** | **__0.0024__** | **__0.054__** | **__0.959__** | **__0.0023__** | **__0.054__** | 0.027 | 0.0104 | 0.109 | **__0.945__** | **__0.0024__** | **__0.044__** | 0.125 | 0.0080 | 0.093 |
| | Q-SL | 0.030 | 0.0032 | 0.057 | 0.034 | 0.0036 | 0.060 | 0.023 | 0.0091 | 0.102 | 0.043 | 0.0035 | 0.060 | 0.093 | 0.0081 | 0.095 |
| | Q-TVAE | 0.617 | 0.0053 | 0.050 | 0.386 | 0.0042 | 0.052 | 0.074 | 0.0104 | 0.108 | **0.716** | **0.0046** | **0.052** | 0.049 | 0.0083 | 0.093 |
| | Q-T | 0.700 | 0.0059 | 0.064 | 0.772 | 0.0056 | 0.065 | 0.495 | 0.0084 | 0.099 | **0.843** | **0.0057** | **0.064** | 0.059 | 0.0073 | 0.089 |
| LF (v1) ($n=5000$) | Q-MN | 0.000 | 0.0006 | 0.025 | 0.0 | 0.0005 | 0.024 | **0.599** | **0.0009** | **0.031** | 0.0 | 0.0006 | 0.025 | 0.583 | 0.0307 | 0.030 |
| | Q-SL | 0.024 | 0.0004 | 0.022 | 0.055 | 0.0005 | 0.023 | **0.536** | **0.0009** | **0.030** | 0.012 | 0.0004 | 0.023 | 0.224 | 0.0008 | 0.029 |
| | Q-TVAE | 0.444 | 0.0007 | 0.024 | 0.182 | 0.0006 | 0.025 | **0.655** | **0.0009** | **0.031** | 0.368 | 0.0007 | 0.024 | 0.414 | 0.0008 | 0.030 |
| | Q-T | **__0.975__** | **__0.0010__** | **__0.032__** | 0.989 | 0.0010 | 0.032 | 0.935 | 0.0010 | 0.033 | 0.823 | 0.0010 | 0.032 | 0.854 | 0.0010 | 0.033 |
| LF (v1) ($n=10000$) | Q-MN | 0.0 | 0.0008 | 0.027 | 0.0 | 0.0007 | 0.025 | 0.366 | 0.0004 | 0.020 | 0.0 | 0.0006 | 0.023 | **0.648** | **0.0006** | **0.022** |
| | Q-SL | 1.0 | 0.0002 | 0.015 | **0.993** | **0.0002** | **0.016** | 0.625 | 0.0004 | 0.019 | **__0.997__** | **__0.0002__** | **__0.015__** | 0.996 | 0.0004 | 0.019 |
| | Q-TVAE | 0.666 | 0.0004 | 0.019 | 0.956 | 0.0004 | 0.019 | 0.392 | 0.0004 | 0.020 | 0.782 | 0.0004 | 0.018 | 0.861 | 0.0004 | 0.020 |
| | Q-T | 0.341 | 0.0004 | 0.020 | 0.331 | 0.0004 | 0.021 | 0.326 | 0.0004 | 0.021 | 0.141 | 0.0004 | 0.021 | 0.122 | 0.0004 | 0.021 |
| LF (v2) ($n=500$) | Q-MN | **0.439** | **0.0018** | **0.043** | 0.149 | 0.0018 | 0.044 | 0.014 | 0.0119 | 0.114 | 0.075 | 0.0019 | 0.044 | 0.0 | 0.0102 | 0.108 |
| | Q-SL | 0.0 | 0.0027 | 0.054 | 0.001 | 0.0029 | 0.057 | 0.025 | 0.0097 | 0.103 | 0.0 | 0.0029 | 0.057 | 0.0 | 0.0095 | 0.103 |
| | Q-TVAE | 0.097 | 0.0075 | 0.043 | 0.031 | 0.0059 | 0.045 | 0.046 | 0.0125 | 0.114 | 0.016 | 0.0067 | 0.045 | 0.0 | 0.0112 | 0.112 |
| | Q-T | **__0.817__** | **__0.0066__** | **__0.064__** | 0.721 | 0.0062 | 0.065 | 0.047 | 0.0094 | 0.102 | **0.753** | **0.0064** | **0.065** | 0.003 | 0.0110 | 0.108 |
| LF (v2) ($n=5000$) | Q-MN | 0.0 | 0.0011 | 0.029 | 0.0 | 0.0009 | 0.027 | 0.925 | 0.0010 | 0.031 | 0.0 | 0.0011 | 0.027 | **0.662** | **0.0010** | **0.030** |
| | Q-SL | **0.765** | **0.0008** | **0.028** | 0.669 | 0.0008 | 0.028 | **__0.921__** | **__0.0009__** | **__0.030__** | 0.661 | 0.0007 | 0.028 | 0.678 | 0.0008 | 0.030 |
| | Q-TVAE | 0.607 | 0.0007 | 0.028 | 0.795 | 0.0007 | 0.027 | **__0.912__** | **__0.0009__** | **__0.031__** | **0.799** | **__0.0006__** | **__0.027__** | 0.672 | 0.0008 | 0.029 |
| | Q-T | **0.793** | 0.0010 | 0.032 | 0.644 | 0.0010 | 0.032 | 0.724 | 0.0011 | 0.033 | 0.755 | 0.0010 | 0.032 | 0.0 | 0.0010 | 0.037 |
| LF (v2) ($n=10000$) | Q-MN | 0.0 | 0.0009 | 0.027 | 0.0 | 0.0008 | 0.025 | **0.707** | **0.0005** | **0.020** | 0.0 | 0.0011 | 0.025 | **0.783** | **0.0009** | **0.026** |
| | Q-SL | **__0.859__** | **__0.0004__** | **__0.019__** | 0.649 | 0.0004 | 0.019 | **0.964** | 0.0005 | 0.020 | 0.594 | 0.0004 | 0.019 | 0.676 | 0.0004 | 0.021 |
| | Q-TVAE | 0.0 | 0.0177 | 0.412 | 0.448 | 0.0004 | 0.021 | **0.981** | 0.0004 | 0.020 | 0.240 | 0.0004 | 0.021 | 0.227 | 0.0004 | 0.021 |
| | Q-T | 0.0 | 0.0124 | 0.430 | **0.719** | **0.0005** | **0.022** | 0.566 | 0.0005 | 0.022 | 0.574 | 0.0005 | 0.022 | 0.285 | 0.0005 | 0.022 |
| IHDP ($n=747$) | Q-MN | 0.0 | 0.0647 | 0.409 | 0.0 | 0.0451 | 0.405 | 0.0 | 0.0390 | 0.400 | 0.0 | 0.0515 | 0.411 | 0.0 | 0.614 | 0.844 |
| | Q-SL | 0.0 | 0.0440 | 0.317 | 0.0 | 0.0353 | 0.346 | 0.0 | 0.0340 | 0.344 | 0.0 | 0.0390 | 0.346 | 0.0 | 0.625 | 0.846 |
| | Q-TVAE | 0.0 | 0.0178 | 0.412 | 0.0 | 0.0132 | 0.404 | 0.0 | 0.0134 | 0.405 | 0.0 | 0.0142 | 0.408 | 0.0 | 0.191 | 0.582 |
| | Q-T | 0.0 | 0.0124 | 0.430 | 0.0 | 0.0123 | 0.422 | 0.0 | 0.0118 | 0.424 | 0.0 | 0.0125 | 0.426 | 0.0 | 0.0293 | 0.446 |

# B    Additional Results/Analysis

In Table 4 we also provide a set of results (without any marginalization over any of the G-, U-, or Q-dimensions) for a subset of the methods considered in the full-factorial design. Note that whilst we have tried to highlight the best results in bold and underline, many of the results are close/competitive and illustrate (again) that the performance is dataset and combination dependent, as well as that there exist multiple possible 'best' options for a given situation.

## B.1    Shapley Results

In addition to the interaction plots given in the text, here we provide results for the regressor performance and the predictor impacts.

### B.1.1    Regressor Performance

Table 5 show the performance of the random forest regressor in predicting the three outcomes (MSE, ATE estimate standard error, and *p*-value). These results help us understand whether there is any information in the set of predictors which is useful in predicting the outcome (and therefore, in turn, whether there exist any potentially meaningful patterns). We provide results for the fraction of explained variance, $R^2$,

Table 5: Meta-analysis results for random forest regression performance for MSE, standard error of the estimates (ATE s.e.) and $p$-values as the outcomes. Results for $R^2$, explained variance, and MSE are given as the mean $\pm$ the standard derivation across the 10-fold cross-validation procedure.

| Outcomes: | MSE | ATE s.e. | $p$-value |
|---|---|---|---|
| $R^2$ | 0.42±0.382 | 0.66±0.400 | 0.78±0.266 |
| Explained Variance | 0.43±0.380 | 0.66±0.400 | 0.78±0.266 |
| MSE | 0.201±0.580 | 0.039±0.078 | 0.024±0.058 |

and MSE. The results are averages over a 10-fold cross-validated evaluation. It is useful to interrogate this table first to understand whether any further investigation is needed - if the predictive performance is poor, there is no point explaining the regressor's behaviour; if it is good, then it is worth investigating what the regressor is using to achieve that performance.

From Table 5 it can be seen that all outcomes were somewhat well predicted, with the arguable exception of the MSE, which had large standard errors for the $R^2$ and fraction of explained variance suggesting that test data in some of the folds in the 10-fold cross-validation scheme were poorly predicted, whilst others were predicted well. Even though $R^2$ and explained variance are not reliable metrics in non-linear regression tasks, they can nonetheless be seen that a relatively high R2 is achieved, indicating that information about the MSE is predictable from the set of predictors. This was especially true for the prediction of the Shapiro-Wilk test $p$-values and ATE estimate standard error, which both had average fraction of explained variance and average $R^2$ greater than 0.7.

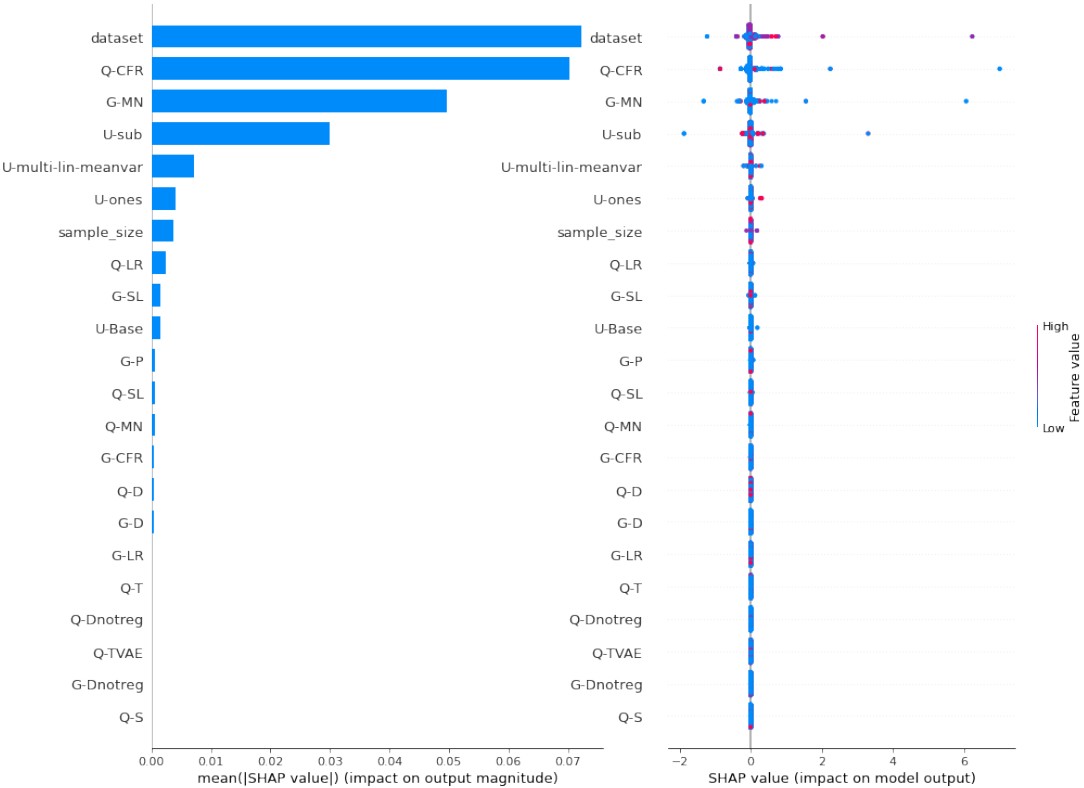

Figure 15: Shapley predictor random forest regressor impact results for the MSE.

## B.1.2 Shapley Predictor Impact

Given the regression performance results indicate some patterns exist in the data, let us now turn to the Shapley explainability results. For each of the three respective outcomes (MSE, ATE estimate standard

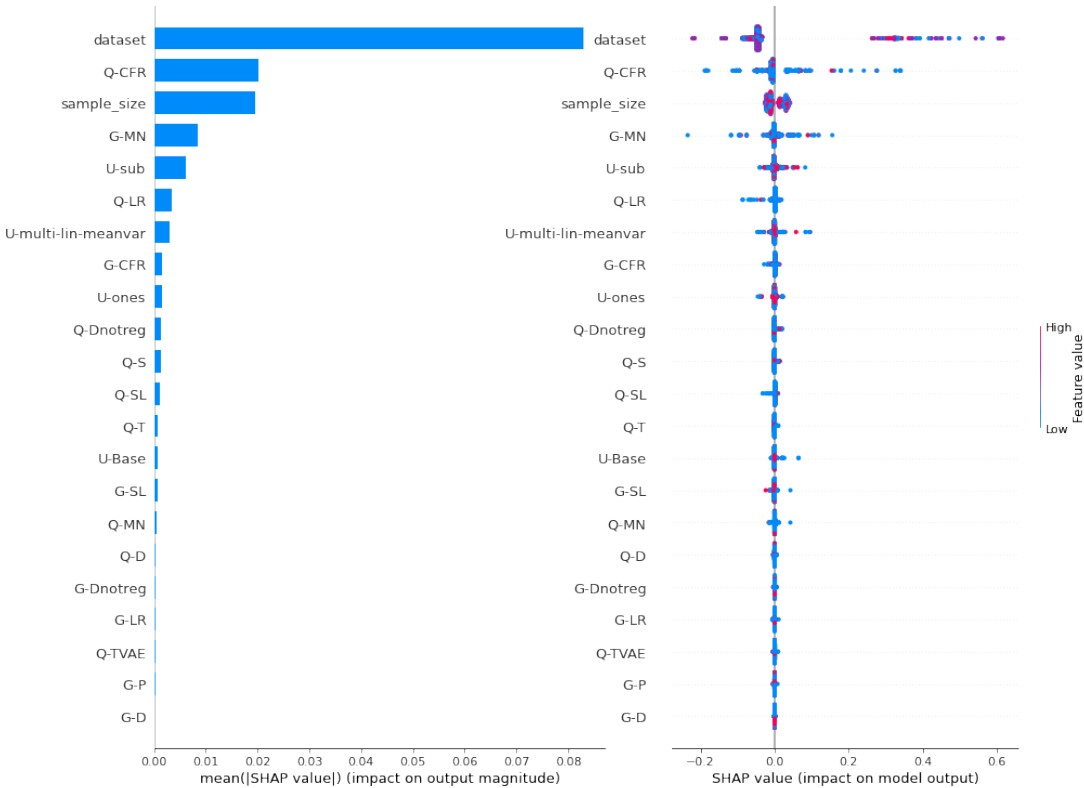

Figure 16: Shapley predictor random forest regressor impact results for the ATE estimate standard error.

error, and $p$-values), Figures 15, 16, and 17 depict the global predictor impacts on the regressor (left-hand plots) as well as the per-predictor, per-datapoint impact on the regressor (right-hand plots).

The right-hand side plots in Figures 15-17 are useful in visualising the spread of impact, and in which direction this impact is (*i.e.*, does it push the prediction of the outcome up or down in value). For example, in Figure 15 one can see that the dataset is the most important predictor for predicting MSE, followed by Q-CFR, G-MN, and U-sub. We see that most of the impact (for all predictors) is clustered around 0, which indicates that they are stable in terms of their relationship to MSE. However, the right-hand side plot shows that some outliers exist, with certain combinations yielding higher (or, to a lesser extent, lower) MSEs.

In Figure 16 for ATE estimate standard error we again see dataset as the most important predictor, followed by Q-CFR and sample size. There is a larger variance in the impact these predictors have on the outcome than for MSE. Finally, looking at Figure 17 for the $p$-values, we see heavy dependence on sample size. Note that the $p$-value outcome here is a function of a *different* sample size to the one used in the regression - it is computed based on the number of simulations which is fixed for all experiments. Thus it is still interesting to note that sample size was important as a determinant of the Shapiro-Wilk test for normality $p$-value.

Perhaps more interesting than these predictor impact results are the interaction plots, which are given in Section 7.4.

## B.2 Alternative Perspectives

In the main text we provided summary results by estimating the probability that a particular Q (outcome), G (propensity), or U (update step) method would result in a performance advantage. This was done because the number of results was large, making it difficult to judge the efficacy of a method in isolation. In Figures 18-24 we provide the complete results for each of the seven dataset variants: LF (v1) with $n = \{500, 5000, 10000\}$, LF (v2) with $n = \{500, 5000, 10000\}$, and the IHDP dataset $n = 747$. For each Figure we provide the

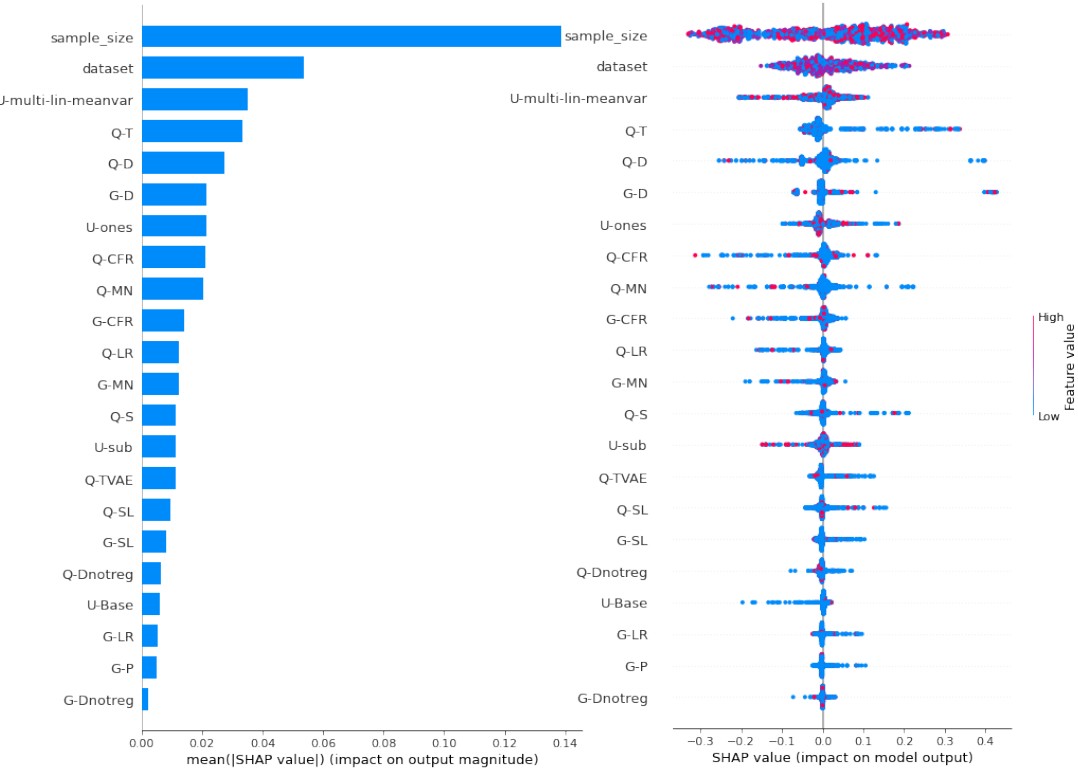

Figure 17: Shapley predictor random forest regressor impact results for the Shapiro-Wilk test $p$-values.

comparison of each Q-method with each G- and U-method, and include a red dashed line to include the base method (just the Q-method without the IF update step) for comparison.

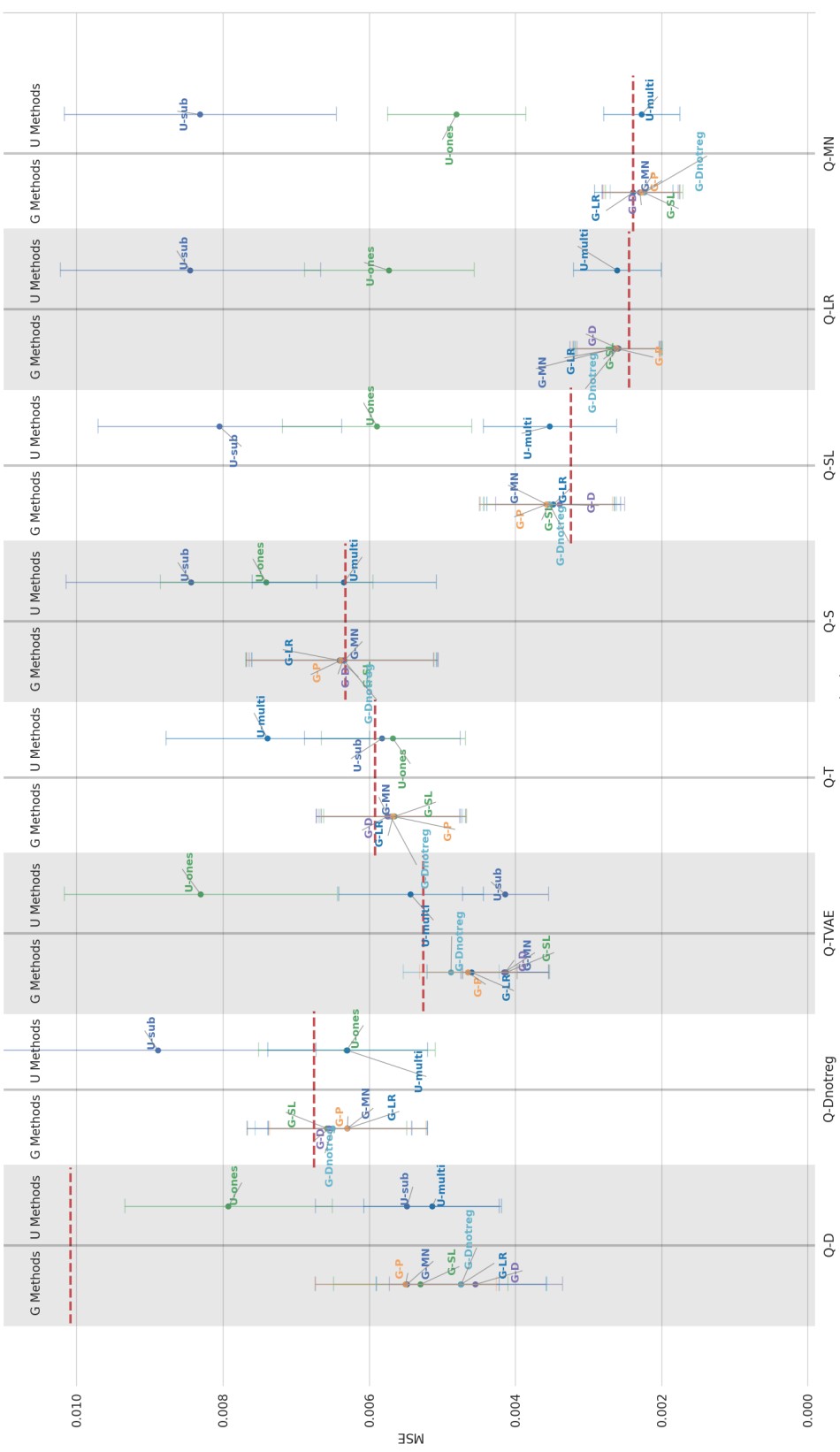

Figure 18: LF (v1) $n = 500$ results. For each outcome model $Q$ (x-axis) we plot the corresponding Mean Squared Error (y-axis) for each of the possible propensity models $G$ (left sub-column) and each of the possible update methods $U$. The base performance (no update step and therefore no $G$ or $U$) is given as a horizontal dashed red line. Because we undertook all combinations of $G$ and $U$, each point represents a marginalization over the other dimension. For instance, for Q-D (DragonNet), the 'U-ones' point is an average result for the onestep update process, using all possible propensity models G. Graph best viewed in color.

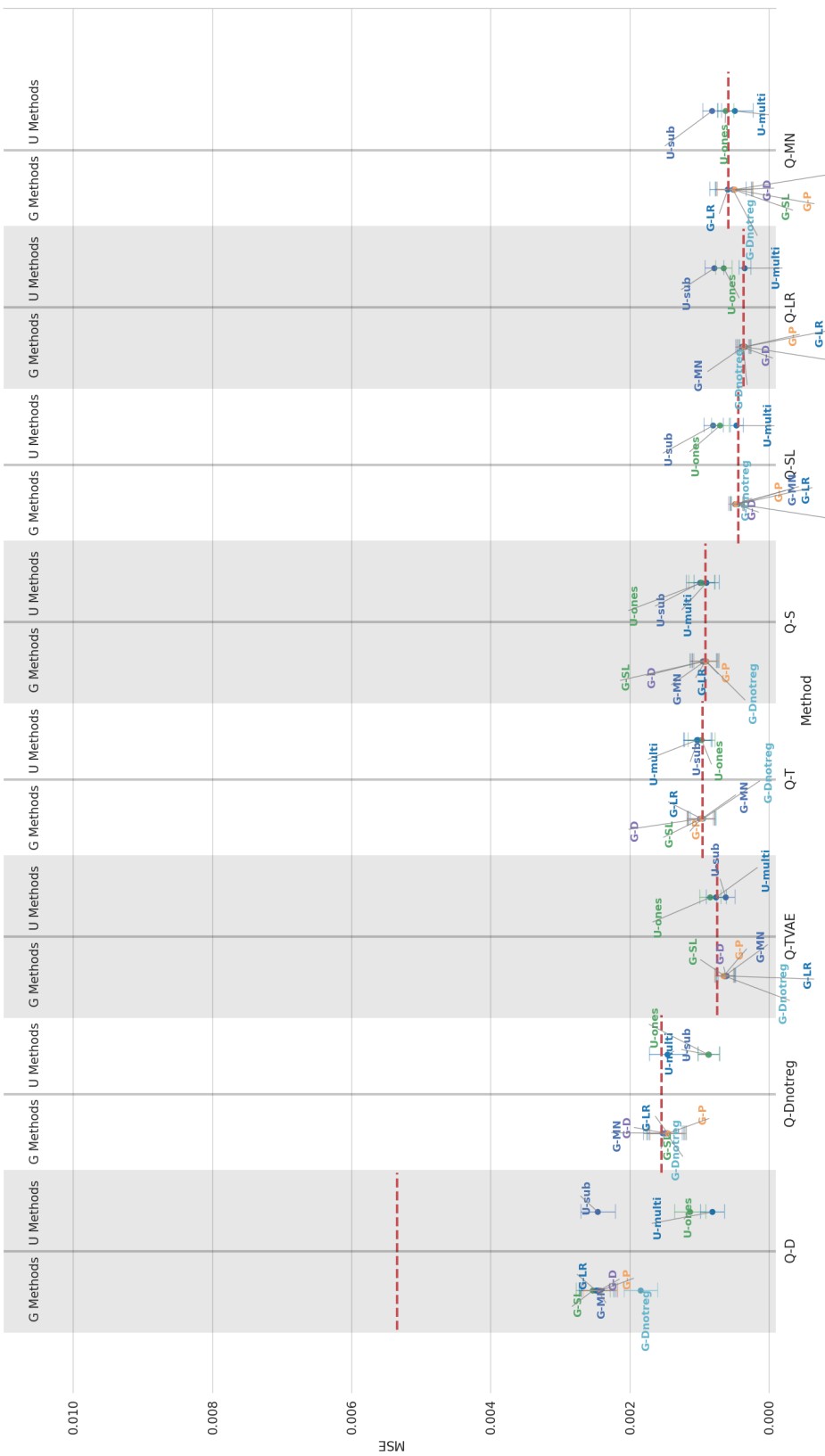

Figure 19: LF (v1) $n = 5000$ results. For each outcome model $Q$ (x-axis) we plot the corresponding Mean Squared Error (y-axis) for each of the possible propensity models $G$ (left sub-column) and each of the possible update methods $U$. The base performance (no update step and therefore no $G$ or $U$) is given as a horizontal dashed red line. Because we undertook all combinations of $G$ and $U$, each point represents a marginalization over the other dimension. For instance, for Q-D (DragonNet), the 'U-ones' point is an average result for the onestep update process, using all possible propensity models G. Graph best viewed in color.

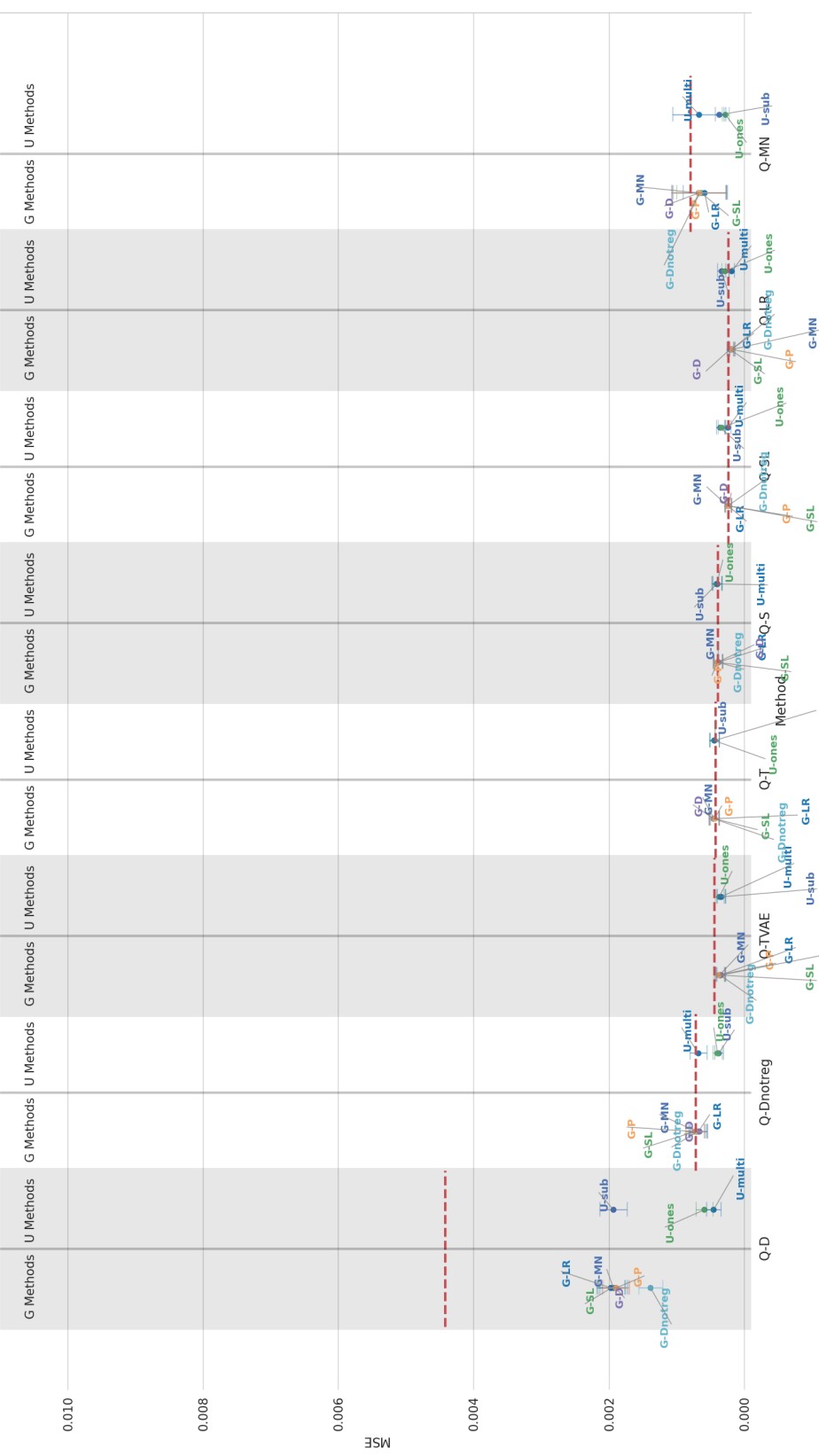

Figure 20: LF (v1) $n = 10000$ results. For each outcome model $Q$ (x-axis) we plot the corresponding Mean Squared Error (y-axis) for each of the possible propensity models $G$ (left sub-column) and each of the possible update methods $U$. The base performance (no update step and therefore no $G$ or $U$) is given as a horizontal dashed red line. Because we undertook all combinations of $G$ and $U$, each point represents a marginalization over the other dimension. For instance, for Q-D (DragonNet), the 'U-ones' point is an average result for the onestep update process, using all possible propensity models G. Graph best viewed in color.

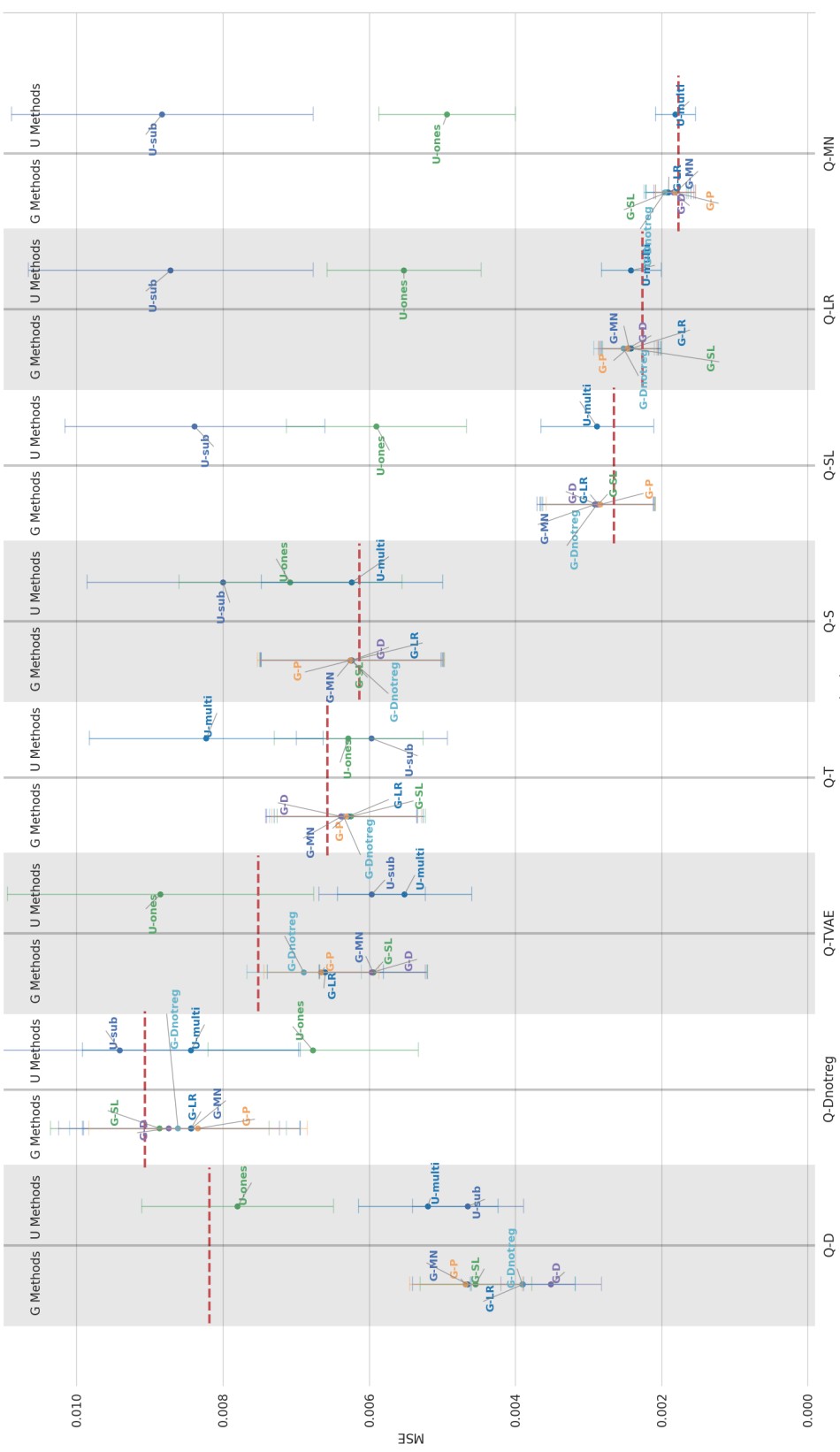

Figure 21: LF (v2) $n = 500$ results. For each outcome model $Q$ (x-axis) we plot the corresponding Mean Squared Error (y-axis) for each of the possible propensity models $G$ (left sub-column) and each of the possible update methods $U$. The base performance (no update step and therefore no $G$ or $U$) is given as a horizontal dashed red line. Because we undertook all combinations of $G$ and $U$, each point represents a marginalization over the other dimension. For instance, for Q-D (DragonNet), the 'U-ones' point is an average result for the onestep update process, using all possible propensity models G. Best viewed in color.

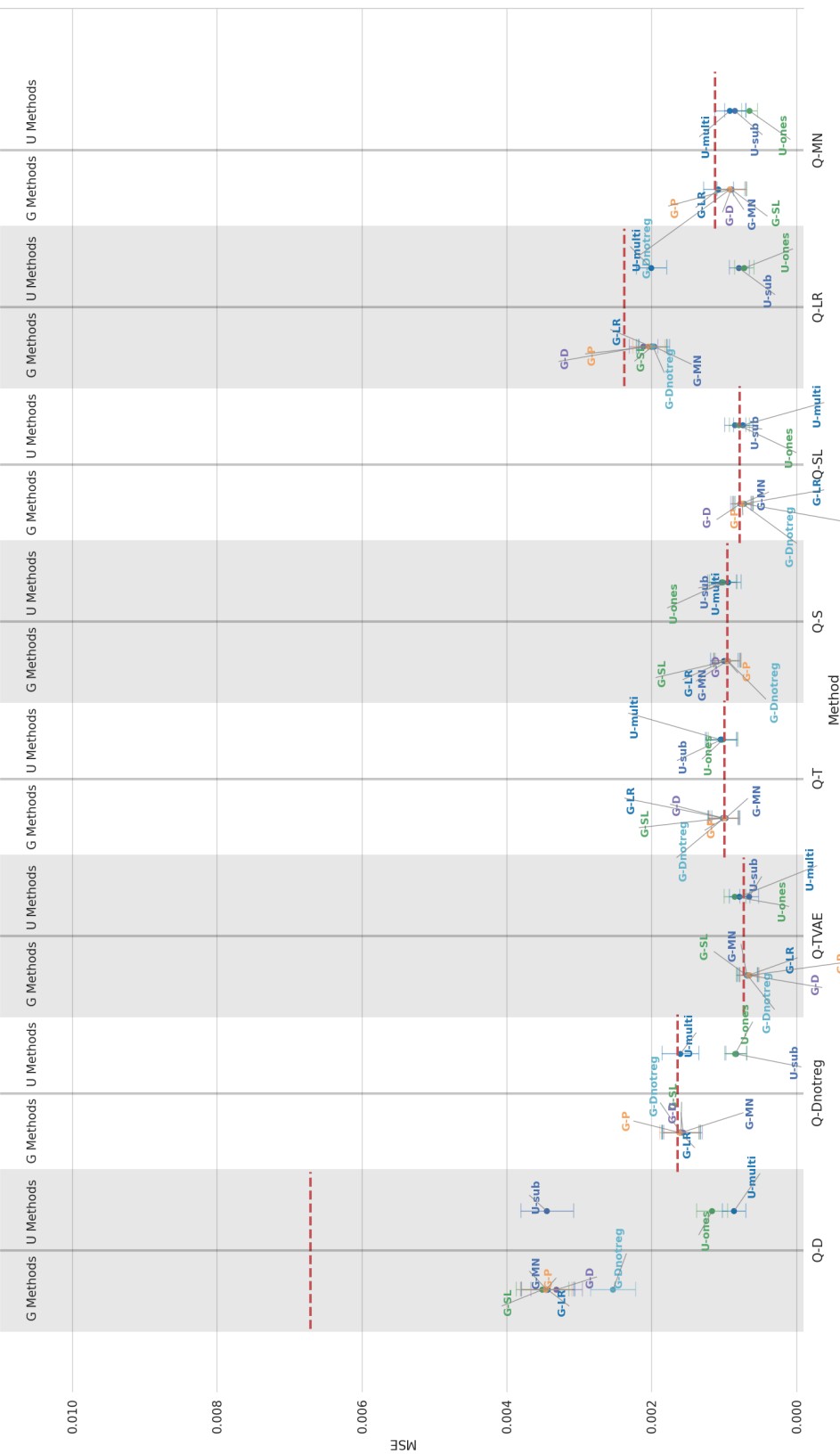

Figure 22: LF (v2) $n = 5000$ results. For each outcome model $Q$ (x-axis) we plot the corresponding Mean Squared Error (y-axis) for each of the possible propensity models $G$ (left sub-column) and each of the possible update methods $U$. The base performance (no update step and therefore no $G$ or $U$) is given as a horizontal dashed red line. Because we undertook all combinations of $G$ and $U$, each point represents a marginalization over the other dimension. For instance, for Q-D (DragonNet), the 'U-ones' point is an average result for the onestep update process, using all possible propensity models G. Graph best viewed in color.

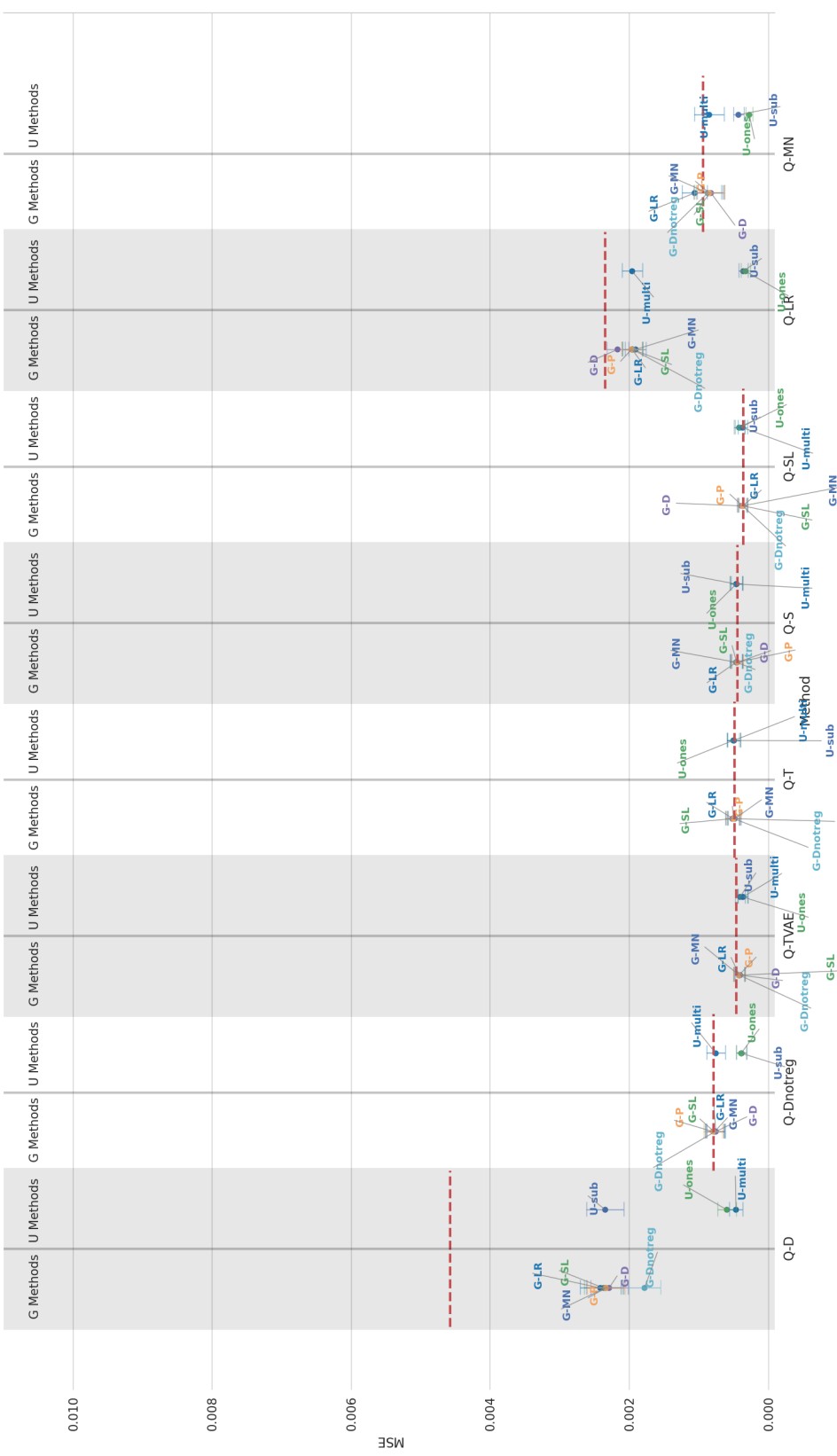

Figure 23: LF (v2) $n = 10000$ results. For each outcome model $Q$ (x-axis) we plot the corresponding Mean Squared Error (y-axis) for each of the possible propensity models $G$ (left sub-column) and each of the possible update methods $U$. The base performance (no update step and therefore no $G$ or $U$) is given as a horizontal dashed red line. Because we undertook all combinations of $G$ and $U$, each point represents a marginalization over the other dimension. For instance, for Q-D (DragonNet), the 'U-ones' point is an average result for the onestep update process, using all possible propensity models G. Best viewed in color.

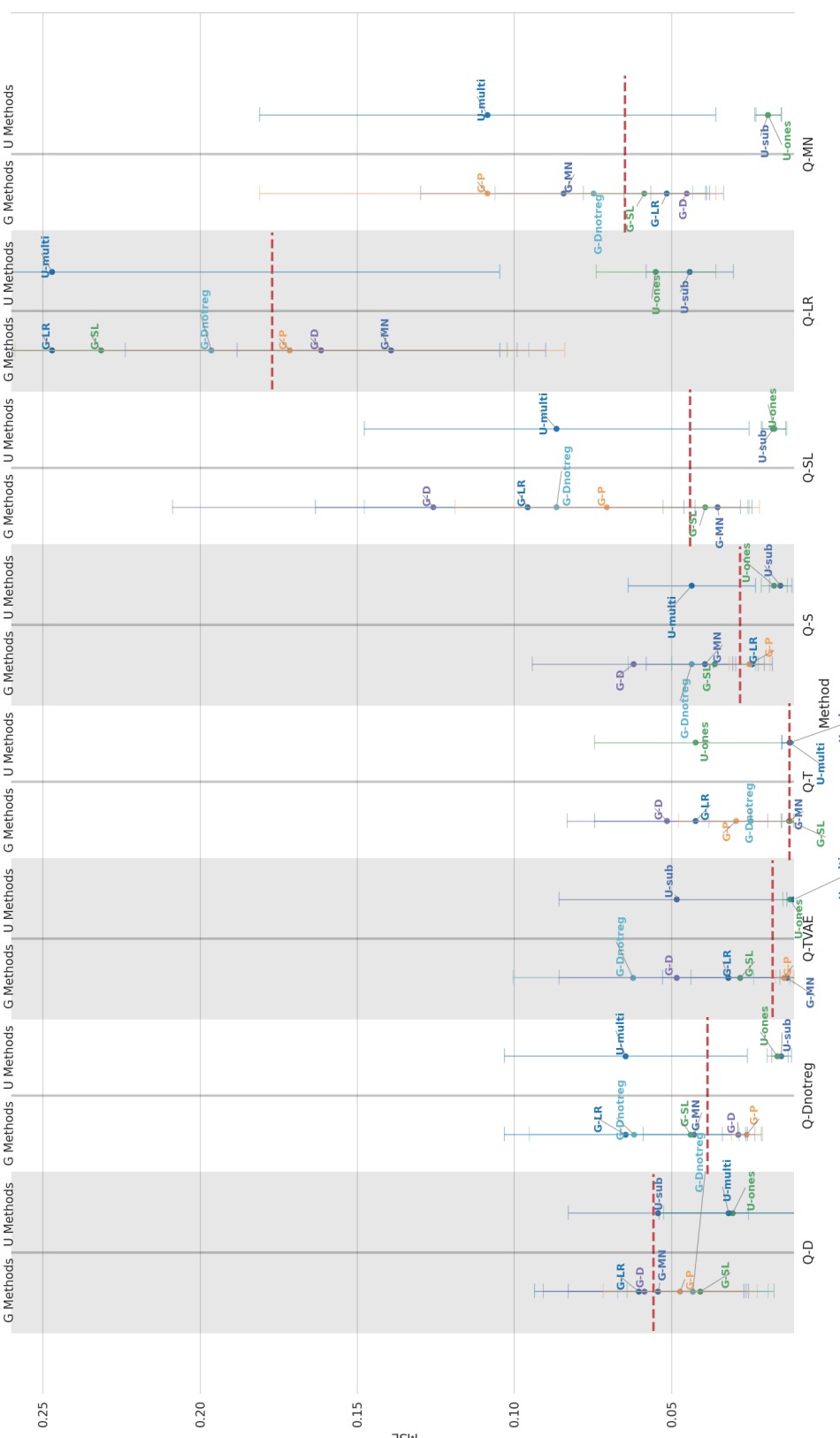

Figure 24: IHDP $n = 747$ results. For each outcome model $Q$ (x-axis) we plot the corresponding Mean Squared Error (y-axis) for each of the possible propensity models $G$ (left sub-column) and each of the possible update methods $U$. The base performance (no update step and therefore no $G$ or $U$) is given as a horizontal dashed red line. Because we undertook all combinations of $G$ and $U$, each point represents a marginalization over the other dimension. For instance, for Q-D (DragonNet), the 'U-ones' point is an average result for the onestep update process, using all possible propensity models G. Best viewed in colour.

