# OpenReview forum: "A Free Lunch with Influence Functions? Improving Neural Network Estimates of the Average Treatment Effect with Concepts from Semiparametric Statistics"
_TMLR — Rejected by TMLR_

### Review · Reviewer_eFLE · 2022-07-11

**Summary Of Contributions:**

In this work, the authors provide an overview of some key concepts that appear in the literature on semi-parametric statistical inference, with the primary focus being on parameter estimators whose error can be expressed using the empirical mean of an influence function. Strong emphasis is placed on learning tasks in the context of causal inference (e.g., estimating the average treatment effect), and the authors give detailed instructional examples within this context. As an overarching theme, the authors advocate for the use of semi-parametric statistical techniques and ideas in the vein of developing machine learning techniques which are flexible, yet are still conducive to precise statistical statements. On top of the aforementioned general exposition, the authors discuss different principled approaches to design novel estimators (their Section 4; limited to the causality context), including a new proposal ("MultiStep") which aims to minimize the mean and variance of the random IF error term. They also propose a family of learning procedures ("MultiNet") using layer-wise training on disjoint subsets of the data and aggregated predictions in an overall attempt to achieve performance similar to an aggregated ensemble of weak learners, and they report the results of a series of empirical tests using both synthetic and real-world data which evaluate their proposed procedures against a wide variety of alternatives.

**Broader Impact Concerns:**

Not applicable.

**Requested Changes:**

I think the authors need to significantly re-consider what it is they are trying to achieve in this work, and what they want to communicate to the reader in this paper.

**Strengths And Weaknesses:**

On the positive side, the overall notion of integrating semi-parametric statistical models, ideas, and techniques into the modern machine learning workflow is interesting and may be a promising research direction in the coming years. The paper itself is written in an inviting tone with generally accessible prose, the literature cited is relevant and quite comprehensive. Furthermore, the novel MultiStep procedure is described quite clearly against the backdrop of other tactics for merging IFs and causal inferential procedures. Also the detailed empirical tests have a setup that is described quite clearly, and a rather rich collection of test results that are organized in an accessible way.

On the negative side, while most individual paragraphs of this paper are fairly well-written and are of reasonable quality, at a macroscopic level, I feel that this paper is a bit of a mess. The paper contains a few conceptual "modules" as it were:

- Semi-parametric inference and influence functions
- Causal inference, in particular estimation of average treatment effect
- MultiNet, a family of learning procedures that try to mimic an ensemble effect (also combined with a causal inference sub-routine)

In the current manuscript, these modules are not very cohesive, making it very hard to evaluate what this paper is about. The title and abstract say nothing about causal inference, and yet the most concrete parts of this paper are completely specialized to the causality context. I get the heuristic in the MultiStep idea, but the MultiNet proposal seems to come out of nowhere.

There are also numerous small details that the reader can trip up on. Here are a few examples:

- After equation (7), the authors say that an "unbiased" estimator is one whose estimation error "converges in probability to zero," but I think most readers will refer to such a property as _consistency_. In several places throughout the paper, I find the use of the term "bias" highly unnatural.
- In equation (32), the authors characterize their MultiStep idea in terms of minimizing a weighted sum of the mean and variance of the random IF value; can the minimizer always be said to satisfy (33) and (34)? Why do $\alpha_{1}$ and $\alpha_{2}$ not appear in these two formulas?
- The description of the "submodel" approach in 4.1 is critical to the derivation of MultiStep in 4.2, but I find it extremely hard to follow. The nature of $H$, and the relation of (31) to (22) is very unclear, and as a result, the conceptual basis upon which MultiStep is built is quite weak.

Overall, I cannot really parse this paper to extract any new knowledge or insights. Is this a survey paper on IF-based techniques in ATE estimation? Is this a general introduction to semi-parametric methods and their utility in machine learning? Is this a paper studying neural network learning algorithms using heuristics to mimic ensemble methods? The overall story is diluted by a large number of concepts which each play only a very minor role. The whole idea of a "free lunch" is great if we know the influence function, but since we do not, and a lot of effort and uncertainty arises from trying to put together a reasonable approximation of this error term, the net gain for the learning algorithm designer/user using the approach advocated by the authors is not really evident, to me at least.

---

> ### Author Response · Authors · 2022-07-26
> **Initial response to reviewer eFLE**
>
> We thank the reviewer for their valuable time and feedback – it is very gratefully received. Below, we respond to each of the noted weaknesses.
>
>
> 1. Re: ‘modules are not very cohesive’, and ‘what it is they are trying to achieve in this work’. \
> Indeed, when we submitted this paper, we were aware that the work has a broad scope including a tutorial element, a brief survey, and a new neural network / update proposal with detailed evaluation.
>
>
>     In the past we have fallen foul from not including enough background - semiparametric theory is not a topic which is widely familiar to researchers, particularly those principally working the domain of deep learning. To this extent, a gentle introduction could be worthwhile (and we thank the reviewer for noting the accessibility of the writing), and also reviewers GzoT and DiMQ note its potential to be useful to other researchers in ML. We had actually intended to highlight our openness to having a discussion about the focus of the paper in a cover letter, but could not find a way to do this on the OpenReview platform. This is, to some extent, borne out by the three reviewer opinions where you highlighted the potential redundancy of the tutorial element whereas the other two seemed to appreciate this aspect. We are very open to discussing this with the reviewers, and would appreciate any guidance or advice they have regarding the direction they feel the paper should take / focus on.\
>
>     Removing the more tutorial like elements may be one way to focus the paper, but again defer to any advice the reviewer(s) may be able to share in this regard.  If we were to restructure and reduce the tutorial then we would suggest the following structure:
> - Shorter background sections, focussing on causal inference as the application area (and of course highlighting this in the abstract)
> - Proposal of MultiNet and MultiStep
> - Evaluations
>
>
> 2. We agree with the reviewer that this is not currently well motivated. The heuristic for the MultiNet proposal will be improved accordingly. Indeed, the method represents a combination and extension of two ideas from two very popular methods from the causal inference literature (one from the deep learning side - CFR-net - and one from the applied statistics side - SuperLearner) and is well suited to the application domain of causal inference, and we believe this can be made clearer.
> 3. We will also review the usage of ‘bias’ and agree that this certainly needs to be adjusted.
> 4. We will clarify the submodel approach to also strengthen the conceptual basis for MultiStep

---

> > ### Comment · Reviewer_eFLE · 2022-07-29
> > **Re: Initial response to reviewer eFLE**
> >
> > I thank the authors for their response.
> >
> > > We are very open to discussing this with the reviewers, and would appreciate any guidance or advice they have regarding the direction they feel the paper should take/focus on.
> >
> > This is not really something I can answer; it is up to the authors to determine what they can and want to communicate to the community. Personally, the more I read this paper, the more confused I get about the overarching goals of this paper. Let me just mention a couple more points.
> >
> > As I mentioned in my initial review, I think the notion of a "free lunch" is pushed on the reader with minimal evidence. Yes, of course, if we have complete information about the influence function, then we can make a convenient "correction" to estimators based on more naive assumptions, but since this function is unknown, the IF approach always requires some additional learning task (e.g., the propensity score model), and unless I'm missing something, traditional learning theory guarantees would require a fresh set of data, since each summand in the IF term depends on the empirical distribution induced by the whole initial sample. In the end, it's just a game of assumptions. In some cases, a non-linear correction will help, and other times it will not help - this is self-evident, and is just reinforced by the experimental results. If one can find some patterns in the successful cases and draw more general conclusions about how and when to use the IF-based tactics, then that is a valuable empirical contribution, but it is far from what I would call a "free lunch."
> >
> > Another point is that the notion of "statistical inference" is extremely vague here. As motivation (cf. abstract), non-parametric methods are said to be flexible but not always convenient in terms of "statistical inference," and IF-driven semi-parametric approaches a possible middle ground with enough flexibility that can perhaps "(c) facilitate statistical inference," yet I seriously wonder whether any evidence has been provided for this in the paper. See 3.2.4 ("Statistical Inference with Influence Functions"), where the authors show that we get statistics with a convenient Normal distribution, however, this is "assuming a normal distribution" from the outset. I think readers passing through this section for the first time will be rather puzzled; what assumptions have been weakened? What if we don't want to assume parametric distributions at all, and just leverage a central limit theorem; is that unsatisfactory? Maybe my reading was too casual, but in my opinion the benefits of the proposed approach in terms of aiding some ancillary "statistical inference" (whatever that means) in the context of some machine learning task are unclear.
> >
> > My personal opinion is that the composition of this paper needs some significant work, but since the other reviewers have provided lots of concrete suggestions for improvement, please feel free to follow their advice in moving forward.

---

> > > ### Author Response · Authors · 2022-08-01
> > > **Response to reviewer eFLE (part 2)**
> > >
> > > The reviewer makes some good and fair points regarding the use of the term 'free lunch'. Indeed, the update step requires the training of an additional propensity 'G' learner. Furthermore, our results indicate mixed efficacy, as the reviewer indicates when they say: 'In some cases, a non-linear correction will help, and other times it will not help - this is self-evident, and is just reinforced by the experimental results.'  As such, we propose to soften the advertisement for influence function based methods, specifically, the argument that they might provide us with a 'free lunch'.
> > >
> > > This notwithstanding, and having read the other reviews, we would argue that an exploration and empirical/experimental evaluation of Q-, G- and U-methods (such as the one provided in our work) in the context of causal inference is valuable in understanding their potential and general applicability. Of course, identifying patterns in the successful cases would be ideal, but in our view it is also valuable to understand whether the performance of today's approaches can be consistently relied upon (somewhat akin to the argument for the importance of publishing 'null' findings). This is especially important to understand in the domain of causal inference, where we have no 'ground truth' in real-world application settings. Previous work by Curth et al. (2021b) found similar evidence of dataset dependence, and this in part inspired our experimental methodology and presentation of results. Our results corroborate their findings on a different set of estimators, as well as demonstrating Q-, G-, and U-method have a <combination> dependence. In summary, we would agree that the 'free lunch' element can be softened in the paper (and will make changes accordingly), but we would also counter by reasoning that our broad evaluation of performance and the identification of mixed-performance is still valuable to the community,
> > >
> > > Regarding the point about the assumption of the normal distribution - one core task in most empirical research in the human/medical sciences is that of null hypothesis significance testing (NHST). If our parameter estimates are normally distributed they are directly amenable to such testing (of course, other methods exist, such as the jack-knife approach [Efron 1981] or bootstrapping). Accordingly, in Figure 10 we provide probabilities of achieving $p < 0.01$ for the Shapiro-Wilk test for normality. NHST (as a common form of statistical inference) is one of the principal motivators for IF based Targeted Learning in the literature, and parameter estimates derived from flexible (non-parametric) machine learning estimators are not directly amenable to such tests. We would therefore argue that the motivation for the development and evaluation of influence-function based methods for statistical inference is already well-motivated in the literature - we can, however, clarify this motivation in our paper, and more clearly explain what we mean by 'statistical inference' in this context.
> > >
> > > In terms of the reviewer's comment regarding central limit theorem - we are not 100% sure we understand this point and would be grateful for clarification. On this note, we recall the application of central limit theorem already implicit in the influence function mechanics (and the assumption of asymptotic linearity/normality). The strength of these assumptions, we would argue, is relatively weak, and does not imply e.g. that the causal effect is normally distributed across the population, only that the average causal effect (i.e. the parameter representing this average quantity, as a potentially non-linear, non-parametric function of a sample from the population) is normally distributed across <different samples from the population>. This does not impact the functional form we use to describe the data generating process itself (which can be highly non-linear and non-parametric), only the parameter, which can be expressed as a function of realisations from this process. Perhaps this was already clear to the reviewer, but will nonetheless certainly clarify these points in the paper.
> > >
> > > Finally, as the reviewer has not expressed a strong opinion regarding the removal of the tutorial element (and considering the other reviewers' comments), we will leave it in the paper and make its purpose, and indeed the objectives of the paper as a whole, clearer.

---

### Review · Reviewer_GzoT · 2022-07-15

**Summary Of Contributions:**

This paper considers the application of semi-parametric methods to machine learning, in particular to the estimation of treatment effects using neural networks. The authors start from influence functions, which describe the asymptotic distributional behavior of estimators, show how they can be used to debias and obtain confidence intervals, and propose a new method for debiasing. They conduct an experimental study of the estimation accuracy and asymptotic normality of various methods, when combined with various estimators for the average treatment effect (ATE) and the propensity model (including those based on NNs). They also propose MultiNet, a version of the SuperLearner with a CFR network backbone, for ATE and propensity estimation. The empirical results are dataset dependent. For the synthetic dataset (where the ground truth is known), the MultiNet works well for estimating the ATE, but are outperformed by others in terms of asymptotic normality. The same pattern holds for the new method for debiasing and no debiasing at all.

**Broader Impact Concerns:**

No broader impact statement is given. However, since a large part of the paper discusses evaluating methods for ATE (commonly used for medical and public policy applications where the stakes are high), I think a discussion of the appropriateness of the metrics they use for real-world applications should be included. For example, why is the MSE an appropriate metric? Is it because larger errors are penalized more?

**Requested Changes:**

In order for me to recommend acceptance, I would like the authors to (corresponding to the two weaknesses discussed above):
1. Include analysis on other, preferably real-world, datasets, or explain why none are available. I do not work in the causal analysis field, so I am not sure which are appropriate. Perhaps Twins and Jobs mentioned in [A] are options?
2. Include an analysis of part of the factorial design mentioned at the beginning of section 7, since analyzing the full design would be difficult. Specifically, select a subset of the ATE estimators, a subset of the propensity models, and a subset of the debiasers, and show the full factorial design on them. The subsets could be selected based on the current analysis, their representativeness, or some other criterion.

Things I would like the authors to clarify:
1. What was the reasoning behind the choosing the baselines described in section 6.3.1?
2. How was the MSE computed?
3. How is the multistep debiasing process proposed by Neugebauer & van der Laan different from the proposed method?

Minor comments on writing, to strengthen the work:
1. The notation in equation 16 is unclear. Some subscripts seem to be missing.
2. On page 16, what is the loss used to train the constrained regression?
3. It may be better to place figure 3 earlier in the text, since it describes the data analysis procedure of interest. Currently the jump from IFs/debiasing to MultiNet is not very well connected.

[A] Hatt, Tobias, and Stefan Feuerriegel. "Estimating average treatment effects via orthogonal regularization." Proceedings of the 30th ACM International Conference on Information & Knowledge Management. 2021.

**Strengths And Weaknesses:**

# Strengths #
1. The authors provide a high-level but mathematically rigorous exposition of influence functions and how they can be used to debias estimators, which I think would be of interest to the machine learning community. They also give generic instructions for computing IFs for general estimators.
2. Experiments are done on a wide range of combinations of algorithms for estimating the ATE, estimating the propensity model, and debiasing. The authors consider up to second order effects (i.e. marginalizing over only one of these three variables).
3. The authors provide conjectures on why certain algorithms work well on certain datasets.
These indicate that the results in this paper would be of interest to practitioners in machine learning.

# Weaknesses #

1. The datasets used in the empirical evaluation have some limitations. LFv1 and LFv2 are synthetic. IHDP contains real world data, but the authors ask us to discount the findings.
2. The results suggest that a finer-grained analysis may be necessary. For example, table 3 suggests that debiasing can be quite effective, while later figures (which marginalize away the effect of the propensity model) suggest that no debiasing is comparable to debiasing. In my opinion, this behavior indicates that the interaction of the ATE estimator, the propensity model, and the debiaser may be quite important.

---

> ### Author Response · Authors · 2022-07-26
> **Initial response to reviewer GzoT**
>
> We thank the reviewer for taking their time to review our work and provide constructive feedback – it is very much appreciated. Below, we respond to the noted weaknesses and suggestions.
>
>
> ### Requested Changes
>
>
> 1. Re: the use of synthetic datasets – whilst the use of Twins and Jobs has been common for benchmarking causal methods in machine learning, Curth, Svensson, Weatherall, & van der Schaar (2021) note that these benchmarks (including the IHDP dataset, we which use in our evaluations) are not reliable for demonstrating and testing estimators, hence why the interpretation is discounted. Specifically, they note that: “authors should be incentivized to demonstrate how the performance of their proposed algorithm changes as experimental knobs, capturing the relevant dimension, change from most to least favourable settings. Requiring the use of ‘standard’ benchmark datasets can be detrimental in this context; especially when there are no relevant experimental knobs or settings.” This is the reason we use the LF (v1) and (v2) datasets, where we know their characteristics a priori (which vary in terms of sample size and positivity). We can discuss this more in the paper, and note that the use of Jobs and Twins would, unfortunately, be equally ‘discounted’. If the reviewer still feels that further experiments on e.g. the Jobs dataset are necessary, we would be happy to conduct and share them, although tentatively suggest they be added to the appendix given the factorial nature of the experiments section.
> 2. Re: for the full factorial design – please could the reviewer clarify what they mean here? In Table 3, we already select a subset of the full factorial design to reduce the search space to only ‘competitive’ approaches. Otherwise, we also provide the complete ‘unmarginalized’ results for the subset of methods we identified to be competitive in the appendix (Figures 12-18).
>
>
> ### Clarifications:
>
>
>
> 1. Re: Choosing baselines. These baselines were chosen based on a tradeoff between the number of possible comparisons, and the methods which are widely used in application (i.e. Logistic regression, Super Learners and the T-, P-, and S- learners) and widely used as comparison methods in machine learning, or relevant to semi-parametric methods (i.e. CFR, DragonNet, TVAE). We can clarify this in the paper.
>
> 2. Re: how MSE was computed. MSE was computed across simulations, i.e. for $K$ simulations, and where $\\hat{\\tau}_k$ is the kth estimate for the true ATE $\\tau$, the MSE is calculated as:
>
>
> $MSE=\\frac{1}{K}\\sum_{k=1}^K(\\hat{\\tau}_k - \\tau)^2$
>
> We will include this information in the paper.
>
> 3. Re: the Neugebauer & van der Laan multistep approach – we were actually unable to confirm the details of this method from the paper and available materials online. We were also unsuccessful in contacting the authors for more details and implementation. As far as we can tell, their update step is different insofar as they use an ‘iterative Newton-Raphson algorithm with linear search’, and they update to solve for when the mean empirical estimate of the influence function is equal to zero, but not to minimize the variance. Unfortunately, this is the extent to which we have been able to glean details about their method.
>
>
> ### Minor comments
>
>
> 1. Re: missing subscripts. Unfortunately, we are not sure which subscripts are missing here. If the reviewer is able to clarify, we would be happy to fix it.
> 2. Re: constrained regression loss. The loss for the regression is least squares –this is mentioned in paragraph 5 on page 16 (‘non-negative least squares solver’).
> 3. Re: figure 3 move. This is a good suggestion, we will also tie the presentation of MultiNet more strongly to the preceding sections in the paper.
>
>
> ### Broader Impact
> We agree this is important to discuss, and whilst we included a statement of impact in the last paragraph of the paper, we can dedicate a subheading to it and expand upon it. We also thank the reviewer for the idea to discuss here the MSE metric. Indeed, the RMSE is used in other evaluations (see e.g. Curth & van der Schaar 2021), but the MSE was found to yield a more informative spread across results (rather than some being heavily clustered and difficult to compare) and, as the reviewer explains, draws attention to larger errors.
>
>
> ### Lastly...
> Finally, we note that one of the other reviewers has highlighted that the paper has multiple facets and might instead benefit from a more streamlined structure. We would be very grateful if the reviewer would be open to engaging in a discussion about whether the structure and content of paper should be changed accordingly. Indeed, some readers might feel the background elements of the paper are valuable, particularly given the subject matter, whereas perhaps others would feel it to be redundant. We thank the reviewer in advance for their engagement with this.

---

> > ### Comment · Reviewer_GzoT · 2022-07-28
> > **Thank you for the response!**
> >
> > Here are some further clarifications to my points:
> > - RE Requested Changes #2: In figures 12-18, for each point one dimension is being marginalized over, which removes the third-order interactions between Q, G, and U. It seems that this third-order interaction may be quite important (table 3 indicates that debiasing with U-ones is helpful in LFv1, but this is not clear in the right three columns of figure 13 where the model Q is the same as in the table). Therefore, my suggestion was to remove the marginalization, but only for an even smaller subset of Q, G, and U methods for tractability.
> > - In equation 16, should $f$ be replaced with $f_\epsilon$?

---

> > > ### Author Response · Authors · 2022-07-31
> > > **Clarifications well understood**
> > >
> > > Many thanks to the reviewer for their patience and time clarifying these two points - in both cases we agree. We will replace the missing subscripts, and add a table of additional results.
> > >
> > > In terms of the choice of methods to include in the factorial comparison, we suggest using the following on the basis that (a) the Q-MultiNet and U-MultiStep are proposed in the paper and therefore ought to be compared, (b) Q- and G-SuperLearner is popular in the biostatistics literature (as both a Q and a G method), (c) Q-TVAE is a neural network method and performed competitively across the evaluations, (d) the U-submodel 'targeted' updated step is well-discussed in the biostatistics literature, and (e) the Q-T-learner is the more complex/sophisticated of the S- and T-learner methods proposed by Kunzel et al. 2019 (although this choice is somewhat arbitrary, because the S-learner seems to perform well in complementary settings).
> > >
> > > ### Q methods:
> > > - MultiNet
> > > - Super Learner
> > > - TVAE
> > > - T-learner
> > >
> > > ### G methods:
> > > - Super Learner
> > > - MultiNet
> > >
> > > ### U methods:
> > > - MultiStep
> > > - Submodel
> > >
> > > This yields 4x2x2 = 16 results.
> > >
> > > We also also very open to other suggestions for inclusion/exclusion from the reviewer.

---

### Review · Reviewer_DiMQ · 2022-07-23

**Summary Of Contributions:**

The authors revisit causal inference, through the lenses of deep learning and semiparametrics. Their goal is to study the relevance of deep neural nets as functional estimates withing a causal DAG. They argue that these neural nets lead to biased estimates for the average treatment effect (ATE), and explore various ways of debiasing them, using ideas from semiparametrics.

They conduct a thorough simulation/ablation study, where they compare different approaches to causal inference, based or not on deep nets, and potentially debiased. They also add a new method to the list, called MultiNet/MultiStep, which is based on two innovations: an ensemble-like architecture (where each element of the ensemble is the output of a layer), and a new loss function, based on the layer-wise ensembling idea.

**Broader Impact Concerns:**

I have no particular concern about this work.

**Requested Changes:**

I think the following minor changes are quite important:

1) Writing the claims I mentioned in the "Weaknesses" section in a less misleading way.

2) Discussing other approaches to neural ensembling, and relating yours to the others. In particular, since you use dropout, do you use the stochasticity for ensembling as well?

3) I think the relationship between your paper and Farrell et al. (2021) should be discussed a bit more. While I am not very familiar with this paper, it makes it look like just plugging in neural nets will lead to good inferences, which somewhat contradicts a bit your story.

Another thing that would clearly strengthen the paper, but is much less straightforward, is to make the simulation study more clear. Unfortunately, I have no good recipe for this, but here are a few suggestions:
- the quantiles curves are, I think, a good idea, but are not very easy to understand, it would be helpful to write down clearly what would a "ideal curve" would look like.
- adding some statistical tests would be helpful to assess how much of the results are significant. For instance, in the very large-scale study of Fernandez-Delgado et al. (2014), they used Friedman's ranks to summarise the results.


Additional reference

Fernandez-Delgado et al., Do we Need Hundreds of Classifiers to Solve Real World Classification Problems?, JMLR 2014


**Strengths And Weaknesses:**

Strengths

- The paper is very well-written, and contains a well-narrated review of causal inference and deep learning, I believe the review itself is a quite nice contribution
- The new method called MultiNet/Multistep is a sensible approach that seems to work well
- The simulation study seems quite valuable, albeit difficult to read, it allows both to assess the usefulness of recent deep learning-based ideas for causal inference, as well as the new contribution of the authors (MultiNet/Multistep)


Weaknesses

- Some claims in the abstract/introduction are phrased in a misleading way. For instance, the last sentence of the abstract "We also show that it is possible to improve existing neural networks for ‘free’, without needing more data, and without needing to retrain them" is indeed consistent with the experiments (which show, that, sometimes, updating works better, but it also sometimes does not), but suggests that updating is generally preferable, which is not that clear from the experiments. I think this sentence could be weakened, or additional details on the marginal or negative value of updating should be mentioned. Similarly, in the introduction, the authors say "we confirm that initial estimation methods benefit from the application of the semiparametric techniques. In general, however, the improvements are dataset dependent, highlighting possible interactions between the underlying data generating process, sample sizes, and the estimators and update steps used." Again, this is technically accurate, but slightly misleading, since I belive it make is sound like the updates always improve upon the original techniques, which is not the case.

- The MultiNet architecture seems a bit complex, and not motivated well enough. Why not use more standard ensembles of neural nets, for instance MC dropout (Gal & Ghahramani, 2016), or deep/batch ensembles (Lakshminarayanan et al., 2017, Wen et al., 2020) ?

- The simulation study is a bit hard to read. The experiments are thorough indeed, which makes presenting them very difficult, and I appreciate the authors's efforts in that direction, although I had a hard time understanding it (notably the quantile curves). I have put a few suggestions in the "requested changes" section.


Minor remarks

- I do not fully agree with "most machine learning models are non-parametric and do not readily facilitate statistical inference" (page 1). Indeed, many popular ML models are parametric (albeit sometimes overparametrised), including deep nets, logistic regression
- I do not understand "notwithstanding finite sample associations" page 5
- On the relationship between semiparametrics and deep learning, relevant papers were recently authored by Zhong et al. (2021, 2022)


Some minor references issues

- Farrell et al. was published in Econometrica in 2021
- the book by Bickel et al. on semiparametrics is from 1998, not 2007
- Shalit et al. was published at ICML 2017


Additional references

Wen et al., BatchEnsemble: An Alternative Approach to Efficient Ensemble and Lifelong Learning, ICLR 2020
Gal and Ghahramani, Dropout As a Bayesian Approximation: Representing Model Uncertainty in Deep Learning, ICML 2016
Lakshminarayanan et al. Simple and scalable predictive uncertainty estimation using deep ensembles, NeurIPS 2017
Zhong et al., Neural Networks for Partially Linear Quantile Regression, Annals of Statistics, 2022
Zhong et al., Neural Networks for Partially Linear Quantile Regression, arXiv:2106.06225, 2021

---

> ### Author Response · Authors · 2022-07-26
> **Initial response to reviewer DiMQ**
>
> We thank the reviewer for their valuable time reviewing and providing constructive feedback – it is very gratefully received. Below, we respond to each of the noted weaknesses and remarks in turn.
>
> ### Under Weaknesses
> 1. Re: misleading claims. We agree with the reviewer and will adjust the claims accordingly.
> 2. Re: the complexity of the MultiNet architecture. We will motivate this more strongly in the paper. In summary, the networks is motivated by combining ideas from two well-known methods in the domains of causal inference: CounterFactualRegression/CFR-Net (Shalit et al. 2017), and SuperLearner (van der Laan et al. 2007).  The complexity is actually not much greater than that of CFR-net, which provides the backbone for the proposal. Indeed, rather than taking a single output from the last layer, we take intermediate outputs (which therefore represent different functions of the input which increase in complexity with the layer depth) and then combine these using the constrained regression. Of course, we explore variants of this idea which include the use of disjoint subsets of the data etc.
> 3. Re: complexity of the experiments. We thank the reviewer for acknowledging the associated difficulty of summarising the numerous results from the experiments, and are grateful for their suggestions to improve the readability. We will return to this point below.
>
> ### Under Minor Remarks
> 1. Re: parametric. There is a chance we are using the term parametric in a slightly different way. We agree that neural networks (and other ML algorithms) have parameters per se, but in general our use of the word parametric pertains to the model assumed to describe the distribution we are estimating. For instance, if we are modelling p(y|x), we have a choice to parameterise our model of this conditional distribution using, e.g., a Gaussian with two parameters (location and scale). In our usage, this would make the model parametric. In contrast, many ML approaches do not assume a parametric model for the distribution they are trying to model, i.e. p(y|x) is non-parametric insofar as it has no 'simple' analytical representation. We can clarify this in the introduction to make the usage of the term clearer.
> 2. Re: finite sample associations. By this we mean that in ‘finite samples’, i.e. a sample from the population with a finite number of datapoints (e.g. n=500), statistical associations may exist which are a consequence simply of random sample variation. We can clarify this to make this clearer.
> 3. Re: additional references. These are very helpful references – thank you. We will include them in our literature review.
>
> Re: the reference issues – thank you for highlighting these, we will certainly amend the reference list accordingly.
>
> ### Under Requested Changes
>
> 1. Re: adjusting weaknesses. We agree and will modify this.
> 2. Re: discuss other neural ensembling methods. We agree and will modify this. Regarding the question about stochasticity – we are not sure we completely understand the question, but if the reviewer is referring to the use of dropout to generate multiple outcome predictions then no, this is not explored as a technique in the paper.
> 3. Re: discussing Farrell et al. (2021). We agree, this would be a good addition to the literature and discussion sections and will add it.
> 4. Re: making the simulations clearer: the ‘ideal curve’ is a great idea to make the plots more easily interpretable, and will do so. For the Friedman’s rank test, we thank the reviewer for this great reference and suggestion, and whilst this method would be useful for ranking all experiments, based on our understanding it is not clear how this method would be adapted for combining and evaluating the G-, Q- and U- methods (i.e. marginalizing over the different components of each method). If the reviewer would be satisfied with a modification of the paper to clearly describe the ‘idea curve’ for the results, we would be inclined to use this suggestion.
>
>
> Finally, we note that one of the other reviewers has highlighted that the paper has multiple facets and might instead benefit from a more streamlined structure. We would be very grateful if the reviewer would be open to engaging in a discussion about whether the structure and content of paper should be changed accordingly. Indeed, some readers might feel the background elements of the paper are valuable, particularly given the subject matter, whereas perhaps others would feel it to be redundant. We thank the reviewer in advance for their engagement with this.

---

> > ### Comment · Reviewer_DiMQ · 2022-08-05
> > **Thanks**
> >
> > Thanks for your response, rebuttal, and revision! I notably appreciate the revised conclusion, and the note on the ideal curve.
> >
> > I am still reading the revised version, and I have a question: what is exactly the goal of the shapley values analysis? I am afraid I really do not get it... In what sense is it a meat-analysis?
> >
> > > Finally, we note that one of the other reviewers has highlighted that the paper has multiple facets and might instead benefit from a more streamlined structure. We would be very grateful if the reviewer would be open to engaging in a discussion about whether the structure and content of paper should be changed accordingly. Indeed, some readers might feel the background elements of the paper are valuable, particularly given the subject matter, whereas perhaps others would feel it to be redundant. We thank the reviewer in advance for their engagement with this.
> >
> > After giving it some thought, I do agree with Reviewer eFLE that the paper is "at a macroscopic level (...) a bit of a mess" even though I do not think this is a huge issue, and I believe some of the clarifications that you made (in particular in the abstract/introduction/conclusion).

---

> > > ### Author Response · Authors · 2022-08-05
> > > **Response pt. 2**
> > >
> > > Thanks for taking the time to read the updated version - your continued engagement is greatly appreciated!  Many thanks, also, for your feedback so far.
> > >
> > > For the Shapley analysis, we are happy to avoid calling it a 'meta' analysis - perhaps this unnecessarily suggests an analysis of other prior research. That said, the Shapley approach represents a second level of analysis on top of the results from the first (an analysis of the results of an analysis), so to this extent it is 'meta'.
> > >
> > > Perhaps we can clarify the motivation for this secondary analysis as follows - we have a lot of results from a full-factorial set of experiments, and this makes it challenging to identify patterns between the choice of method and the corresponding performance. e.g. it is difficult to say that using Q-TVAE + G-SL + U-sub is a good combination in terms of MSE, normality, and/or standard error  across the different datasets and sample sizes. Indeed, and as reviewer GzoT, such interactions may be lost using only the prior presentation of results.
> > >
> > > With the Shapley approach, we take the results from each combination of methods and each of the datasets and sample sizes, and use these as predictors of the outcomes (MSE, standard error, and p-values) in a regression task. Under the assumption that the regressor will leverage patterns linking the combination of methods and the corresponding performance, we can interrogate this regressor using SHAP. The results tell us (a) which predictors are most important (and we have included these results in the appendix), and (b) whether there exist interactions between the choice of methods. The results currently presented highlight that many methods are stable across e.g. sample size and dataset, but one or two interact strongly (e.g. Q-CFR).  We hope this helps explain the methodology, and we can certainly clarify this more in the paper.

---

### Author Response · Authors · 2022-08-02
**Paper Updated**

Dear all,

We usually receive email updates from new comments made on OpenReview, but we did not receive any for our list of changes and updated submission above. We hope it is OK if we therefore make this additional comment with the hope that the normal notifications will be sent out and that you all have the opportunity to view and respond to the updated version. Apologies if you have already received the associated notification(s).

Kind regards,

The authors

---

### Decision · Action_Editors · 2022-08-15

**Recommendation:** Reject

**Comment:**

The authors study the very general problem of the estimation of a function $\Psi(\mathcal{P})$ of a probability distribution $\mathcal{P}$ given a sample drawn from $\mathcal{P}$. Given an estimate $\hat{\mathcal{P}}$, they consider the first order approximation of $\Psi(\mathcal{P})-\Psi(\hat{\mathcal{P}})$. It can be written in terms of an influence function (IF, a central concept in robust statistics). They propose various methods to reduce this term (and in particular, to make its expectation close to $0$ so that $\Psi(\hat{\mathcal{P}})$ is an unbiased estimator). While their method is more general, it is essentially developped in the context of causal inference, more precisely, for the estimation of the ATE functional (Average Treatment Effect). A first method is described in Subsection 4.2, called MultiStep, is aimed to be an improvement of the existing One-Step method. A second method, MultiNet, is described by the authors as trying to "mimic the performance of an ensemble with a single neural network". It is discussed in Section 5, but I must say it is not clear to me what this method is actually doing.

The reviewers pointed out the following major problems:

((1)) the paper is poorly organized. It is very difficult to identify what the authors contribution actually is, and a take-home message from the paper. I will ellaborate on this:

1.1) the paper is far too long. It starts with "mini-tutorials" on causal inference and ATE, then on IF. The reviewers praised the quality of the writing of these mini-tutorials, but also pointed out the lack of connection between them. Quoing Reviewer eFLE: "while most individual paragraphs of this paper are fairly well-written and are of reasonable quality, at a macroscopic level, I feel that this paper is a bit of a mess". After 4 pages of introduction and discussion of previous works, these tutorials take 10 pages, which means that the contribution of the authors is not discussed before page 14. Some background on IFs and ATE is of course useful. The problem here is that it seems that the authors want to write as much as possible about IFs, at the risk of getting the reader lost or confused. It would be better to focus on the aspects of IFs that are necessary to understand your contribution. To this respect, I think that Equation (13) (first order expansion) + Equation (11) (writing the first order term in terms of an IF) is almost enough. I don't understand at all the "simple example" of Subsection 3.2.1 helps, in this case, the expecation of the term is already equal to 0...

1.2) the scope of the paper should be more focused. From the abstract and the introduction, it is a little difficult to see what is the scope of the proposed method. The introduction sounds a little like it can solve universally the tradeoff between parametric and nonparametric approaches.

1.3) after the very detailed tutorials of Section 3, it is a little disappointing that the method proposed by the authors are only sketched Subsection 4.2, and Section 5. To be honest, after reading the paper several times, I'm not sure I totally understand what MultiNet is actually doing.

I acknowledge that Reviewers GzoT and DiMQ were more positive on the writing of the paper. On the other hand, they point out as contributions of the paper: the tutorials on ATE and IF, and the experiments, but put little emphasis on MultiStep and MultiNet. Thus, it seems to me that the authors fail at conveying a clear message here.

((2)) Reviewers GzoT and DiMQ praised the empirical study; however, in the discussion phase, they also agreed that it is difficult to see what conclusion can be drawn from it, and that the improvement over existing methods is not clear.

Overall, my feeling after reading the paper many times is that the paper is not suitable for publication in its current state. It seems to me that ((1)) and ((2)) above justify the rejection of the paper. On the other hand, Reviewers GzoT and DiMQ are enthusiastic about the use of IFs in ATE estimation. They initiated a discussion with the authors, and I also want to acknowledge the work of the authors during the discussion phase. I will therefore recommend to reject the paper, but leave the door open for re-submission, and hope that the authors will submit a new version of the paper. The decision may seem harsh, but I don't believe that the contribution of the authors will receive the attention it deserves if some readers struggle to identify it.

I *strongly* encourage the authors to rewrite the paper entirely instead of trying to fix it. I totally agree with reviewer eFLE that the problem is not the "local" quality of the writing, but the organization of the paper. What follows is probably more a personal opinion, but I would start by a clear explanation of both new methods proposed: MultiStep and MultiNet. Once this is clearly written, I would insert before that the only facts about IFs that are absolutely necessary to understand these methods. Of course, theoretical results on the convergence of these methods would be awesome -- I understand that they might be beyond the scope of this paper. However, it is clear that all the theory presented in Section 3 requires smoothness assumption on $\Psi$, and moments assumptions on the data (what happens to the simple example in 3.2.1 if $y$ has an expecation, but no variance?). Thus, it seems necessary that the experiments convey a clear take-home message: what are the settings where your methods will work? and the settings where they will improve on existing methods? You can re-structure the experimental study taking into account the the suggestions of GzoT and DiMQ. I sincerely wish you good luck in the revision of the paper.